# The GEF Trio controls endothelial cell size and arterial remodeling downstream of Vegf signaling in both zebrafish and cell models

Alina Klems[1,12], Jos van Rijssel [2,12], Anne S. Ramms[1,3,12], Raphael Wild[1], Julia Hammer[1], Melanie Merkel[1], Laura Derenbach[1], Laetitia Préau[1], Rabea Hinkel [4], Irina Suarez-Martinez[5], Stefan Schulte-Merker[5], Ramon Vidal [6], Sascha Sauer [6], Riikka Kivelä [7], Kari Alitalo [8], Christian Kupatt [9], Jaap D. van Buul [2,10,12] & Ferdinand le Noble [1,3,11,12 ✉]

Arterial networks enlarge in response to increase in tissue metabolism to facilitate flow and nutrient delivery. Typically, the transition of a growing artery with a small diameter into a large caliber artery with a sizeable diameter occurs upon the blood flow driven change in number and shape of endothelial cells lining the arterial lumen. Here, using zebrafish embryos and endothelial cell models, we describe an alternative, flow independent model, involving enlargement of arterial endothelial cells, which results in the formation of large diameter arteries. Endothelial enlargement requires the GEF1 domain of the guanine nucleotide exchange factor Trio and activation of Rho-GTPases Rac1 and RhoG in the cell periphery, inducing F-actin cytoskeleton remodeling, myosin based tension at junction regions and focal adhesions. Activation of Trio in developing arteries in vivo involves precise titration of the Vegf signaling strength in the arterial wall, which is controlled by the soluble Vegf receptor Flt1.

[1] Department of Cell and Developmental Biology, Institute of Zoology (ZOO), Karlsruhe Institute of Technology (KIT), Fritz Haber Weg 4, 76131 Karlsruhe, Germany. [2] Molecular Cell Biology lab, Department Molecular and Cellular Hemostasis, Sanquin Research and Landsteiner Laboratory, Academic Medical Center at the University of Amsterdam, Plesmanlaan 125, 1066CX Amsterdam, The Netherlands. [3] Institute for Biological and Chemical Systems—Biological Information Processing, Karlsruhe Institute of Technology (KIT), PO Box 3640, 76021 Karlsruhe, Germany. [4] Laboratory Animal Science Unit, Leibnitz-Institut für Primatenforschung, Deutsches Primatenzentrum GmbH, Kellnerweg 4, 37077 Göttingen, Germany and DZHK (German Center for Cardiovascular Research), partner site Göttingen, Göttingen, Germany. [5] Institute of Cardiovascular Organogenesis and Regeneration WWU Münster, Münster, Germany & Faculty of Medicine, WWU Münster, Münster, Germany & Cells in Motion Cluster of Excellence, Münster, Münster, Germany. [6] Max Delbrück Center for Molecular Medicine (MDC), Berlin Institute of Medical Systems Biology & Berlin Institute of Health, Robert Rössle Strasse 10, 13092 Berlin, Germany. [7] Stem Cells and Metabolism Research Program, Research Programs Unit, Faculty of Medicine, University of Helsinki, and Wihuri Research Institute, Helsinki, Finland. [8] Translational Cancer Medicine Program, Research Programs Unit, Faculty of Medicine, University of Helsinki, and Wihuri Research Institute, Helsinki, Finland. [9] Klinik und Poliklinik für Innere Medizin I, Klinikum rechts der Isar, TUM Munich, Germany, and DZHK, (German Center for Cardiovascular Research), partner site Munich Heart Alliance, Munich, Germany. [10] Leeuwenhoek Centre for Advanced Microscopy, section Molecular Cytology at Swammerdam Institute for Life Sciences at University of Amsterdam, Amsterdam, The Netherlands. [11] Institute of Experimental Cardiology, University of Heidelberg, Heidelberg Germany and DZHK (German Center for Cardiovascular Research), partner site Heidelberg/Mannheim, Heidelberg, Germany. [12] These authors contributed equally: Alina Klems, Jos van Rijssel, Anne S. Ramms, Jaap D. van Buul, Ferdinand le Noble. ✉email: ferdinand.noble@kit.edu

The arterial vascular network distributes blood flow through the body, a process crucial for sustaining organ function and homeostasis. During embryonic development arterial networks enlarge and expand to meet the increasing metabolic demand of the growing and differentiating tissue. Also in ischemic cardiovascular diseases, revascularization, and regeneration of hypoperfused organs involve adaptations of the arterial system[1,2]. Selective targeting of the arterial growth process, and in particular creating arteries with a structurally large diameter to enhance flow conductance and relief hypoxia, is considered therapeutically relevant for treating patients with ischemic cardiovascular diseases[1,2]. The molecular and cellular mechanisms controlling arterial diameter are only partly understood but involve interactions between blood flow and endothelial cells, and BMP-Smad signaling controlling distinct endothelial cell behaviors during the vascular remodeling process[3–5].

An increase in flow promotes endothelial cell proliferation and migration collectively facilitating the transition of a small caliber vessel segment, with only few endothelial cells, into a larger caliber arterial segment, with many endothelial cells[6–9]. Flow activates the Akt-PI3-kinase signaling pathway responsible for endothelial proliferation. In addition, flow activates the BMP9-Alk1-Eng-Smad4 signaling pathway, which restricts flow-induced Akt activation and promotes endothelial quiescence[5,6,10–14]. The extent of diameter remodeling of arteries is determined by the amount of flowing blood and the shear stress setpoint[15]. Blood flow creates frictional forces exerted on endothelial cells, termed shear stress. Shear stress level has to be kept within strict boundaries to maintain endothelial quiescence and vessel stability[3]. A sustained increase in blood flow and shear stress promotes outward remodeling of the arterial lumen, resulting in a return of shear stress levels to the original setpoint. Vegf receptor-3/Flt4 is a component of the endothelial junctional mechanosensory complex, and acts as a determinant of the shear stress setpoint[16,17]. Lowering Flt4 expression and increasing the shear stress setpoint causes arteries to narrow thereby allowing shear stress to return to the setpoint[16]. In addition, increases in flow, after causing an initial diameter increase, have been shown to induce vessel contraction mediated via endothelial cell shape changes[10]. The transforming growth factor beta co-receptor Endoglin regulates vessel contraction, and loss of endoglin results in enlarged endothelial cells and blood vessel diameter in response to flow increases[10]. In adult pre-existing collateral arteries, flow-dependent arterial outward remodeling also involves activation of NFκB and inflammatory pathways in endothelial cells, leading to the recruitment of monocytes/macrophages that assist with remodeling by secretion of matrix metalloproteinases, and growth factors[18,19]. In *gridlock* mutant zebrafish embryos, macrophages contribute to flow driven outward remodeling of aortic collaterals, suggesting a developmentally conserved role for macrophages in arterial remodeling[20].

Cell shape is determined by the activity of small Rho GTPases. Their activity contributes to maintaining an equilibrium between the forces providing centripetal tension and forces ensuring cell spreading and avoidance of cell collapse[21–23]. Here we show that in endothelial cells, the guanine nucleotide exchange factor Trio, and activation of the small GTPases Rac1 and RhoG, trigger F-actin remodeling events in the endothelial cell periphery, increasing endothelial cell size. Arterial-specific expression of Trio in vivo augments endothelial cell size, resulting in functional arteries with a structurally larger lumen diameter, without change in endothelial cell numbers. Activation of Trio in vivo requires subtle fine-tuning of local arterial Vegf-Kdrl signaling levels, which is achieved by arterial Flt1 acting as a Vegf trap. Genetic targeting of the local arterial Flt1-Vegf balance results in endothelial cell enlargement, and significant outward arterial diameter remodeling, even during low flow conditions. Increases in vessel diameter reduce the resistance to flow. Trio-induced endothelial shape changes, and diameter remodeling in response to Vegf, may therefore aid to fine-tune local flow distribution in response to changes in tissue metabolism or hypoxia.

## Results

**Arterial Flt1 determines arterial diameter.** Vegf is an attractive candidate for targeting arterial caliber as it controls key aspects of arterial development[1,24,25], and is capable of activating small Rho-GTPases. Yet, beyond a narrow therapeutic window Vegf may induce adverse side-effects such as vessel overgrowth, e.g., hemangioma formation, and increased vessel permeability. How to deliver Vegf without deleterious side-effects is still an outstanding issue[26]. To fine-tune the spatio-temporal delivery of Vegf needed to obtain large arteries, we examined formation of arterial networks in the trunk of developing zebrafish embryos using different *vegfaa* gain of function scenarios.

We first employed transgenic embryos with constitutive or inducible expression of *vegfaa* under control of the somite muscle-specific *503unc* promoter[27] (here termed *vegfaa^musc*; Fig. 1a, b; Supplementary Fig. 1a–d). Constitutive *vegfaa* overexpression increased endothelial cell number and disrupted the arterial vasculature, abrupting blood flow perfusion when compared to wild-type (WT) (Fig. 1a, b). Similar results were obtained upon inducible *vegfaa* overexpression (Supplementary Fig. 1b, d, e–w). Because such *vegfaa* transgenic approaches resulted in Vegfaa overdose inappropriate for targeting the diameter of arteries, we next aimed at obtaining more subtle changes by manipulating Vegf protein bioavailability, which is determined by the Vegf decoy receptor Flt1.

Flt1 has a very high affinity for Vegf and mainly acts as a Vegf trapping receptor, limiting Vegf bioavailability and signaling through Kdrl[28,29]. Conversely, ablating *flt1* or displacing Flt1-trapped Vegf produces a Vegf gain-of-function scenario, with endogenous production of Vegf. We hypothesized that Flt1 can be used as a vehicle to deliver Vegf directly into growing arteries. For this approach to work efficiently, Flt1 protein must be expressed in close proximity to the Vegf signaling receptor Kdrl on arterial endothelium. We found that this is indeed the case (Fig. 1c–g). The developing arterial intersegmental vessels (aISV) displayed *flt1* promoter activity (Supplementary Fig. 2a–c) and expressed both *flt1* isoforms; the *membrane bound flt1 (mflt1)* and *soluble flt1 (sflt1)* (Supplementary Fig. 2d, e). To determine Flt1 protein distribution, we generated a knockin line, harboring two HA tags in exon 3 of the endogenous *flt1* locus, which labeled both mFlt1 and sFlt1 (Supplementary Fig. 2f). Immune staining of the transgene demonstrated Flt1 protein expression in the developing aISVs (Fig. 1c, d). To substantiate the distribution of the secreted, soluble Flt1 isoform around developing arteries, we furthermore examined protein distribution in a transgenic line expressing an engineered HA-tagged *sflt1* transgene with an inactive Vegfaa binding domain (*Tg(flt1^enh:sflt1_Δ7-HAHA)^ka612*, Supplementary Fig. 2g–n). Immune staining showed sFlt1 protein co-localizing with *kdrl* in arterial endothelium (Fig. 1e-g; Supplementary Fig. 2l–n). Transgenics with tagged wild-type *sflt1-HAHA* could not be used, because gain of *wt-sflt1-HAHA* decreased local Vegf[29] and inhibited aISV formation, thus rendering this experiment uninterpretable (Supplementary Fig. 2i–k).

Arterial ISVs, which express the Flt1 protein, are juxtaposed to Vegf-producing cells in developing somites[24]. Ablation of *flt1*, thereby creating a Vegfaa gain of function scenario[29], resulted in

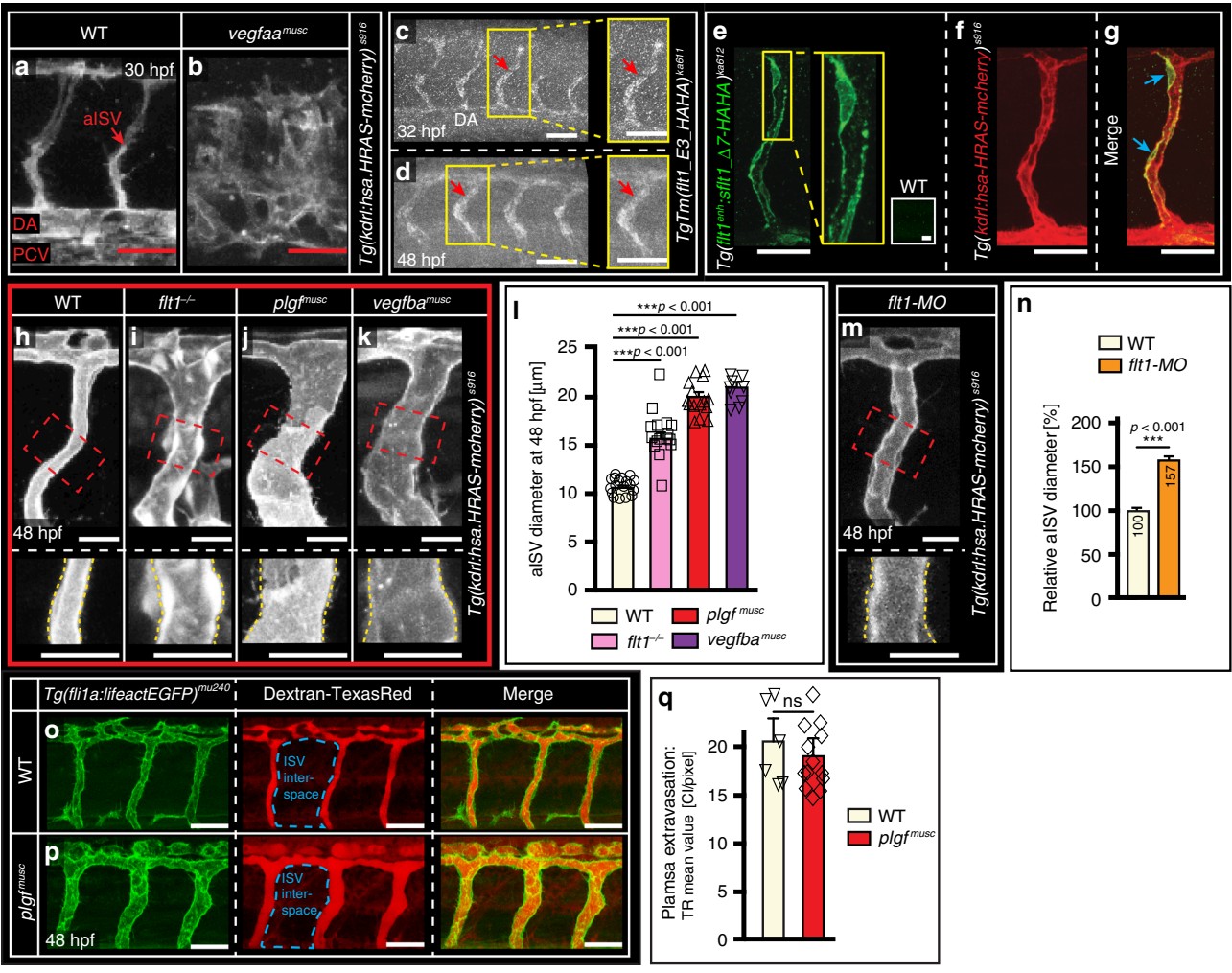

**Fig. 1 Arterial Flt1 determines vessel lumen dimensions. a, b** In vivo confocal imaging of trunk vascular architecture in WT (**a**) and *vegfaa^musc* transgenic embryos (**b**). Note disrupted vascular development in *vegfaa^musc* transgenics. **c, d** Whole mount immune staining with anti-HA antibody in *TgTm (flt1_E3_HAHA)^ka611* embryos at 32 hpf (**c**) and 48 hpf (**d**) to show Flt1 protein distribution. Arrows indicate aISVs and box shows zoom. **e–g** Immunostaining with anti-HA antibody showing sFlt1 protein distribution in *Tg(flt1^enh:sflt1_Δ7-HAHA)^ka612; Tg(kdrl:hsa.HRAS-mcherry)^s916* double transgenic embryos. aISV express sFlt1 protein (green; **e**); *kdrl* is shown in red (**f**) and merge shows colocalization of sFlt1 and *kdrl* in aISV (**g**, blue arrows). The white squared inset in **e** indicates control staining. **h–k** Confocal imaging of aISV in WT (**h**), *flt1^−/−* mutant, *flt1^ka601* (**i**), *plgf* GOF transgenic *plgf^musc* (**j**), and *vegfba* GOF transgenic *vegfb^musc* (**k**). Upper panels show overview; lower panels show detail of red boxed area. **l** Quantification of aISV diameter for indicated genotype; mean ± s.e.m, unpaired two-sided students t-test, n = 18, 16, 16, 10 aISVs per genotype. ***p < 0.001. **m** Confocal imaging of aISV in WT injected with *flt1* targeting morpholino. Area indicated by the red dotted box in the upper panel is displayed at higher magnification in the lower panel. **n** Relative aISV diameter change in embryos injected with *flt1* targeting morpholino (WT = 100%). mean ± s.e.m, unpaired two-sided students t-test, n = 20, 25 aISVs for indicated scenario, ***p < 0.001. **o, p** Imaging of plasma extravasation in WT and *plgf^musc* (n = 14, 16 embryos) injected with 70kD Dextran Texas-Red from 3 biologically independent experiments. **q** Quantification of images in o,p. mean ± s.e.m; unpaired two-sided students t-test, n = 6, 8 embryos per indicated group. ns, not significantly different. Scale bar, 50 μm in **a–g**, **o**, **p**; 25 μm in **h–k**, **m**. aISV intersegmental artery, DA dorsal aorta, PCV posterior cardinal vein. hpf hours post fertilization, MO morpholino, GOF gain of function.

a significant increase of aISV diameter (Fig. 1h–l). A similar result was obtained with a *flt1* targeting morpholino[29] (Fig. 1m, n).

High levels of the Flt1 ligands Plgf or Vegfb cause displacement of Vegf from Flt1 to Kdrl, creating a local Vegfaa gain of function scenario. In line with this competition model, *plgf* and *vegfba* gain of function transgenics (*Tg(503unc:eGFP-p2A-plgf)* here termed *plgf^musc*; and *Tg(503unc:eGFP-p2A-vegfba)* here termed *vegfba^musc*) showed significantly increased arterial lumen diameters (Fig. 1j, k, l; Supplementary Fig. 2o, p). On average aISV diameter increased 1.9 and 2.0 fold in *plgf^musc* and *vegfba^musc* embryos respectively (Fig. 1l). The larger aISVs were functional and not leaky (Fig. 1o–q; Supplementary Fig. 2q–t; Supplementary Movies 1–3). Membrane bound mFlt1 signaling was not

required for diameter growth in this setting; aISVs were significantly larger in *mflt1^−/−* mutants (Fig. 2a–d), and overexpression of *plgf* in *mflt1^−/−* mutants still resulted in a pronounced increase in arterial diameter when compared to WT or *mflt1^−/−* diameter (Fig. 2a–d).

While both loss of *flt1* and *plgf* gain of function induce a Vegf gain of function scenario, one component that may contribute to the difference in arterial diameter between *plgf^musc* embryos and *flt1^−/−* mutants involves the spatial distribution of arterial Flt1 and Kdrl receptors. In *plgf^musc*, Flt1, and Kdrl are both expressed in the same arterial ECs (Fig. 1g, Supplement Fig. 2u). Such expression pattern is absent in *flt1^−/−* mutants as Flt1 is not expressed. If Flt1-Kdrl co-expression in arterial EC indeed

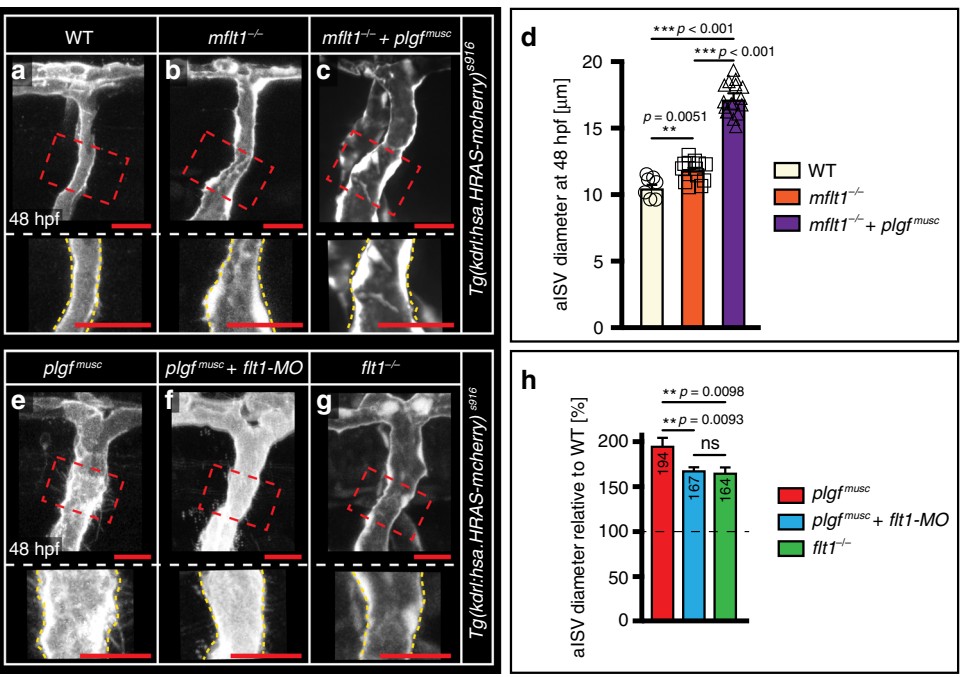

**Fig. 2 mFlt1 is not required for plgf-induced arterial diameter growth. a–c** In vivo confocal imaging of aISV in WT (**a**); *mft1⁻/⁻* mutant, *mflt1^ka605*, (**b**); *mft1⁻/⁻* combined with *plgf* gain of function *plgf^musc* (**c**); n = 58,64 and 57 aISVs per genotype derived from three autonomous experiments. The areas indicated by the red dotted boxes are displayed at higher magnification in the lower panels. **d** Quantification of images in **a–c**. Mean ± s.e.m, unpaired two-sided students *t*-test, n = 7,13, and 24 aISVs for indicated scenario. **p = 0.0051, ***p < 0.001. Note: significantly increased aISV diameter in *mflt1⁻/⁻+plgf^musc* scenario. **e–g** In vivo confocal imaging of aISV in *plgf^musc* (**e**), *plgf^musc* concomitant with morpholino knock-down of *flt1* (**f**; *plgf^musc + flt1-MO*), *flt1⁻/⁻* mutant, *flt1^ka601* (**g**). The areas indicated by the red dotted boxes are displayed at higher magnification in the lower panels. **h** Quantification of images in **e–g**. mean ± s.e.m, unpaired two-sided students *t*-test, n = 40,19, and 13 aISVs for indicated scenario. ns not significant, **p < 0.01. Scale bar; 25 μm in all panels. hpf hours post fertilization, aISV intersegmental artery, MO morpholino.

accounts for the observed difference, the model predicts that reducing *flt1* expression in *plgf^musc*, should annihilate the difference between *plgf^musc* and *flt1⁻/⁻* mutants. Nevertheless, arteries in *plgf^musc* transgenics injected with a *flt1* targeting morpholino should still show some degree of arterial diameter growth as the knockdown of *flt1* itself induces a *vegfaa* gain-of-function scenario, similar as in *flt1⁻/⁻* mutants. We tested these assumptions, and in-line with our hypothesis we indeed found that knockdown of *flt1* in *plgf^musc* embryos reduced arterial diameter growth, to the levels observed in *flt1⁻/⁻* mutants (Fig. 2e–h).

**Arterial diameter growth requires Vegf receptor-2 signaling.** Our data support the model in which excessive Plgf displaces trapped Vegfaa from arterial Flt1, thereby allowing the released free Vegfaa to bind arterial Kdrl receptors to induce signaling. In-line with Vegfaa-Kdrl signaling in *plgf^musc* embryos, we found that inhibiting Kdrl signaling or reducing *vegfaa* using a low dose *vegfaa* ATG targeting morpholino reduced lumen diameter growth in *plgf^musc* (Fig. 3a–f). Arterial ISVs express *kdrl* and the Vegf receptor-3, *flt4*, which can regulate Vegf receptor-2 signaling output[30]. Genetically ablating *flt4* or morpholino-mediated knockdown of *flt4* did not affect arterial diameter growth in *plgf^musc* or *flt1* loss of function embryos (Fig. 3g–k; Supplementary Fig. 3a–g). As Plgf may bind to neuropilin-1 receptors we examined *nrp1* mutants. Plgf increased aISV diameter in *nrp1a⁻/⁻* mutants (Supplementary Fig. 3h). Taken together these data indicate that Vegf-Kdrl is the main signaling pathway accounting for diameter growth in *plgf^musc*.

Developing aISVs in *plgf^musc* were enlarged prior to formation of a functional perfused trunk vascular network (Fig. 3l–n). To rule out increased flow as a trigger for Plgf-induced arterial

remodeling, we next exposed aISVs of *plgf^musc* embryos to low flow conditions. To reduce heart rate and trunk perfusion, we treated WT and *plgf^musc* embryos with the L-type calcium channel blocker nifedipine[10,16] (Fig. 3o–r). WT embryos exposed to nifedipine showed reduced aISV and aorta diameter (Fig. 3o–r; Supplementary Fig. 4a–d). However, *plgf^musc* embryos exposed to nifedipine still showed a significant increase in aISV diameter when compared to WT with normal flow, or WT with nifedipine (Fig. 3r). Similar observations were made in *tnnt2* morphants (Fig. 3s–v; Supplementary Fig. 4e–g). Reducing cardiac contractility by knocking down cardiac troponin T2 (*tnnt2*) is an established approach for creating a silent heart and block trunk perfusion (Supplementary Fig. 4h–l). *Tnnt2* morphants showed a reduced aISV size and aortic diameter (Fig. 3s–v; Supplementary Fig. 4h–k). However, despite loss of flow, aISV size was still larger in *plgf^musc* embryos injected with *tnnt2* targeting morpholino, when compared to WT with flow, or WT injected with *tnnt2* morpholino (Fig. 3s–v); while *tnnt2* morphants showed deficits in lumen formation (Fig. 3t), *plgf^musc*—*tnnt2* morphants showed a clear lumen on confocal sections of 3D stacks (Fig. 3u).

As mouse studies attribute the proarteriogenic effect of Plgf to activation of mFlt1 signaling in macrophages, we decided to investigate Plgf in a macrophage loss-of-function model[31]. Reducing macrophage differentiation (using a triple morpholino knockdown of *pu.1*, *gcsfr*, and *irf8*) had no effect on aISV diameters in *plgf^musc* gain of function embryos (Supplementary Fig. 5a–l). Taken together, our data suggest that Plgf-induced aISV remodeling is neither triggered by increased shear stress nor macrophages.

**Endothelial size contributes to diameter remodeling in vivo.** Theoretically, outward arterial diameter remodeling in the

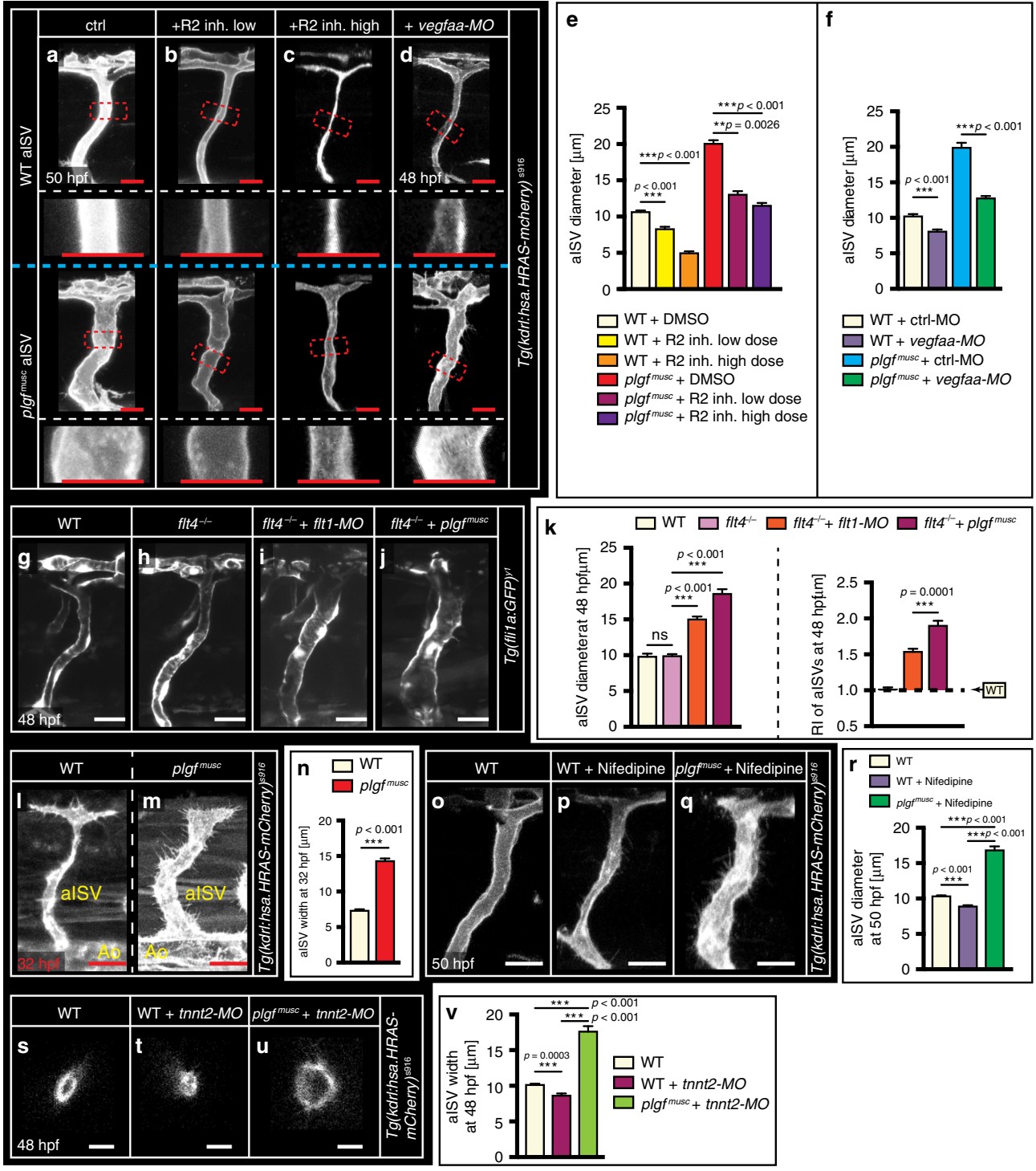

developing aISV can be achieved in three ways: by having more endothelial cells, larger endothelial cells or a combination of both. We detected the combination of both: more and enlarged endothelial cells in aISVs upon *plgf* gain-of-function when compared to WT (Fig. 4a–e; Supplementary Fig. 6a–f; Supplementary Movies 4 and 5). Analyses of life-act reporter embryos showed a significant increase in total endothelial cell surface area and in surface area expansion rate, when comparing *plgf*^*musc* embryos to WT (Fig. 4e; Supplementary Fig. 6g; Supplementary Movies 4 and 5). We reasoned that RhoGTPase activity and changes in F-actin remodeling could contribute to the in vivo phenotypic changes in endothelial cell shape.

To block F-actin polymerization, we administered the actin polymerization inhibitor Latrunculin B (LatB)[32] in vivo. LatB significantly reduced aISV diameter growth in *plgf*^*musc* embryos when compared to vehicle treated *plgf*^*musc* embryos (Fig. 4f; Supplementary Fig. 6h–p). Major aspects of intracellular F-actin dynamics are regulated through the small Rho GTPases family, including Rac1 and RhoG[33]. The Rac1 inhibitor CAS 1177865-17-6 significantly inhibited aISV diameter expansion in *plgf*^*musc* embryos (Fig. 4f; Supplementary Fig. 6h–p). The RhoGEF (Rho-Guanine nucleotide Exchange Factor) Trio is well recognized for its ability to activate Rac1 and RhoG[34] through its GEF1 domain. ITX3, a selective inhibitor of the Trio N-terminal RhoGEF

**Fig. 3 Vegfaa—Vegf receptor-2/Kdrl signaling drives outward arterial lumen remodeling. a–d** aISV in WT (upper panels) and plgf[musc] embryos (lower panels) treated with DMSO vehicle - control (**a**), low dose Vegf-R2 inhibitor (**b**), high dose Vegf-R2 inhibitor (**c**), or vegfaa ATG targeting morpholino (**d**). Red dotted box is displayed at higher magnification in panel below. **e** Quantification of aISV diameter at 50 hpf upon R2 inhibition. Mean ± s.e.m, one-way ANOVA, and post-hoc bonferroni, $n = 18, 20, 20, 15, 20, 20$ aISVs for indicated scenarios. $**p = 0.0026$, $***p < 0.001$. **f** Quantification of aISV diameter at 48 hpf upon morpholino-mediated knockdown of vegfaa. Mean ± s.e.m, one-way ANOVA, and post-hoc bonferroni, $n = 15, 22, 14, 28$ aISVs for indicated scenarios. $***p < 0.001$. **g–j** In vivo confocal imaging of WT (**g**), flt4[−/−] mutant, flt4[mu407](**h**), flt4[−/−] mutant injected with flt1 ATG targeting morpholino (**i**), flt4[−/−] mutant injected with plgf[musc] plasmid (**j**). **k** Quantification of images in **g–j**. Left panel, absolute aISV diameter ($n = 21, 16, 42, 34$ aISVs per indicated genotype). Right panel, aISV diameter change relative to WT. Mean ± s.e.m, unpaired two-sided students t-test. ns not significant, $***p ≤ 0.001$. **l, m** In vivo confocal imaging of developing aISVs prior to the onset of aISV perfusion in WT (**l**) and plgf[musc] embryos (**m**). Note: larger aISV prior to onset of flow in plgf[musc]. **n** Quantification of images in **l, m**. Mean ± s.e.m, unpaired two-sided students t-test, $n = 16$ aISVs/group. $***p < 0.001$. **o–q** Imaging of WT (**o**), WT treated with L-type calcium channel blocker nifedipine (**p**), and plgf[musc] treated with nifedipine (**q**). **r** Quantification of images in **o–q**. Mean ± s.e.m, unpaired two-sided students t-test, $n = 11, 20$, and 15 aISVs per treatment group. $***p < 0.001$. **s–u** Confocal cross-section to show lumen dimensions of aISV in WT (**s**), WT injected with tnnt2 targeting morpholino (**t**), and plgf[musc] embryo injected with tnnt2 targeting morpholino (**u**). **v** Quantification of images in **s–u**. Mean ± s.e.m, unpaired two-sided students t-test, $n = 20$ aISVs per group. $***p < 0.001$. Scale bar: 25 µm in all images. aISV intersegmental artery, MO morpholino, hpf hours post fertilization, RI remodeling index, tnnt2 cardiac muscle troponin T2.

(GEF1) domain[35] significantly reduced the outward remodeling response of aISVs (Fig. 4f; Supplementary Fig. 6h–p). To further substantiate the contribution of Trio in this setting we examined loss of Trio in WT and plgf[musc] embryos using a morpholino approach (Fig. 4g–t; Supplementary Fig. 7a–c). Morpholino-mediated knockdown of Trio resulted in a dose-dependent decrease of endothelial cell size and aISV diameter in WT and plgf[musc] embryos (Fig. 4m, t).

**Endothelial enlargement requires the GEF1 domain of Trio.** Rho GTPases control actin reorganization, and GEFs like Trio activate Rho GTPases by promoting their exchange of GDP for GTP. Trio is a unique Rho GEF, because it has two separate GEF domains, GEF1 and GEF2, that control the GTPases RhoG/Rac1 and RhoA, respectively[36]. To substantiate which GEF domain of Trio is functionally relevant for cell enlargement in our setting, we performed gain-of-function experiments with Trio deletion constructs[37]. In vitro overexpression of a truncated Trio form containing the spectrin domain and GEF1 domain (termed TrioN), but lacking the GEF2 domain, resulted in significantly larger endothelial cells compared to control-transfected cells (Fig. 5a–d, quantification in Fig. 5g). In further support of functional requirement of the GEF1 domain we found that overexpression of the GEF1 domain only also induced endothelial cell enlargement (Fig. 5e,f quantification in Fig. 5g). Besides full length Trio, 4 additional Trio-GEF1 containing isoforms (Trio-A, B, C, D) arising from alternative splicing have been described[38]. In HUVEC, all GEF1 containing isoforms were detectable but full length Trio was the most abundant Trio form (Supplementary Fig. 8a). The Trio-B isoform is truncated at the C-terminus and similar to TrioN only contains the GEF1 domain. In support of the GEF1 domain, endothelial overexpression of Trio-B increased endothelial size (Supplementary Fig. 8b).

The GEF1 domain of Trio can activate both Rac1 and RhoG. Endothelial cells transfected with constitutively active Rac1, or a photo-activatable Rac1 probe[39] (Rac1-PA-WT) that was activated by light showed a significantly larger size when compared to mCherry-expressing control cells (Fig. 5h–l), whereas a catalytic-dead mutant of the photo-activatable Rac1 probe (Rac1-PA-C450A) failed to induce a cell size change upon exposure to light (Fig. 5h–l). We found no evidence for a functional role of the Rac1B splice-isoform (Supplementary Table 1; Supplementary Fig. 8c–g). Endothelial overexpression of a dominant active RhoG form (GFP-RhoG-Q61L) significantly increased endothelial cell area (Fig. 5m). Conversely, silencing of Rac1 or RhoG in endothelial cells overexpressing TrioN significantly reduced the TrioN-induced cell size increase (Fig. 5n–p; Supplementary Fig. 8h). Unfortunately, simultaneous silencing of both Rac1

and RhoG in TrioN expressing cells resulted in massive endothelial cell death.

The in vivo data suggest that Vegf signaling can activate Trio. Exposing endothelial cells to VEGF in vitro, triggered a rapid and transient activation of Rac1 (Supplementary Fig. 8i–l). Activation of Rac1 involved Trio as shRNA-mediated silencing of Trio using two different shRNAs reduced VEGF-induced Rac1 activation (Supplementary Fig. 8i–l). Besides Trio, there are a number of other known regulators for Rac1, including Tiam1, and multiple regulators may coordinate in regulating the function of Rac1 in this context. In endothelial cells overexpressing TrioN, we observed that Tiam1 was expressed in junctional regions, as opposed to the more cytosolic localization observed in control-transfected cells (Fig. 5q–v). Endothelial overexpression of Tiam1 resulted in larger endothelial cells, similar to the effects observed upon overexpression of TrioN (Fig. 5w). Combining TrioN and Tiam1 overexpression had no additive effect on increasing EC size when compared to TrioN alone (Fig. 5w). These data suggest that one part of cell enlargement may require careful spatial positioning of GEFs, and activation of RhoGTPases in junctional regions.

**Trio activates Rac1 in junctional regions.** Using a Rac1 FRET-based biosensor we found that Trio activated Rac1 in particular at endothelial cell junctional regions (Fig. 6a, b). TrioN and Trio-GEF1 were equally potent in activating Rac1 at junction regions (Fig. 6c, d; Supplementary Fig. 9a–c). Mutating the functional GEF1 domain in TrioN (TrioNmut[40]) abrogated Rac1 activation (Fig. 6c, d; Supplementary Fig. 9d). TrioN-transfected cells showed formation of lamellipodia, and enlargement of the endothelial cells along the front of the lamellipodia extensions (Supplementary Fig. 9e–g; Supplementary Movie 6). Control-transfected cells also showed lamellipodia formation, but in control cells these lamellipodia more often retracted and no subsequent enlargement was observed (Supplementary Fig. 9h–j; Supplementary Movie 7). TrioN significantly increased the life-time of lateral lamellipodia when compared to transfected cells; accordingly the number of protrusions per hour was reduced in TrioN-transfected cells (Fig. 6e, f). After 17 h post-TrioN transfection, no additional enlargement was observed and the VE-Cadherin expression pattern was reminiscent of a honeycomb structure, suggestive of a force-equilibrium between the interacting cells[41].

Actomyosin complexes and myosin-II-mediated contractility contribute to tension development[42,43]. Using GFP-tagged Myosin II we found that Myosin II localized on F-actin bundles throughout the cell but was most prominently present on junctional F-actin bundles upon TrioN transfection (Fig. 6g, h).

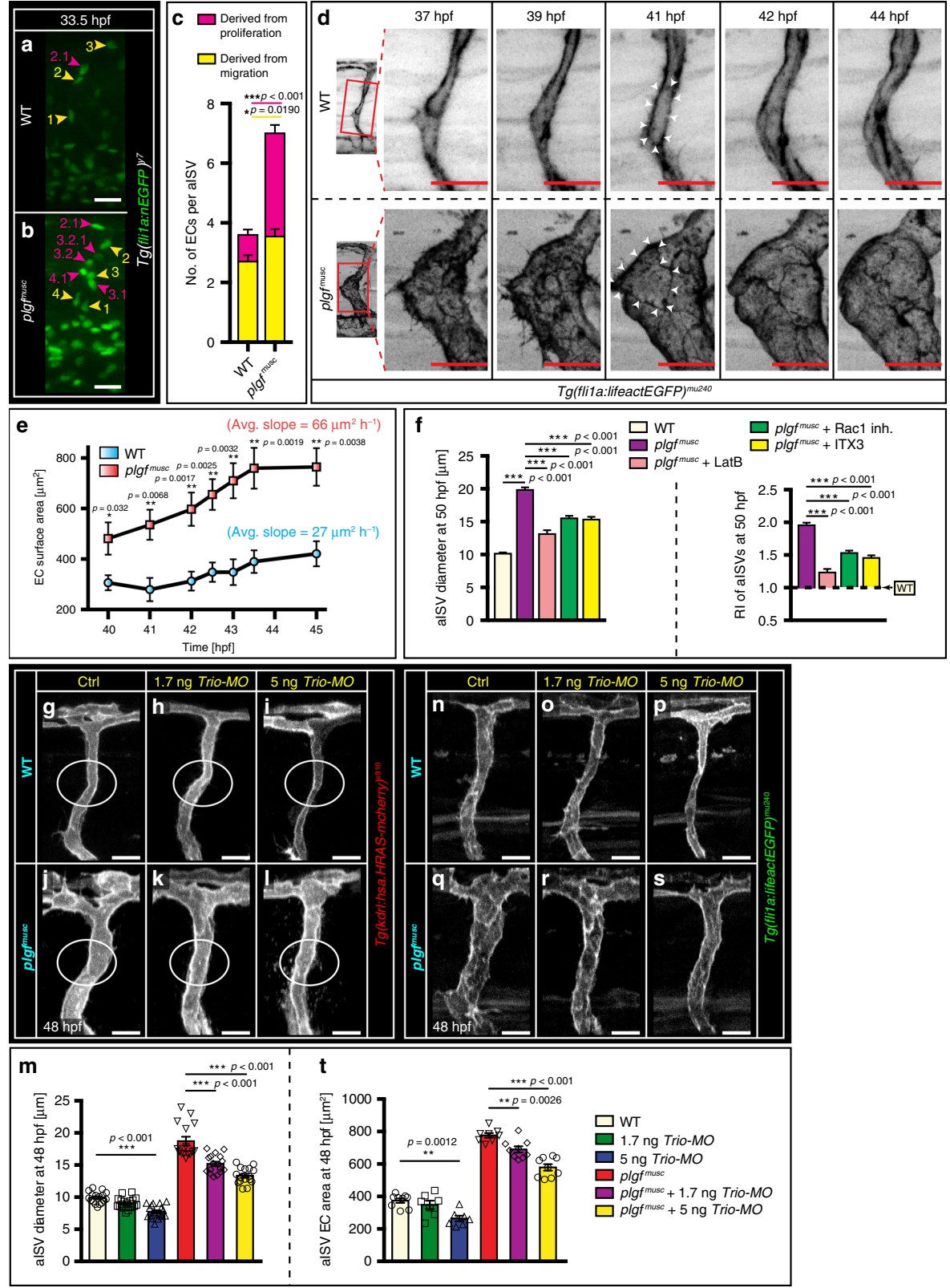

Also, the majority of active myosin, distinguished by staining for mono-phosphorylated myosin light chain S19, localized strongly at the junction region upon *TrioN* overexpression (Fig. 6i, j). In addition, we observed that fully di-phosphorylated myosin light chain (T18/S19) localizes even more specifically to junctional F-actin bundles (Fig. 6k, l). Inhibition of myosin activity from 1 h after *TrioN* transfection significantly abrogated Trio-induced endothelial cell size enlargement (Supplementary Fig. 9k–m). Inhibiting myosin activity in already enlarged endothelial cells resulted in the formation of gaps in particular at tri-cellular junctions (Supplementary Fig. 9n, o).

We next reasoned that Trio-induced changes in myosin-II and cytoskeletal remodeling affect the tension distribution along the F-actin cytoskeleton. To test this, we examined local tension at

**Fig. 4 Plgf-induced arterial diameter growth requires the Rho GEF Trio. a**, **b** endothelial cell (EC) nuclei distribution in green (eGFP) during aISV formation in WT (**a**) and in *plgf^musc* embryos (**b**). ECs derived from migration events numbered in yellow, from proliferation events in purple. **c** Quantification of migration (yellow) and proliferation (purple) events during aISV formation; mean ± s.e.m, unpaired two-sided students *t*-test, *n* = 10,11 aISVs for indicated genotype. *\*p* = 0.0190, *\*\*\*p* < 0.001. **d** Time-lapse imaging in *Tg(fli1a:lifeactEGFP)^mu240* showing endothelial actin in aISV of WT (upper panels) and *plgf^musc* embryos (lower panels). Red box in the left panel at higher magnification in the right panels. Arrowheads at 41hpf delineate an individual EC. **e** EC surface area changes in WT (blue circles) and *plgf^musc* embryos (red squares); mean ± s.e.m, unpaired two-sided students *t*-test, *n* = 6 ECs for each genotype. **f** Left panel, arterial ISV diameter in *plgf^musc* embryos (magenta bar) treated with the actin polymerization inhibitor Latrunculin B (LatB, pink bar), Rac1 inhibitor CAS 1177865-17-6 (green bar), Trio inhibitor ITX3 (yellow bar). Right panel, RI upon indicated inhibitor treatment. Mean ± s.e.m, one-way ANOVA, post-hoc bonferroni, *n* = 25, 38, 30, 60, 59 aISVs for indicated condition. *\*\*\*p* < 0.001. **g–i** aISV in WT (**g**), WT injected with 1.7 ng *Trio* targeting morpholino (**h**), and WT injected with 5 ng *Trio* targeting morpholino (**i**). **j–l** aISV in *plgf^musc* (**j**), *plgf^musc* injected with 1.7 ng *Trio* targeting morpholino (**k**), and *plgf^musc* injected with 5 ng *Trio* targeting morpholino (**l**). **m** Quantification of images in **g–l**. Mean ± s.e.m, ANOVA & post-hoc bonferroni, n = 17, 21, 18, 15, 17, and 16 aISVs/group. *\*\*\*p* < 0.001. **n–p** aISV in WT (**n**), WT injected with 1.7 ng *Trio* targeting morpholino (**o**), and WT injected with 5 ng *Trio* targeting morpholino (**p**). **q–s** aISV in *plgf^musc* (**q**), *plgf^musc* injected with 1.7 ng *Trio* targeting morpholino (**r**), and *plgf^musc* injected with 5 ng *Trio* targeting morpholino (**s**). **t** Quantification of images in **n–s**. Mean ± s.e.m, ANOVA & post-hoc bonferroni, *n* = 18, 14, 14, 16, 18, and 18 aISVs per group. ns not significant, *\*\*p* < 0.01, *\*\*\*p* < 0.001. Scale bar, 25 μm in all panels. aISV intersegmental artery, MO morpholino.

junctional, cortical, and central F-actin bundles, in *TrioN* and control-transfected endothelial cells by laser-ablating single F-actin bundles (Fig. 6m–s). As a read out for local tension, we measured the recoil of the actin structures upon ablation (Fig. 6p)[44]. Transfection of *TrioN* only increased the tension on junctional actin bundles, and not on cortical or central bundles (Fig. 6q–s). Taken together, these findings suggest that TrioN gain-of-function preferentially augmented tension at junctional F-actin bundles.

**VE-Cadherin is not required for Trio-induced EC enlargement.** Trio interacts with VE-Cadherin[45], and endothelial cells gain stability by making contacts with neighboring cells through VE-cadherin–based cell–cell junctions. Endothelial monolayers transfected with *TrioN* in vitro showed augmented junctional integrity as indicated by increased electrical cell-substrate impedance (ECIS) (Fig. 7a). Interestingly, this increase in junctional strength was independent of VE-Cadherin, as *TrioN* over-expression augmented electrical cell-substrate impedance even in the absence of *VE-Cadherin* (Fig. 7b, c). Other junctional molecules may compensate for loss of VE-Cadherin. We found no substantial changes in N-Cadherin expression at junctions in endothelial cells in which overexpression of *TrioN* was combined with silencing of *VE-Cadherin* (Supplementary Fig. 10a–l).

To test if VE-Cadherin-based cell–cell junctions are functionally relevant for the *TrioN*-induced enlargement of endothelial cells, we overexpressed *TrioN* while simultaneously silencing endothelial *VE-Cadherin* (Fig. 7d-m). Loss of *VE-Cadherin* had no significant impact on *TrioN*-induced endothelial cell enlargement (Fig. 7m). To test if coordination between adjacent endothelial cells is critical for endothelial enlargement in vitro, we performed mosaic experiments by mixing *TrioN* overexpressing endothelial cells with control-transfected endothelial cells (Supplementary Fig. 10m), and compared cell size with a scenario in which all (neighboring) cells were expressing TrioN (homogeneous expression scenario). In both the mosaic and in the homogeneous scenario we find increased EC size in *TrioN* overexpressing cells (Fig. 7n). The magnitude of cell size increase was not significantly different between the two scenarios (Fig. 7n).

Endothelial cells anchor to the extracellular matrix (ECM) through focal adhesions. In endothelial lamellipodia extensions, anchoring to the ECM is necessary for the buildup of tractional forces that regulate forward extension and cell shape. We hypothesized that to maintain their shape, the enlarged endothelial cells have to be able to stably anchor to the ECM. In line with this, we found that *TrioN* massively increased the number of focal adhesions, in particular near cell–cell junctions, as judged by phosphorylated Paxillin staining (Fig. 7o–s). *TrioN*

overexpressing cells furthermore showed a shift in the expression of α5 Integrin and β1 Integrin toward junctional regions (Fig. 7t–w).

**Trio augments EC size and arterial diameter in vivo.** To test if Trio can augment endothelial cell size and arterial dimensions in vivo in zebrafish embryos, we generated a transgenic line and overexpressed *TrioN* under the control of the *flt1^enh* promoter (*Tg(flt1^enh:GFP-TRION)* here termed *flt1^enh:TrioN*) (Fig. 8a–g). The *flt1^enh* promoter is mainly active in arterial ISVs. Overexpression of *TrioN* in vivo resulted in significantly larger aISVs (Fig. 8e) without affecting endothelial cell numbers (Fig. 8g). Instead, and in-line with our in vitro observations, *TrioN* increased endothelial surface area in aISVs (Fig. 8f). The aISV diameter increase was therefore solely attributed to larger endothelial cells. The larger aISVs upon *TrioN* overexpression were perfused and nonleaky as evidenced from dextran extravasation analyses (Supplementary Fig. 11a–d).

We next overexpressed *TrioN* under control of the *flt1^enh* promoter in *plgf^musc* embryos (Fig. 8h–o). Overexpression of *TrioN* in *plgf^musc* embryos resulted in significantly larger aISVs and endothelial surface area, when compared to *plgf^musc* alone (Fig. 8m, n, o, u). On an average, combining *plgf* and *TrioN* resulted in 2.5-fold larger aISV diameters (Fig. 8o); whereas in *plgf^musc* embryos the increase was 2-fold (Fig. 8o). The enlarged aISVs in *plgf^musc* + *TrioN* gain of function embryos were actively perfused (Supplementary Movie 8). Overexpression of *TrioNmut* (TrioN with inactive GEF1 domain) in *plgf^musc* transgenics failed to produce any additive increase in aISV diameter or endothelial surface area (Fig. 8o). Taken together, these data suggest that combining *plgf* with artery specific *Trio* transgenic gain-of-function results in arteries having multiple enlarged endothelial cells, which additively facilitate the arterial outward lumen remodeling.

We furthermore found that VE-Cadherin was not required for *TrioN*-induced endothelial cell size increase in vivo. Overexpression of *TrioN* in aISVs of *ve-cadherin* morphants (*cdh5-MO*) or *ve-cadherin^−/−* mutants augmented endothelial area at a similar magnitude as observed in *TrioN* gain-of-function embryos with intact *ve-cadherin* expression (Fig. 8p–t; Supplementary Fig. 11e).

We next examined if the effect of Trio on arterial diameter in vivo and cell size in vitro is conserved when using alternative vascular promoters or endothelial cells derived from different anatomical origins, respectively (Supplementary Fig. 11f–h). The *kdrl* and *fli1a* promoter are ubiquitously expressed in endothelial cells of the zebrafish embryo trunk vasculature. For both *kdrl* and *fli1a* we observed significantly increased aorta and aISV

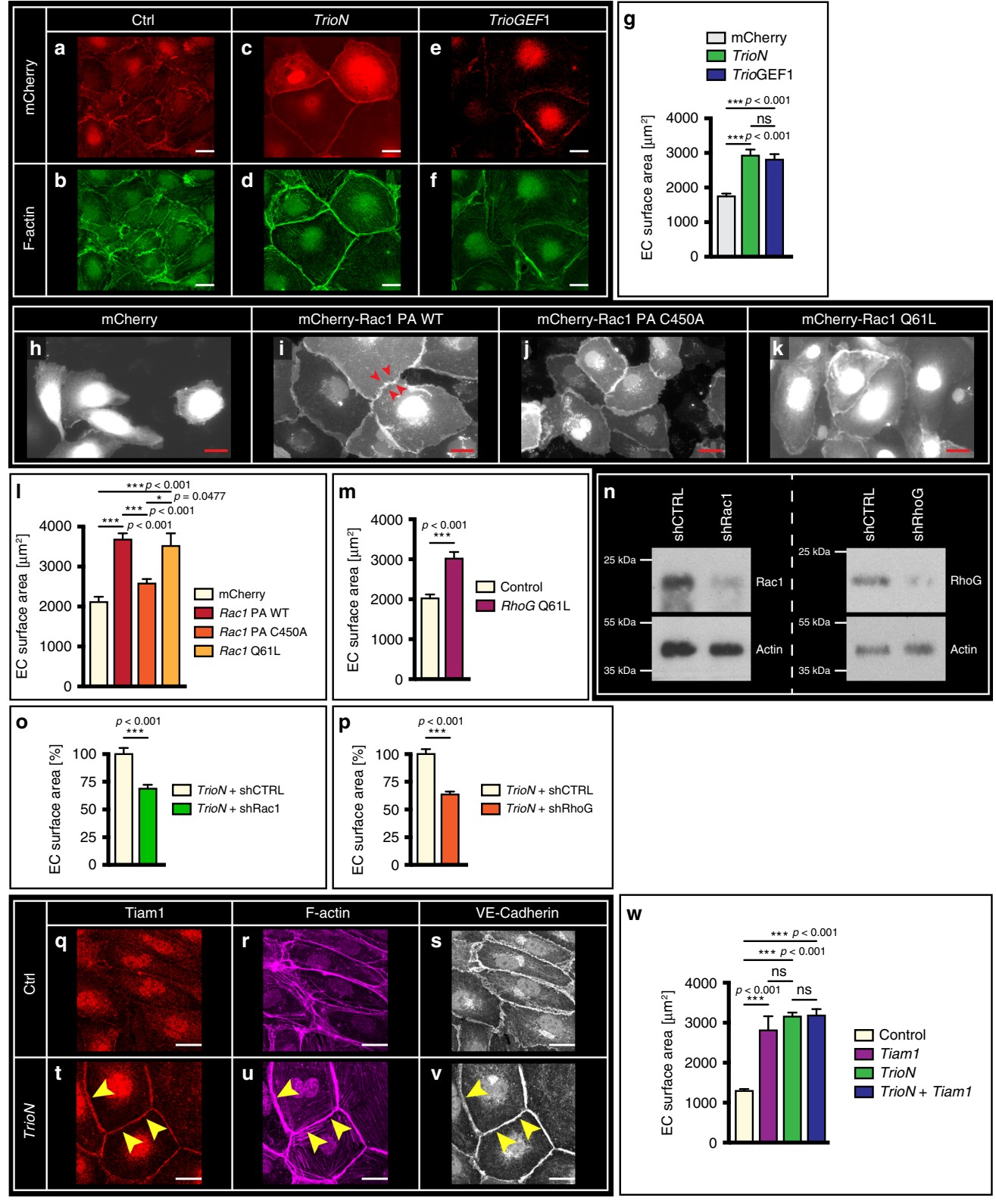

diameters, and EC size, in WT and in *plgf^musc* embryos (Fig. 8v, w, Supplementary Fig. 11g, h). In vitro, human aorta (HAEC) and human umbilical artery derived endothelial cells (HUAEC) showed a significantly increased size upon *TrioN* overexpression (Supplementary Fig. 11f). As Vegf is the major driver of Trio activation in our setting, we hypothesized that our findings regarding enlargement may be conserved in other *vegf* gain of function scenarios. Von Hippel Lindau (Vhl) is a protein relevant

for probing Hypoxia Inducible Factor-1α (HiF-1α), the main driver of *vegf* expression, for proteasomal degradation. *Vhl^{−/−}* mutants show increased *vegfaa* expression, in particular in developing neuronal tissue[29]. Accordingly, *vhl^{−/−}* mutants showed increased cerebral arterial diameters, without significant change in endothelial cell numbers (Supplementary Fig. 11i–p). *Vhl* morphants also showed increased vessel diameter, concomitantly with increased EC surface area (Supplementary

**Fig. 5 Trio-GEF1 domain targets Rac1 and RhoG increase endothelial cell size. a–f** Confocal images showing mCherry (**a, c, e**) and F-actin distribution (**b, d, f**) in ECs transfected with *mCherry*, *mCherry-TrioN* (*TrioN*), or *mCherry-TrioGEF1* (*TrioGEF1*). Scale bar, 25 μm. **g** Endothelial cell surface area in ECs expressing *mCherry*, *mCherry-TrioN*, or *mCherry-TrioGEF1*. Mean ± s.e.m, unpaired two-sided students *t*-test, $n = 59, 69, 63$ cells per indicated group. ***$p <$ 0.001. **h–k** Confocal images of ECs transfected with *mCherry*, photo-activatable Rac1 (*mCherry-Rac1-PA-WT*), mutated photo-activatable Rac1 (*mCherry-Rac1-PA-C450A*), or constitutively active Rac1 construct (*mCherry-Rac1-Q61L*). Red arrowheads indicate cell–cell junctions. Scale bar, 25 μm. **l** Quantification of images in **h–k**. Mean ± s.e.m, two-sided Mann–Whitney U test, $n = 67, 56, 65, 74$ cells per group. *$p = 0.0477$, ***$p < 0.001$. **m** Surface area of ECs transfected with control plasmid or constitutive active RhoG Q61L. Mean ± s.e.m, unpaired two-sided students *t*-test, $n = 89, 83$ cells per group, from three independent experiments. ***$p < 0.001$. **n** Western blots for Rac1 and RhoG; control of knockdown efficiency for shRNA *shRac1* and *shRhoG* as used in O, P and actin for protein loading control, as indicated. **o** Change in EC size upon *TrioN* overexpression after silencing of *Rac1*. Mean ± s.e.m, unpaired two-sided students *t*-test, $n = 73, 97$ cells from three independent experiments. ***$p < 0.001$. **p** Change in EC size upon *TrioN* overexpression after silencing of *RhoG*. Mean ± s.e.m, unpaired two-sided students *t*-test, $n = 101, 146$ cells per group from three independent experiments. ***$p < 0.001$. **q–s** Endothelial cells transfected with control plasmid (ctrl) and stained for Tiam1 (**q**), F-actin with phalloidin (**r**) and VE-Cadherin as junction marker (**s**). Tiam1 localizes cytosolic and to a lesser extent to junction regions. Scale bar, 20 μm. **t–v** Endothelial cells transfected with *TrioN* and stained for Tiam1 (**t**), F-actin with phalloidin (**u**) and VE-Cadherin as junction marker (**v**). Tiam1 localizes at junction regions (arrowheads). Scale bar, 20 μm. **w** Endothelial cell size of ECs transfected with *Tiam1*, *TrioN*, or transfected with both *Tiam1* and *TrioN*. Cell size was measured of $n = 78, 82, 103$ and 80 cells derived from three separate experiments. Mean ± s.e.m, unpaired two-sided students *t*-test. ns not significant, ***$p < 0.001$.

Fig. 11q–u). Blocking Trio function using ITX3 inhibited the diameter increase in *vhl* morphants (Supplementary Fig. 11s, u). This suggests that Trio-mediated cell and vessel dimension changes is conserved in this Vegf gain of function scenario.

In zebrafish embryos, loss of the TGFβ co-receptor *endoglin* augments aortic diameter in response to flow increases[10]. Morpholino-mediated loss of *endoglin* increased aorta diameter at 3dpf, not 2dpf, in-line with the *endoglin* mutant phenotype (Supplementary Fig. 12a–r, u). We found no evidence for loss of *endoglin* and gain of *Trio* acting synergistically to influence aISV or aortic diameters in 2dpf embryos (Supplementary Fig. 12a–i; Supplementary Fig. 12j–r). In vitro, cell size of *TrioN* over-expressing cells was not significantly different from the size of *TrioN* expressing cells in which *endoglin* was silenced (Supplementary Fig. 12s, t). At 3dpf *endoglin* morphants showed an increased aorta diameter similar to the phenotype reported in *endoglin* mutants (Supplementary Fig. 12u)[10]. Since in the *endoglin* mutant the change in aorta diameter was attributed to an increase in EC size and not EC number, we considered a potential contribution of Trio in mediating the diameter increase. Accordingly, loss of *Trio* in *endoglin* morphants, or inhibiting Trio function using ITX3 in *endoglin* morphants inhibited the aorta diameter increase and reduced EC surface area (Fig. 9a, b). This suggests that at 3dpf the TGFβ pathway may act as an inhibitor of Trio function and diameter control.

## Discussion

Here we identified the GEF Trio as a central signaling hub in driving endothelial cell enlargement and outward arterial lumen diameter remodeling, downstream of Vegf receptor-2 signaling. Endothelial cell enlargement requires the GEF1 domain of Trio, and activation of the Trio-GEF1 targets Rac1 and RhoG, inducing F-actin remodeling events, focal adhesions, and actomyosin tension specifically at the cell periphery. Actomyosin-activity induced tension at extracellular matrix anchors in lamellipodia extensions is subsequently conveyed into a homothetic expansion of the endothelial cell. Actin remodeling in the endothelial cell periphery requires activation of Rho GTPases specifically in junctional regions. In line with this, we show that Trio activates Rac1 in the cell periphery. Active Trio is expressed at junctional regions, and Trio also directs other GEFs including Tiam1 to the junctions. Tiam1 has thus far not been linked to Vegf receptor signaling. Tiam1 is a known activator of Rac1, and we show that Tiam1 promotes endothelial cell size, similar to Trio. Combining Trio and Tiam1 overexpression has no additive effect on augmenting endothelial cell size, thus leaving open the option that Trio and Tiam1 may act in a linear way to induce Rac1 activity at

junctional regions. Trio and Tiam1 are not the only GEFs that can activate Rac1 and it is conceivable that activation of additional Rac1 pools, at different intracellular locations and coupled to other downstream effectors, also contributes to the complex process of cell enlargement.

Besides activation of Rac1, the GEF1 domain of Trio is also recognized for its ability to exchange GTP on the related small GTPase RhoG. Similar to Rac1, we show that RhoG gain-of-function results in larger endothelial cells, while loss of RhoG reduces Trio-induced cell size. How RhoG contributes to endothelial cell size increase may however differ from Rac1, as RhoG is dispensable for lamellipodia formation[37]. We noticed that upon Trio activation, the distribution of the Integrins β1 and α5 shifted from a well-known focal adhesion pattern towards a more junctional pattern. Also Paxillin, a well-recognized marker for focal adhesions shifted to the junction regions. Integrins may be involved in the regulation of Trio-mediated increase in cell size by controlling ECM contacts near junctions as a means to keep the enlarged cell in its shape and preventing it from collapsing. RhoG may contribute herein as it has been shown to modulate focal adhesion turnover[46] and microtubule dynamics, at least in cancer cells and epidermal keratinocytes[46–48]. Vegf has also been shown to activate RhoG independently of Trio[49]. Interestingly, here RhoG subsequently activated Rac1 through the atypical GEF DOCK4, whereas SGEF, instead of Trio, was the responsible upstream activator of RhoG. Thus, downstream from VEGF, SGEF, and Trio may work in parallel, to activate RhoG and Rac1, respectively. Both pathways may also function as a reciprocal backup, as it was reported that SGEF knock out animals do not show any vascular malformations[50].

Increasing Vegf—Trio signalling strength in developing arteries and augmenting both arterial endothelial cell size and their number allows for a 2–2.5-fold structural increase of arterial lumen diameter in vivo (Fig. 10a, b). Using Vegf to obtain arterial diameter growth in aISVs in vivo requires precise fine-tuning of local Vegf signaling strength in arterial endothelium. This can be achieved by targeting the arterial Flt1-Vegf binding equilibrium with the Flt1 specific ligands Plgf and Vegfb. We demonstrate that developing arteries express Flt1 protein, and analyses of *flt1* mutants and *flt1* morphants shows that arterial Flt1 traps sufficient endogenous Vegf to modulate arterial diameter growth. Overexpression of Plgf or Vegfb, and displacing the arterial Flt1-trapped Vegf, results in an artery specific Vegf gain of function scenario. This triggers an increase in arterial endothelial cell size and number, culminating in a two-fold increase in arterial diameter. Plgf-induced arterial diameter growth requires Vegfaa-Kdrl signaling, but is independent of mFlt1 signaling, as the

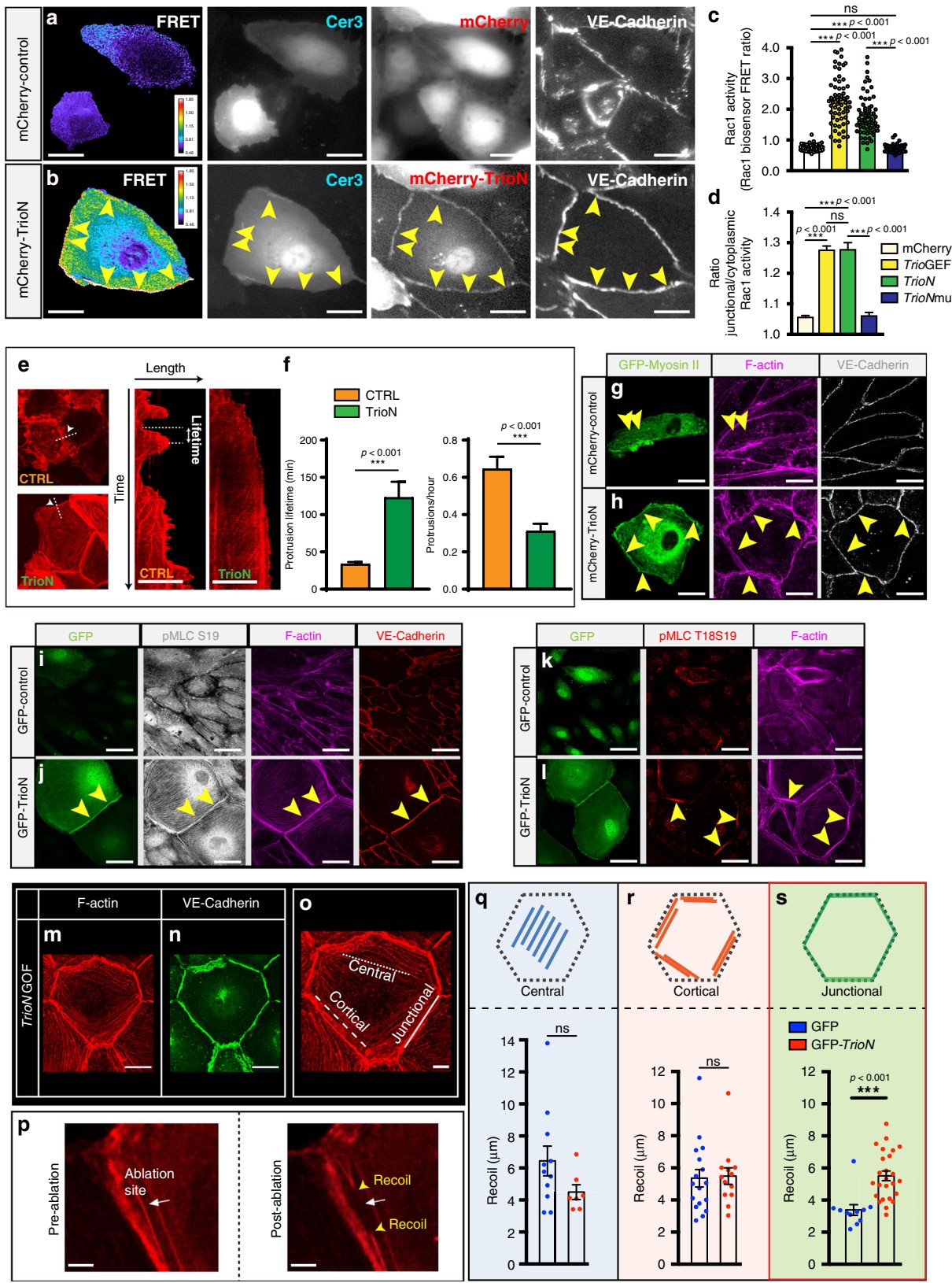

growth effect of Plgf is maintained in *mflt1* mutants. Arterial endothelial cells of aISVs enlarge considerably in *plgf^musc* transgenics. Consistent with Trio driving endothelial enlargement in this Vegf gain of function scenario, we find that inhibiting Trio reduces endothelial cell size, and aISV diameter in *plgf^musc*. Our model suggests that arterial diameter growth in *plgf^musc* involves a

combination of both increased cell number and enlarged size, enabling the shaping of a structurally larger lumen (Fig. 10a, b).

Trio gain- and loss-of-function experiments in WT and *plgf^musc* embryos show that Trio augments arterial lumen dimensions through modulation of endothelial cell shape. The in vitro data suggest that Trio regulates cell shape in a cell-

**Fig. 6 Trio activates Rac1 and induces tension in endothelial junctional regions. a** EC transfected with *mCherry* and Rac1 biosensor (Cer3, cerulean3 channel) and stained for VE-Cadherin. FRET signals depicted as warm colors. **b** EC transfected with *mCherry-TrioN* and Rac1 biosensor (Cer3) and stained for VE-Cadherin. FRET signals depicted as warm colors. Arrowheads indicate FRET signal in junctional regions. **c, d** Rac1 activation for indicated scenario. Mean ± s.e.m, two-sided Mann–Whitney U test, $n = 69, 59, 65, 69$ cells per indicated group (**c**). Ratio junctional to cytoplasmic Rac1 activation (**d**): mean ± s.e.m, two-sided Mann–Whitney-test, $n = 69, 59, 65, 69$ cells per group, respectively. ***$p < 0.001$. **e** Kymograph illustrating the actin cytoskeleton dynamics along indicated line in control and *TrioN*-transfected cells. **f** Quantification of protrusion lifetime (left panel). Mean ± s.e.m, unpaired two-sided students *t*-test, $n = 50, 110$ cells. (right panel) Quantification of protrusion dynamics expressed as protrusion number per hour from data in left panel. ***$p < 0.001$. **g** EC transfected with *mCherry* and *GFP-Myosin II*, stained for F-actin and VE-cadherin. Myosin-II localizes at actin bundles (arrowheads). **h** EC transfected with *mCherry-TrioN* and *GFP-Myosin II*, stained for F-actin and VE-cadherin. Myosin-II localizes at junction actin bundles (arrowheads). **i** EC transfected with *GFP* stained for pMLC-S19, F-actin, and VE-cadherin. **j** EC transfected with *GFP-TrioN*, stained for pMLC-S19, F-actin and VE-cadherin. Arrowheads show pMLC-19 at junctional actin bundles. **k** EC transfected with *GFP*, stained for pMLC-T18S19 and F-actin. **l** EC transfected with *GFP-TrioN*, stained for pMLC-T18S19 and F-actin. Arrowheads show pMLC-T18S19 at junctional regions. **m, n** distribution of F-actin (**m**) and VE-Cadherin (**n**) in *TrioN* overexpressing EC. **o** Intracellular distribution of central, cortical and junctional F-actin bundles in ECs. **p** F-actin recoil response upon laser ablation. Shown are pre-ablation (left panel) and post-ablation states (right panel); yellow arrowheads indicate the extent of recoil. **q–s** Recoil distance in central (**q**), cortical (**r**) and junctional (**s**) F-actin bundles, in *GFP* and *GFP-TrioN*-transfected ECs; mean ± s.e.m, unpaired two-sided students *t*-test, $n = 11, 7, 17, 13, 11$ and 25 cells per indicated group. ns not significant ***$p < 0.001$. Scale bar, 25 μm in **a**, **b**, **m**, **n**; 20 μm in **e**, **g–l**, 10 μm in **o**, 5 μm in **p**.

autonomous manner. Trio furthermore augments junctional integrity and stabilizes endothelial–endothelial cell–cell junctions. Such increased junctional stability would be beneficial to keep expanding vessels sealed and prevent plasma leakage during the outward remodeling process[45,51]. The TGFβ co-receptor Endoglin controls arterial diameter in response to shear stress through modulation of endothelial cell shape[6,10]. With respect to the regulation of 2dpf aISV and aorta diameter, we find no interaction between Vegf-Trio signaling and Endoglin. However, observations in 3dpf aorta suggest that Trio may be required for mediating aorta diameter increase upon loss of *endoglin*. How loss of *endoglin* couples to Trio remains to be determined but human endothelial cells devoid of Endoglin show altered VEGFR-2 kinetics, and exhibit differential activation of VEGFR-2 downstream pathways, including AKT[6]. We hypothesize that Vegf can activate Trio during early stages of arterial network remodeling, whereas the TGF beta pathway may act to restrict Trio function once flow becomes more dominant during the maturation of the arterial network. Although we originally investigated activation of Vegf-Trio as a means to augment arterial diameter in the context of improving perfusion for ischemic cardiovascular disease, the notion that inhibiting Trio and reducing EC and arterial size, may have therapeutic relevance as well for preventing shunt formation and vascular complications in, for example hereditary hemorrhagic telangiectasia (HHT)[10,11].

In early zebrafish embryos, mural cells are absent and vessel size is determined solely by the endothelium. In adult resistance sized arteries the presence of several layers of smooth muscle cells, and sympathetic nerve induced vascular tone may restrict the impact of Trio on diameter remodeling. It therefore seems unlikely that activating Trio and promoting EC size, can directly affect arterial diameter independent of vascular smooth muscle function in mature resistance sized arteries. Flt1 and its ligands PlGF and VEGF-B can regulate vascular adaptation in ischemic cardiovascular conditions in mouse, rat and pig models[52–56]. In rats, constitutive cardiomyocyte specific expression of full length *Vegf-b* results in enlargement of all coronary arteries[53]. In the coronaries, Flt1 and Kdr only colocalize in the most distal areas and capillaries, whereas coronary arteries in the proximal part express only Flt1. Based on the juxtapositioning of Flt1 and Kdr, it is therefore conceivable that arterial enlargement commences in precapillary arteries and that this is propagated retrogradely to the proximal areas[53]. We previously showed that such retrograde communication mechanism between micro- and macrocirculation, exists in an ischemic setting and promotes arterial remodeling[57]. In a more physiological context, increases in metabolism

or tissue hypoxia can trigger the local release of Vegf. Vegf-Trio-induced endothelial enlargement and diameter increase reduce the resistance to flow. At a given pressure gradient this results in a redistribution of flow toward the enlarged vessel segment in the Vegf-producing or hypoxic region. We hypothesize that such vessel adaptations allow matching of changes in metabolism with flow delivery. We propose that this mechanism may operate in particular in vessel segments that lack the ability to actively regulate vascular tone by smooth muscle cells, which is typically the case in the most distal parts of the microcirculation, the areas where Flt1 and Kdr are co-expressed.

## Methods

**Ethics statement.** Zebrafish husbandry and experimental procedures were performed in accordance with local and national German animal welfare standards[58] and were approved by the government of Baden-Württemberg, Regierungspräsidium Karlsruhe, Germany (Akz.: 35-9185.81/G-93/15; Akz.: 35-9185.81/G-93/19), the government of Berlin, Regierungspräsidium Berlin, Germany (Akz.: Reg0318/13) and the government of Nordrhein-Westfalen, Germany (Akz.: 81-02.05.40.19.044).

**Zebrafish maintenance and strains.** Zebrafish were maintained at standard conditions[58]. Embryos were incubated at 28.5 °C and staged by hours or days post fertilization (hpf or dpf, respectively). The following zebrafish lines were used: Tg (fli1a:EGFP)$^{y1}$, TgBAC(flt1:YFP)$^{hu4624}$, Tg(flt1$^{enh}$:tdTomato)$^{hu5333}$, TgBAC(flt4: mCitrine)$^{hu7135}$, Tg(kdrl:hsa.HRAS-mcherry)$^{s916}$, Tg(fli1a:lifeactEGFP)$^{mu240}$, Tg (fli1a:nEGFP)$^{y7}$, Tg(fli1ep:gal4ff)$^{ubs4}$, Tg(UAS:VE-cadherinΔC-EGFP)$^{ubs12}$, Tg (mpeg1:GAL4-VP16)$^{gl24}$, Tg(UAS:E1b:Kaede)$^{s1999t}$, flt1$^{ka601}$, flt1$^{ka605}$, vhl$^{hu2114}$, cdh5$^{ubs8}$ and nrp1a$^{hu10012}$ as published[29,59–68].

**Generation of mutant and transgenic lines.** Zebrafish embryos were injected at the one-cell stage with a glass microneedle and a microinjector (World Precision Instruments). For transgenesis *tol2* mRNA, transcribed from pCS2FA with the mMESSAGE mMACHINE SP6 Transcription Kit, a kind gift from Koichi Kawakami, was injected at a concentration of 25 ng μl$^{-1}$ together with Tol2 destination vectors[69]. For mutagenesis *Cas9* mRNA was in vitro transcribed from the MLM3613 plasmid (Addgene plasmid #42251) using the mMESSAGE mMACHINE T7 ULTRA Transcription Kit (Thermo Fisher Scientific); sgRNA target sequences were cloned into DR274 (Addgene plasmid # 42250) and transcribed with the MAXIscript T7 transcription kit (Thermo Fisher Scientific). Subsequently, 1 nl of a mixture of 600 ng ml$^{-1}$ *Cas9* mRNA and 50 ng ml$^{-1}$ sgRNA were co-injected into one-cell stage embryos[29].

**Generation of *Vegfaa* gain of function lines.** In order to generate a constitutive overexpression construct for *vegfaa165* in skeletal muscle tissue, a Gateway reaction using p5E_503unc, pME_GFP-p2a-vegfaa165, p3E_polyA, and pDestTol2CG2 was performed (Supplementary Fig. 1a). For constitutive overexpression 25 ng μl$^{-1}$ of *503unc:eGFP-P2A_vegfaa* were co-injected with *tol2* transposase mRNA. Embryos were identified by means of the transgenesis marker *cmlc2:EGFP* as well as green fluorescence in the skeletal muscle tissue.

For inducible *vegfaa165* gain of function, an inducible, muscle-specific expression construct was generated in a Gateway reaction with p5E_503unc, pME_gal4ERT2, p3E_polyA, and pDestTol2CG2; the other expression construct

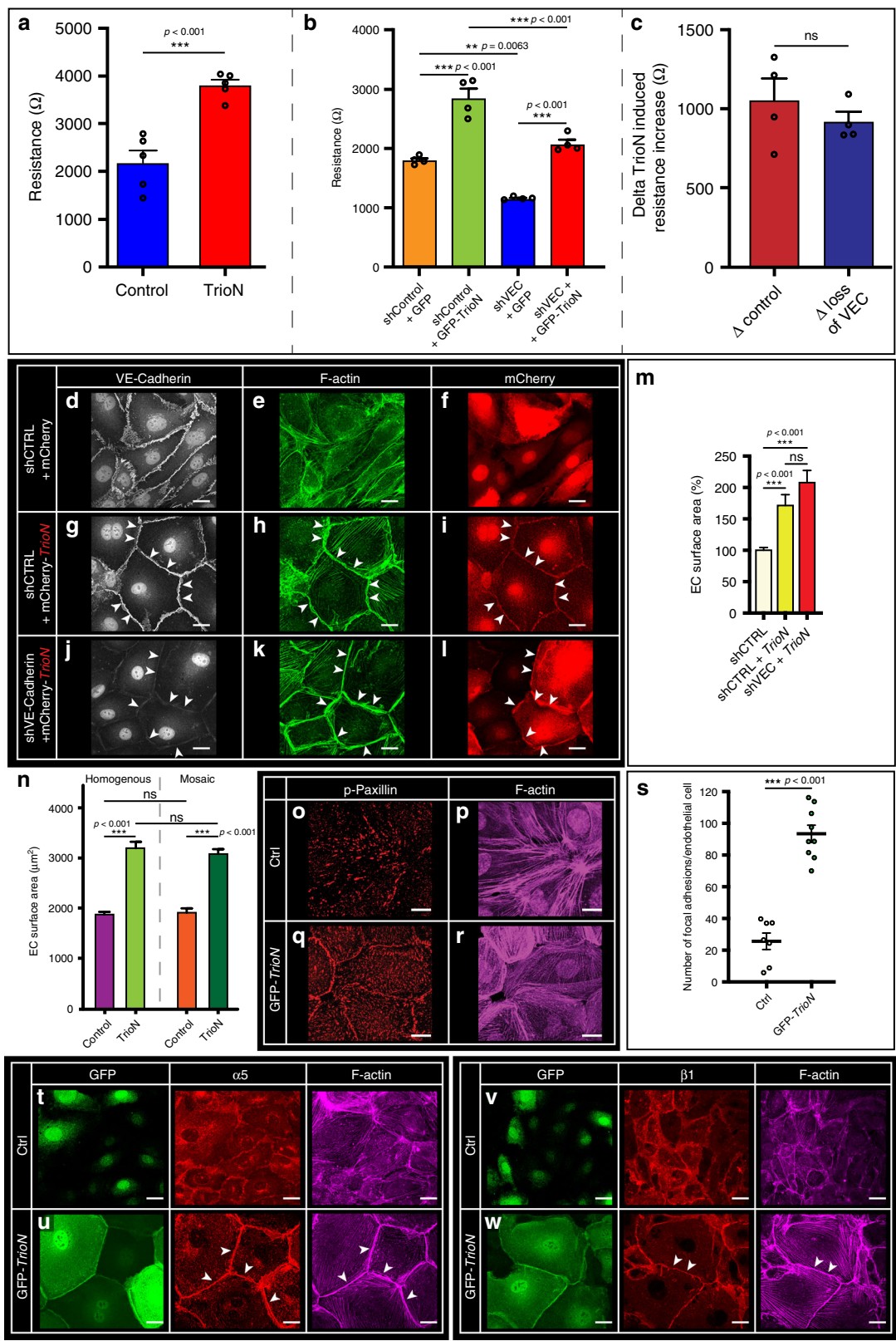

was produced using p5E_UAS, pME_GFP-p2A-vegfaa165, p3E_polyA and pDestTol2CG2 (Supplementary Fig. 1b). 12.5 ng µl⁻¹ of each of these two constructs were co-injected with *tol2* transposase mRNA. *Vegfaa* expression was induced at 28 hpf by addition of 0.5 µM endoxifen in E3 medium (5 mM NaCl, 0.17 mM KCl, 0.33 mM CaCl₂, 0.33 mM MgSO₄) and incubation of the embryos in the dark.

**Generation of an early stop *flt4* mutant allele**. For the generation of the *flt4*[mu407] allele via CRISPR-Cas9 genome editing, sgRNA synthesis was performed according to the standard protocol[70]. The target sequence 5′-GGCTGTTATTGATGGC ACCA-3′ resides in exon 3 and expanded a BsaJI restriction site, which was used for screening and efficacy testing after PCR amplification using the primer pair 5′-GAAGGAGTGTCAAGGGGTGG-3′ and 5′-CGGTTGCACATTCCCCAAAG-3′. A mix containing 300 pg of zebrafish codon-optimized Cas9 mRNA and 20 pg of

**Fig. 7 Trio acts cell-autonomous to increase endothelial cell size. a** Electrical Cell-Substrate impedance (ECIS) array in control and *TrioN* overexpressing EC. Experiment in triplicate, repeated three independent times. Mean ± s.e.m, unpaired two-sided students *t*-test, ***$p$ < 0.001. **b** ECIS array in control plasmid or *TrioN*-transfected EC, and co-transfected with control shRNA or VE-Cadherin silencing shRNA. Experiment in duplicate, repeated three independent times. Mean ± s.e.m, unpaired two-sided students *t*-test, **$p$ = 0.0063, ***$p$ < 0.001. **c** Relative impedance change for indicated scenario based on data in **b**. Experiment carried out in duplicate, three independent times. Mean ± s.e.m, unpaired two-sided students *t*-test. ns not significant. **d**–**f** VE-Cadherin (**d**), F-actin (**e**), and mCherry (**f**) distribution in EC transfected with *mCherry* plasmid and control *shRNA* (shCTRL). **g**–**i** VE-Cadherin (**g**), F-actin (**h**), and mCherry-TrioN (**i**) distribution in ECs transfected with *shCTRL*, and co-transfected with *TrioN*. Arrowheads show colocalization of VE-Cadherin, F-actin and TrioN at cell–cell junctions. **j**–**l** VE-Cadherin (**j**), F-actin (**k**), and mCherry-TrioN (**l**) distribution in ECs transfected with *shVE-Cadherin*, and co-transfected with *TrioN*. Arrowheads show intact F-actin bundles and TrioN expression at junctions. **m** EC size for indicated scenario. mean ± s.e.m, two-sided Mann–Whitney U test, $n$ = 69, 26, and 22 cells per group derived from four independent experiments. ***$p$ < 0.001. **n** EC size of *TrioN* expressing cell, when in contact with *TrioN* expressing neighbouring cell (homogenous expression) or when in contact with control-transfected neighbouring cells (mosaic expression). Cell-sizes from 52 cells per condition, and two independent experiments. Mean ± s.e.m, unpaired two-sided students *t*-test. ns not significant, ***$p$ < 0.001. **o**, **p** Distribution of focal adhesions, labeled with phospho-Paxillin (**o**); and F-actin (**p**) in control-transfected ECs. **q**, **r** Distribution of focal adhesions, labeled with phospho-Paxillin (**q**); and F-actin (**r**) in *TrioN*-transfected ECs. **s** Focal adhesion density in control and *TrioN*-transfected ECs. Mean ± s.e.m, unpaired two-sided students *t*-test, $n$ = 7, 9 cells per indicated group. ***$p$ < 0.001. **t** Integrin-alpha-5 and F-actin expression in GFP control-transfected EC. **u** Integrin-alpha-5 and F-actin expression in GFP-*TrioN*-transfected EC. Arrowheads indicate junctional region. **v** Integrin-beta-1 and F-actin expression in GFP control-transfected EC. **w** Integrin-beta-1 and F-actin expression in GFP-*TrioN*-transfected EC. Arrowheads indicate junctional region. Scale bar, 25 μm in all panels.

*flt4* sgRNA was injected into the cytoplasm of one-cell stage embryos. An 8 bp deletion was identified by sequencing using previously indicated primers. The deletion introduces a predicted premature stop codon within the first immunoglobulin domain (Ig) of the protein. Analysis of the thoracic duct (TD) at 5 dpf revealed this allele to be phenotypically identical to the previously published *expando* allele[71].

**Generation of the *TgTm(flt1_E3_HAHA)^ka611* knockin line**. The knockin line *TgTm(flt1_E3_HAHA)^ka611* was generated by injecting 600 ng μl⁻¹ Cas9 mRNA and 50 ng μl⁻¹ *sgRNA^flt1Exon3* to induce a double strand break (DSB), together with 12.5 pg μl⁻¹ oligonucleotide sFlt1_HAHA_ODN_1 (oligonucleotide sequences are listed in Supplementary Table 2) for homologous recombination. The oligonucleotide encodes two HA tags which are flanked by two 33 nt homology arms for the *flt1* genomic region surrounding the DSB (Supplementary Fig. 2f). Founders were identified by the presence of an additional 60 bp larger PCR product, amplified with flt1_E3_gDNA_fw and flt1_E3_gDNA_rev, and in-frame integration of the double HA tag was verified by Sanger sequencing of the PCR product. The knockin line was viable, showed no vascular defects compatible with HA-tagged Flt1 being functional.

**Generation of the *Tg(flt1^enh:sflt1_Δ7-HAHA)^ka612* line**. For overexpression of sFlt1 tagged with two HA tags, *sflt1* was amplified with Phusion High-Fidelity DNA polymerase (NEB) from cDNA with the primers Flt1_XhoI_fw and sFlt1_HAHA_Xba and cloned into the MiniTol2 vector containing *flt1* promoter/enhancer elements (*flt1^enh*)[63] using the restriction enzymes XhoI and XbaI. The resulting *sflt1* overexpression construct *flt1^enh:sflt1_wt-HAHA* (Supplementary Fig. 2g, upper panel) was injected at a concentration of 12.5 ng μl⁻¹ together with 25 ng μl⁻¹ *tol2* mRNA. Due to abrogation of ISV growth with this *sflt1* over-expression construct (Supplementary Fig. 2i–k), the Vegfaa binding site of sFlt1 was deleted (Supplementary Fig. 2h). Therefore, interface residues of the human sFLT1 receptor domain 2 binding to human VEGF-A were identified with PyMOL (www.pymol.org). The corresponding positions in the zebrafish sFlt1 homologue were identified as E134 to Y140 (Supplementary Fig. 2h), encoded by the base pairs 403–423. These were deleted from *flt1^enh:sflt1_wt-HAHA* by site-directed mutagenesis (SDM) with the primers SDM_Flt1delta7_fw and SDM_Flt1delta7_rev. For SDM the PCR product was phosphorylated with T4 polynucleotide kinase, followed by blunt end ligation with T4 DNA ligase (both NEB). Remaining template plasmid was digested with DpnI. 25 ng μl⁻¹ of the resulting *flt1^enh:sflt1_Δ7-HAHA* Tol2 expression construct (Supplementary Fig. 2g, lower panel) were injected with *tol2* transposase mRNA, fish were raised and founders of the *Tg(flt^enh:sflt1_Δ7-HAHA)^ka612* line were identified by PCR using the primers sflt1_fw and HA_rev.

**Generation of *plgf^musc* and *vegfba^musc* gain of function lines**. For overexpression of zebrafish *plgf* (ENSDART00000156625.2) and *vegfba* (ENSDART00000123364.3), their coding sequences were cloned into Gateway middle entry clones. Therefore, *plgf* and *vegfba* coding sequences were amplified with Phusion High-Fidelity DNA polymerase (NEB) from zebrafish cDNA with SmaI_Plgf_CDS_fw/ XhoI_Plgf_CDS_rev, and SmaI_Vegfba_CDS_fw/XhoI_Vegfba_CDS_rev, respectively (Supplementary Table 2). The vector pME_eGFP_p2A_SmaI and the PCR products were digested with SmaI and XhoI, and after gel purification ligation reactions were performed with T4 DNA ligase (NEB). The resulting middle entry clones pME_GFP-p2A-plgf and pME_GFP-p2A-vegfba were used in Gateway reactions (Thermo Fisher Scientific, LR clonase II plus) with p5e_503unc, p3E_polyA and pDestTol2CG2. For generating the stable transgenic lines *Tg(503unc:eGFP-p2A-plgf)^ka613* (referred to as *plgf^musc*) and

*Tg(503unc:eGFP-p2A-vegfba)^ka614* (referred to as *vegfba^musc*) (Supplementary Fig. 2o, p), 25 ng μl⁻¹ overexpression constructs *503unc:eGFP-p2A-plgf* or *503unc:eGFP-p2A-vegfba* were co-injected with *tol2* transposase mRNA. Fish were raised and founders were identified by means of the transgenesis marker *cmlc2:EGFP* as well as green fluorescence in the skeletal muscle tissue.

**Trio gain of function**. TrioN contains the N-terminal part of Trio (amino acids 1-1685) including the Sec14 domain, spectrin repeats, the Rac1/RhoG GEF1 domain and SH3 domain, but excluding the RhoA-specific GEF2 domain and kinase domain[72]. For TrioN overexpression under the *flt1^enh* promoter in zebrafish embryos, the p5E_flt1enh, pENTR1a_GFP-TrioN and p3E_polyA sequences were recombined into the pDestTol2CG2 in a Gateway reaction (LR clonase II plus; Thermo Fisher Scientific). The resulting expression construct was injected into the cytoplasm of one-cell stage embryos at a concentration of 25 ng μl⁻¹ together with *tol2* mRNA. For generation of the stable transgenic line *Tg(flt1^enh:GFP-TRION)^ka615* (referred to as *flt1^enh:TrioN*), injected fish were raised and founders were identified by means of the transgenesis marker *cmlc2:EGFP*. Overexpression of TrioN under the *kdrl* and *fli1a* promoter in zebrafish embryos was performed likewise via Gateway recombination reactions using the respective p5E_kdrl and p5E_fli1a plasmids. The mutant GFP-TrioNmut construct (TrioNmut) was generated by site-directed mutagenesis of nucleotides 4399-4405 (NM007118) AAT-GAC to GCTGCC. This results in the amino acid mutations N1406A and D1407A, located in the C-terminus of the DH1 domain. The mutated GEF domain is still able to bind Rac1, but unable to induce guanine nucleotide exchange. Overexpression of TrioNmut under the *flt1^enh* promoter was performed similarly to TrioN. For Trio gain of function in vitro, primary HUVECs, which were acquired from Lonza (Verviers, Belgium) and regularly checked for mycoplasma contamination, were seeded on culture flasks. HUVECs were grown in EGM-2 medium (Promocell, Heidelberg, Germany). Cells were cultured on fibronectin-coated glass coverslips and transfected with human GFP-TrioN, mCherry-TrioN, mCherry-TrioNmut, mCherry-TrioGEF1 or the isoform TrioB. The mCherry-TrioN/TrioNmut/TrioGEF1 constructs were generated by replacing GFP in GFP-TrioN/TrioNmut/TrioGEF1 with mCherry using AgeI and Kpn2I restriction enzymes. After treatment, cells were washed with warm PBS, containing 1 mM CaCl₂ and 0.5 mM MgCl₂, and fixed in 4% (v/v) formaldehyde for 10 min. After fixation, cells were permeabilized in PBS supplemented with 0.1% (v/v) Triton X-100 for 10 min followed by a blocking step in PBS supplemented with 2% (w/v) BSA and stained as indicated[45].

**Rac1 gain of function**. In vitro Rac1 gain of function experiments were performed similarly to Trio gain of function approaches. pTriEx-mCherry-PA-Rac1 and pTriEx-mCherry-PA-Rac1-C450A were a kind gift from Klaus Hahn (Addgene plasmid # 22027: http://n2t.net/addgene:22027, RRID:Addgene_22027 and Addgene plasmid # 22028: http://n2t.net/addgene:22028, RRID:Addgene_22028). Cloning of mCherry-Rac1-Q61L and Rac1B was performed as previously described[39,73].

**RhoG gain of function**. In vitro RhoG gain of function experiments were performed similarly to Trio & Rac1 gain of function approaches. GFP-RhoG Q61L was a kind gift from Dr. Keith Burridge (UNC Chapel Hill, USA).

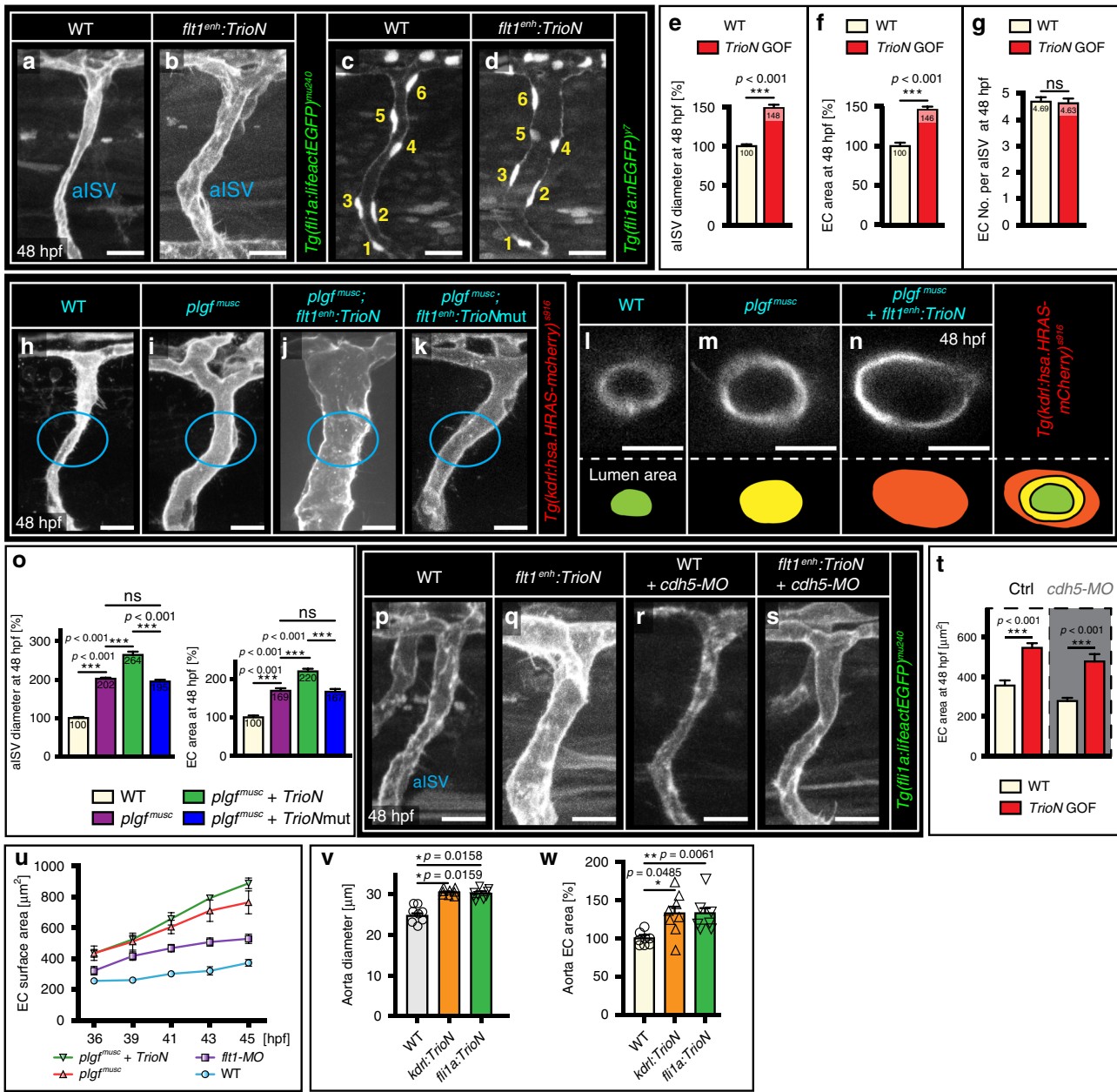

**Fig. 8 Trio augments endothelial cell size and arterial diameter in vivo. a, b** Imaging of aISV in WT (**a**) and in *flt1^enh^:TrioN* embryos (**b**). **c, d** EC nuclei distribution in WT (**c**) and *flt1^enh^:TrioN* embryos (**d**). Note: six ECs in both scenarios and larger aISV diameter upon *TrioN*. **e** Quantification of aISV diameter for indicated scenario. Mean ± s.e.m, two-sided Mann–Whitney U test, $n = 35$ aISVs per genotype. ***$p < 0.001$. **f** Quantification of EC surface area in aISVs for indicated scenario. Mean ± s.e.m, two-sided Mann–Whitney U test, $n = 34$, 48 aISVs per indicated genotype. ***$p < 0.001$. **g** Quantification of EC nuclei number in aISVs for indicated scenario. mean ± s.e.m, two-sided Mann–Whitney U test, $n = 51$, 70 aISVs per genotype. ns not significant. **h–k** aISV in WT (**h**), in *plgf^musc^* (**i**), *plgf^musc^* combined with *flt1^enh^:TrioN* (**j**), and *plgf^musc^* combined with *flt1^enh^:TrioN*mut (**k**). **l–n** Transverse optical section of aISV vessel lumen in WT (**l**), *plgf^musc^* (**m**), and *plgf^musc^+flt1^enh^:TrioN* scenario (**n**). The lumen area is color indicated (lower panels). **o** Left panel: Quantification of aISV diameter in WT, *plgf^musc^*, *plgf^musc^+flt1^enh^:TrioN*, and *plgf^musc^+flt1^enh^:TrioN*mut scenario. Mean ± s.e.m, unpaired two-sided students *t*-test, $n = 18$, 19, 13, 14 aISVs per group. ns, not significant; ***$p < 0.001$. Right panel: Quantification of EC size in WT, *plgf^musc^*, *plgf^musc^+flt1^enh^:TrioN*, and *plgf^musc^+flt1^enh^:TrioN*mut scenario. Mean±s.e.m, unpaired two-sided students *t*-test, $n = 21$, 24, 15, 12 cells per group. ns not significant; ***$p < 0.001$. **p–s** aISV in WT embryo (**p**), *flt1^enh^:TrioN* embryo (**q**), WT embryo injected with *ve-cadherin* (*cdh5*) targeting morpholino (**r**), *flt1^enh^:TrioN* embryo injected with *ve-cadherin* (*cdh5*) targeting morpholino (**s**). **t** Quantification of images in **p–s**; mean ± s.e.m, unpaired two-sided students *t*-test, $n = 15$, 21, 16, and 13 cells per indicated genotype. ***$p < 0.001$. **u** Dynamic changes in EC surface area in *plgf^musc^ + flt1^enh^:TrioN* embryo, *plgf^musc^* embryo, WT embryo injected with *flt1* targeting morpholino, and WT embryo. $n = 12$, 12, 10, 18 cells per indicated genotype, mean ± s.e.m. **v, w** Aorta diameter (**v**) and EC surface area (**w**) in *kdrl:TrioN* or *fli1a:TrioN* scenario. Mean ± s.e.m, unpaired two-sided students *t*-test, $n = 8$, 10, 10 (**v**) and $n = 8$, 16, 14 (**w**) biologically independent embryos/group. *$p < 0.05$, **$p = 0.0061$. Scale bars: **a–d**, **h–k**, **p–s**, 25 μm, **l–n** 10 μm.

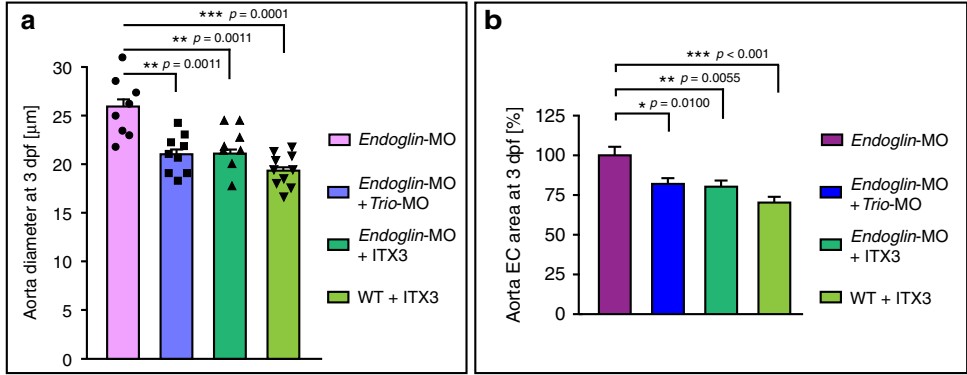

**Fig. 9 Trio and endoglin interact to determine 3dpf aortic EC size. a** Dorsal aorta diameter at 3dpf in *endoglin* morphants (magenta bar), *endoglin* morphants injected with 0.8 ng *Trio* targeting morpholino (blue bar), *endoglin* morphants treated with Trio inhibitor ITX3 (dark green bar) and WT treated with ITX3 (light green bar). Note that loss of Trio or inhibiting Trio reduced diameter growth in *endoglin* morphants. Mean ± s.e.m, unpaired two-sided students *t*-test, $n = 8, 9, 8, 10$ independent embryos per group. **$p = 0.0011$, ***$p = 0.001$. **b** Aorta EC surface area at 3dpf in *endoglin* morphants (magenta bar), *endoglin* morphants injected with 0.8 ng *Trio* targeting morpholino (blue bar), *endoglin* morphants treated with Trio inhibitor ITX3 (dark green bar) and WT treated with ITX3 (light green bar). Note that loss of Trio or inhibiting Trio reduced EC surface area in *endoglin* morphants. Mean ± s.e.m, unpaired two-sided students *t*-test, $n = 18, 20, 22, 16$ cells derived from six independent embryos per group; ***$p < 0.001$; **$p = 0.0055$; *$p = 0.0100$.

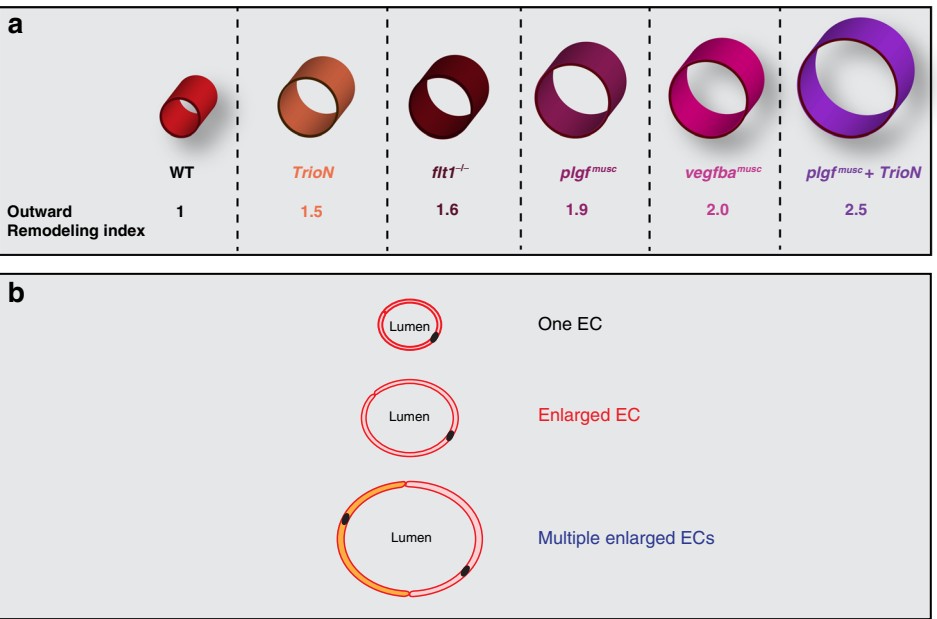

**Fig. 10 Schematic representation of arterial lumen increase. a** Increase in arterial lumen dimensions normalized to WT (WT = 1.0). Artery specific *TrioN* gain of function fish show 1.5 fold, *flt1* mutants show 1.6 fold, *plgf$^{musc}$* 1.9 fold, *vegfba$^{musc}$* 2.0 fold, and *plgf$^{musc}$*+artery specific *TrioN* 2.5-fold larger arterial lumen diameter dimensions compared to WT. **b** Promoting endothelial cell enlargement results in larger arterial diameter. Simultaneously promoting endothelial cell number and size, and creating arteries with multiple enlarged ECs results in even more pronounced arterial diameter increase.

**Tiam gain of function**. In vitro Tiam1 gain of function experiments were performed similarly to RhoG gain of function approaches. HA-TIAM C1199 was a kind gift from Dr. John Collard (NKI Amsterdam, Netherlands).

**Further plasmids**. Myosin IIA-GFP was a kind gift from Dr. Ana Pasapera (NHLBI Bethesda, USA).

**Morpholino knockdown**. The following morpholino antisense oligomers (MOs; Gene Tools) were injected into the yolk of one-cell stage embryos: 1 ng *flt1* ATG MO (5′-ATATCGAACATTCTCTTGGTCTTGC-3′); 4 ng *flt4* ATG MO (5′-CT CTTCATTTCCAGGTTTCAAGTCC-3′); 0.3 ng *vegfaa* ATG MO (5′-GTATCAA ATAAACAACCAAGTTCAT-3′); 2.7 ng *tnnt2* ATG MO (5′-CATGTTTGCTCT GATCTGACACGCA-3′); a combination of 8.5 ng *pu.1* ATG MO (5′-GATATA CTGATACTCCATTGGTGGT-3′), 8.5 ng *gcsfr* ATG MO (5′-GAAGCACAAGCG AGACGGATGCCAT-3′) and 10 ng *irf8* splice-blocking MO (5′-AATGTTTCG CTTACTTTGAAAATGG-3′) to inhibit macrophage formation; 0.8 ng, 1.7 ng, 3.3 ng, 5 ng and 8.4 ng *trio* ATG MO (5′-AGCTCATGGCTGACGAAAAACA CA-3′); 1.7 ng *cdh5* ATG MO (5′-TTTACAAGACCGTCTACCTTTCCAA-3′);

5 ng *endoglin* (*eng*) ATG MO-1 (5′-AAACACAGCAGATGCTCTTCATGTC-3′); 5 ng *eng* ATG MO-2 (5′-GATGAACTCAACACTCGTGTCTGAT-3′); 6 ng, 3 ng *vhl* E1 splice-blocking MO (5′-GCATAATTTCACGAACCCACAAAAG-3′); 10 ng standard control MO (5′-CCTCTTACCTCAGTTACAATTTATA-3′), all as described[29,74–78].

**RNA interference**. Inhibitory shRNA constructs (Supplementary Table 3) as well as a non-targeting shCtrl (shC002) were packaged into lentivirus in HEK293T cells by means of third generation lentiviral packaging plasmids. Lentivirus-containing supernatant was harvested on day 2 and 3 after transfection. Lentivirus was concentrated via Lenti-X concentrator (Clontech). Target cells were infected and cells were used for assays 3 days after virus infection.

RNA interference using siRNA constructs (Supplementary Table 4) was performed as published[37].

**Chemical and inhibitor treatments**. Embryos were dechorionated prior to chemical and inhibitor treatments using 1 mg ml$^{-1}$ Pronase (Roche, Basel, Switzerland). The chemicals/inhibitors were added at 32 hpf, followed by incubation at

28.5 °C in the dark until the embryos were analysed at 50 hpf. For chemicals/inhibitors dissolved in DMSO, control embryos were mock treated with DMSO (Sigma). Inhibitor and chemical concentrations are listed in Supplementary Table 5 and 6. Embryos were randomly assigned to experimental groups. Investigators were blinded to inhibitor treatment. In vitro chemical treatments were performed at indicated time points. Inhibitor concentrations are further specified in Supplementary Table 5.

**Immunohistochemistry of zebrafish embryos.** Fixed embryos were permeabilized for 1 h in PBT (1% (v/v) Triton X-100 in 1x PBS). In order to reduce unspecific background, samples were subsequently incubated for 2 h in blocking buffer (1% (v/v) Triton X-100, 3% (w/v) BSA in 1× PBS). The primary antibody (Supplementary Table 7) was diluted in 0.5% (v/v) Triton X-100, 1% (w/v) BSA in 1× PBS, and samples were incubated overnight at 4 °C. Embryos were washed five times for 10 min with PBT before the secondary antibody (Supplementary Table 7) was added for 2 h at room temperature. Before imaging, samples were washed five times for 20 min in PBT.

**Cell lines and immunofluorescent stainings in vitro.** Antibodies used for immunofluorescent staining of human endothelial cells in culture are indicated in Supplementary Table 7. The following cell lines were used: Primary Human Umbilical Vein Endothelial Cells (HUVECs, C2519A, Lonza); Primary Human Umbilical Artery Endothelial Cells (HUAECs, isolated as described previously[79]) and Primary Human Aortic Endothelial Cells (HAECs, CC-2535, Lonza). Cells were routinely checked for mycoplasma contamination and authenticated by immunofluorescence, flow cytometry and western blot for expression of standard endothelial cell markers (VE-Cadherin, CD31, ICAM-1, VCAM-1, etc). Endothelial cells were cultured in EGM-2 medium (Promocell, Heidelberg, Germany) up to passage 5.

**Whole mount in situ hybridization.** Whole mount in situ hybridization was performed with DIG labelled RNA probes. For generating the antisense ISH probe of *l-plastin*, the plasmid, kindly provided by Philippe Herbomel, was linearized with SpeI and transcribed with T7 RNA polymerase (Promega) and DIG labelling mix (Roche). *sflt1* and *mflt1* antisense ISH probes were both linearized with SpeI and transcribed with T7 RNA polymerase and DIG labelling mix. Embryos for whole mount in situ hybridization were fixed in 4% (w/v) paraformaldehyde in 1× PBS for 2 h at room temperature or overnight at 4 °C. After dehydration in 100% methanol overnight at −20 °C, the embryos were rehydrated in a decreasing methanol series, washed twice in PBT (1× PBS, 0.1% (v/v) Tween 20) and permeabilized using proteinase K at a concentration of 10 μg ml$^{-1}$ in PBT. In order to minimise unspecific background, embryos were prehybridized in hybridization mix (50% (v/v) formamide; 5× SSC; 0.1% (v/v) Tween 20; 50 μg ml$^{-1}$ heparin; 500 μg ml$^{-1}$ RNase-free tRNA) for 1 h at 65 °C. Embryos were then incubated with the RNA probe in hybridization mix overnight at 65 °C. Next, the removal of the formamid was obtained by a decreasing series of hybridization mix in 2× SSC at 65 °C, and a final washing step in 0.2× SSC at room temperature. Non-hybridized RNA was removed using RNase A (Roth) in RNase buffer (HEPES 0.1 M pH 7.5; NaCl 0.15 M; 0.1% (v/v) Tween 20) for 45 min at room temperature, followed by a washing step in 1× MAB-T (maleic acid buffer; 0.1% (v/v) Tween 20). Subsequently, nonspecific antibody binding sites were blocked by incubation in 2% (w/v) Blocking Reagent (Roche) in 1× MAB-T, followed by incubation with alkaline phosphatase-conjugated antibody (Anti-Dig, Fab fragments, 1:2.000, Roche) overnight at 4 °C. Afterwards, embryos were washed once with 1× MAB-T and two times for 15 min with NTMT (Tris-HCl 0.1 M pH 9.5; NaCl 0.1 M; 1% (v/v) Tween 20; MgCl$_2$ 50 mM). The in situ hybridization was developed with BM Purple (Roche), a chromogenic substrate for alkaline phosphatase. When the staining reaction was complete, embryos were washed with PBT and transferred to 80% (v/v) glycerol in 1x PBS for microscopy and long-term storage.

**Rac1 activity pull-down assay.** Classical biochemical pull-down assays were performed as described[37]. Briefly, a confluent monolayer of HUVECs was washed with ice-cold PBS$^{++}$ and subsequently lysed in lysis buffer (50 mM Tris, pH 7.4, 20 mM MgCl$_2$, 500 mM NaCl, 1% (v/v) Triton X-100) supplemented with protease inhibitors. Lysates were cleared at 14.000 Å ~ g for 5 min. GTP-bound Rac1 was isolated with biotinylated Pak1-Crib peptide coupled to streptavidin agarose. Beads were washed five times in wash buffer (50 mM Tris, pH 7.4, 10 mM MgCl$_2$, 150 mM NaCl, 1% (v/v) Triton X-100) supplemented with protease inhibitors. Pull downs and total cell lysates were immunoblotted with monoclonal Rac1, Trio and beta-actin antibodies (Supplementary Table 7).

**Western blotting.** SDS-PAGE samples were analyzed on 7.5, 10, or 15% (w/v) polyacrylamide gels, depending on the size of the proteins of interest, and transferred onto nitrocellulose membranes (Whatman, Piscataway, NJ). Subsequently of blocking in 5% (w/v) low-fat milk in Tris-buffered saline Tween 20 (TBST), the blots were incubated with respective primary antibodies (Supplementary Table 7) for 1 h at room temperature, washed in TBST and incubated with respective HRP-coupled secondary antibodies (Supplementary Table 7) for 1 h at room temperature. Blots were developed via enhanced chemiluminescence (ECL) (Thermo-

Scientific, Etten-Leur, The Netherlands). For Trio protein expression, 3–8% (w/v) Tris-acetate precast gels (ThermoFisher) were used according to the manufacturer's instructions, and samples were transferred onto nitrocellulose membranes by blotting for 18 h at 20 mA. Full Western blots in Supplementary Fig. 13.

**Laser ablation.** For laser ablation experiments HUVECs were transduced with LifeAct-mScarlet expressing lentiviral particles. The 442 nm laser was used to locally cut specific F-actin bundles. F-actin bundles were ablated for 10 seconds using full laser power and retracted beam expander to concentrate the laser beam intensity. Initial recoil of the cut F-actin bundles was measured immediately after ablation.

**Confocal microscopy.** Zebrafish larvae were embedded in 0.7% (w/v) low-melting agarose (NuSieve GTG Agarose, Lonza) in 35 mm glass bottom microscopy dishes (MatTek). The agarose was covered with E3 medium supplemented with 0.112 mg ml$^{-1}$ Tricaine and 0.003% (w/v) PTU (Sigma). Confocal t- (for time-lapse imaging) and z-stacks were acquired using a Leica SP8 confocal microscope with ×20 multi-immersion and ×40 water immersion objectives, resonance scanner, HyD detectors and LAS X software. Images are displayed as maximum intensity projections of the z-stacks. For imaging of red blood cell perfusion, six images per second were acquired at a single sagittal plane in the middle of the ISVs with the SP8 confocal microscope. The transmitted light channel was used for the red blood cells and the mCherry channel for the *Tg(kdrl:hsa.HRAS-mcherry)$^{s916}$* signal marking the endothelial cell membranes. Images were processed with ImageJ/ Fiji. aISV numbers are indicated in figure legends. For cell culture experiments, t- and z-stack image acquisition was performed on a confocal laser scanning microscope (Leica SP8) using a ×40 NA 1.3 or ×63 NA 1.4 oil immersion objective and LAS X software. Maximum intensity projections were also generated and analyzed using ImageJ/ Fiji software.

**Transmitted light microscopy.** Zebrafish larvae were anaesthetized in 0.112 mg ml$^{-1}$ Tricaine and transferred to E3 medium in a glass bottom microscopy dish. Red blood cell perfusion was visualized by imaging a single sagittal plane in the middle of two ISVs using a Zeiss Axioskop 5 microscope with ×40 objectives and ×4 optical gain for 40 ms exposure time.

**Microangiography.** Embryos were anaesthetized with 0.112 mg ml$^{-1}$ Tricaine in E3 medium and mounted in 0.7% (w/v) low-melting agarose (NuSieve GTG Agarose, Lonza) in glass bottom microscopy dishes (MatTek). For the micro-angiography Texas Red-dextran with a molecular weight of 70 kDa (Thermo Fisher Scientific, Supplementary Table 6) was solubilized in E3 medium to a concentration of 2 mg ml$^{-1}$. Using a glass microneedle and a microinjector (World Precision Instruments), the dextran was injected into the sinus venosus. Subsequently, images were taken with an SP8 confocal microscope (Leica).

**Measurements of aISV diameter, DA diameter, endothelial cell sizes, and numbers.** For vessel parameter measurements, confocal z-stacks of ISVs plus dorsal aorta were acquired of up to five aISVs per animal in the middle of the yolk sac elongation and parameters were measured with Fiji using maximum projection images of the z-stacks. For aISV diameter analysis, lateral lines were drawn at seven evenly distributed positions along the aISV perpendicularly to the vessel wall between the dorsal aorta and the DLAV, approximately every 18 μm. The average of these seven lines was calculated as the average aISV diameter. The remodeling index (RI) was calculated as follows: RI = (aISV diameter of the respective treatment group in μm)/(average aISV diameter of the WT control group). For zebrafish mutants and transgenic lines more than 50 embryos derived from more than three different breedings were analysed per genotype. In morpholino experiments morphologically malformed embryos were excluded from analysis. To determine the dorsal aorta diameter, seven to nine vertical lines were drawn at ISV sprouting positions and between two ISVs perpendicularly to the aortic vessel wall at the level of the yolk sac elongation. The average of these lines per embryo was calculated as average dorsal aorta diameter. Endothelial cell sizes in the zebrafish embryo were visualized with the *Tg(fli1a:lifeactEGFP)$^{mu240}$* line. Endothelial cell outlines were traced with the polygon selection tool in Fiji and the resulting area was measured. Endothelial cell numbers were visualized with the *Tg(fli1a:nEGFP)$^{y7}$* line. For in vitro experiments, cell size was determined either manually or by using a MatLab script (MATLAB, The MathWorks, Inc., Natick, Massachusetts, United States).

**Rac1 FRET biosensor analysis.** The DORA Rac1 FRET-based biosensor was a kind gift from Y. Wu (University of Connecticut Health Center, Farmington, CT). Development and characterization of the DORA single-chain Rac1 biosensor are described in more detail elsewhere[45]. Briefly, dimeric cerulean3 coupled to the Rac1 effector p21-activated protein kinase (PAK) is linked via ribosomal protein-based linker (L9H) to circular-permutated Venus coupled to Rac1. HUVECs were transfected with the Rac1 biosensor via electroporation using the Neon transfection system (Life Technologies, one pulse, 1300 V, 30 ms) and used 24 h post transfection. A Zeiss Observer Z1 microscope equipped with a ×40/numerical aperture 1.3 oil immersion objective, an HXP 120-V excitation light source, a Chroma 510

DCSP dichroic splitter, and two Hamamatsu ORCA-R2 digital charge-coupled device cameras was used for simultaneous monitoring of Cer3 and Venus emission. Zeiss Zen 2012 microscope software was used to control the system. Offline ratio analyses between Cer3 and Venus images were processed using the MBF ImageJ collection. Image stacks were background corrected, subsequently aligned, and a smooth filter was applied to both image stacks to improve image quality by noise reduction. An image threshold was applied exclusively to the Venus image stack, converting background pixels to "not a number (NaN)," eliminating artefacts in ratio image derived from the background noise. Finally, the Venus/Cer3 ratio was calculated with high activation shown in red/white and low activity in blue/black. For FRET quantification, data were analysed using a MatLab script (MATLAB, The MathWorks, Inc., Natick, Massachusetts, United States). Prior to any of the ratiometric FRET analyses, donor CFP and acceptor YFP channels were background corrected by subtracting the modal pixel value, and both channels were aligned to each other. Prior to watershed segmentation, cells were manually selected by adding seed points and touching cells were separated by manually drawing boundaries between them. A local threshold was applied to the watershed region, exclusively including pixels that were higher than 15% of the maximum intensity in that region. For all channels the mean fluorescence intensity was calculated for each segmented region. Ratiometric FRET analysis was applied to each segmented region in the cell. The endothelial junction marker VE-cadherin was used as positive control. Quantification of FRET signal overlapping with VE-cadherin was done between the cells edge and 10 pixels inwards. These areas were defined as junction regions.

**Resistance measurements.** Monolayer integrity was determined by measuring the electrical resistance using ECIS as published[37]. Electrode-arrays (8W10E; IBIDI, Planegg, Germany) were treated with 10 mM l-cysteine (Sigma) for 10 min and subsequently coated with 10 μg/ml fibronectin (Sigma) in 0.9% NaCl for 1 h at 37 °C. Cells were seeded at 40.000 cells per well (0.8 $cm^2$) and grown to confluence. Electrical resistance was continuously measured at 4000 Hz at 37 °C under 5% $CO_2$ using ECIS model 9600 (Applied BioPhysics, New York, MA).

**Focal adhesion quantification.** Phospho-Paxillin-positive (pY118, Supplementary Table 7) focal adhesions were quantified on thresholded images by analysing particles sized between 0.5–10 μm$^2$.

**Statistical analysis and reproducibility.** Statistical analysis was performed using GraphPad Prism 6. Each dataset was tested for normal distribution with the D'Agostino-Pearson test. Only if the data were normally distributed, a parametric method (unpaired two-sided Students $t$-test) was applied. A nonparametric test (two-sided Mann–Whitney test) was applied for non-normally distributed data sets. In case of multiple comparisons, one-way ANOVA plus Bonferroni correction or 2-way ANOVA were applied. $P$ values < 0.05 were considered significant. Data are represented as mean ± s.e.m., unless otherwise indicated. *$P$ < 0.05, **$P$ < 0.01 and ***$P$ < 0.001, #$P$ < 0.001 to indicated ctrl group. No data were excluded from statistical analysis. No statistical method was used to predetermine sample size. For every treatment, treated and control embryos were derived from the same egg lay; embryo were selected on the following pre-established criteria: normal morphology, a beating heart and circulating red blood cells. All images shown in the figures are representative examples of the respective phenotypes and expression patterns. The following section indicates how often experiments have been repeated independently with similar results. Figure 1a, b, representative micrographs from $n$ = 48, and $n$ = 32 independent embryos per indicated condition. Figure 1c, d, images are representative for $n$ = 24 embryos per condition examined over three independent experiments. Figure 1h–k, $n$ = 112, 86, 126, 93 aISVs per genotype derived from four autonomous experiments. Figure 1l, mean ± s.e.m, unpaired two-sided students $t$-test, $n$ = 18, 16, 16, 10 aISVs per genotype. ***$p$ < 0.001. Figure 1m, $n$ = 56 aISVs examined over three independent experiments. Figure 1n, mean ± s.e.m, unpaired two-sided students $t$-test, $n$ = 20, 25 aISVs for indicated scenario, ***$p$ < 0.001. Figure 1o, p, plasma extravasation in WT ($n$ = 14) and $plgf^{musc}$ ($n$ = 16 embryos) injected from three independent experiments. Figure 1q, mean ± s.e.m; unpaired two-sided students $t$-test, $n$ = 6, 8 embryos per indicated group. ns, not significantly different. Figure 2d, mean ± s.e.m, unpaired two-sided students $t$-test, $n$ = 7, 13, and 24 aISVs for indicated scenario. **$p$ = 0.0051, ***$p$ < 0.001. Figure 2e–g, representative image of indicated scenario ($n$ = 40, 19, 13, respectively). Figure 2h, mean ± s.e.m, unpaired two-sided students $t$-test, $n$ = 40, 19, and 13 aISVs for indicated scenario, from three independent experiments. ns, not significant. **$p$ < 0.01. Figure 3a–d, representative micrographs from three independent experiments. Figure 3e, mean ± s.e.m, one-way ANOVA and post-hoc bonferroni test, $n$ = 18, 20, 20, 15, 20, 20 aISVs for indicated scenarios. **$p$ = 0.0026, ***$p$ < 0.001. Figure 3f, mean ± s.e.m, one-way ANOVA and post-hoc bonferroni test, $n$ = 15, 22, 14, 28 aISVs for indicated scenarios. ***$p$ < 0.001. Figure 3k, $n$ = 21, 16, 42, 34 aISVs per indicated genotype, mean ± s.e.m, unpaired two-sided students $t$-test. ns, not significant, ***$p$ ≤ 0.001. Figure 3l, m, representative micrographs from 83 and 94 images, from three independent experiments. Figure 3n, mean ± s.e.m, unpaired two-sided students $t$-test, $n$ = 16 aISVs/group. ***$p$ < 0.001. Figure 3o–q, representative micrographs from $n$ = 48, 55, 57 images per condition, from three independent experiments. Figure 3r, mean ± s.e.

m, unpaired two-sided students $t$-test, $n$ = 11, 20, and 15 aISVs per treatment group. ***$p$ < 0.001. Figure 3v, mean ± s.e.m, unpaired two-sided students $t$-test, $n$ = 20 aISVs per group. ***$p$ < 0.001. Figure 4a, b, $n$ = 6 independent embryos per group. Figure 4c, mean ± s.e.m, unpaired two-sided students $t$-test, $n$ = 10, 11 aISVs for indicated genotype. *$p$ = 0.0190, ***$p$ < 0.001. Figure 4d, illustrations are representative for each four independent embryos. Figure 4e, mean ± s.e.m, unpaired two-sided students $t$-test, $n$ = 6 ECs for each genotype. Surface area expansion velocity is indicated as average slope (in μm$^2$ h$^{-1}$) calculated from regression analysis. Figure 4f, mean ± s.e.m, one-way ANOVA and post-hoc bonferroni test, $n$ = 25, 38, 30, 60, 59 aISVs for indicated condition. ***$p$ < 0.001. Figure 4g–i, $n$ = 67, 59, 60 aISVs examined over three independent experiments. Figure 4j–l, $n$ = 53, 48, 62 micrographs from three independent experiments. Figure 4m, mean ± s.e.m, one-way ANOVA and post-hoc bonferroni test, $n$ = 17, 21, 18, 15, 17, and 16 aISVs per group. ***$p$ < 0.001. Figure 4n–p, $n$ = 58, 58, 59 aISVs examined over three independent experiments. Figure 4q–s, $n$ = 66, 58, 59 aISV micrographs per condition from three independent experiments. Figure 4t, mean ± s.e.m, one-way ANOVA and post-hoc bonferroni test, $n$ = 18, 14, 14, 16, 18, and 18 aISVs per group. ns, not significant, **$p$ < 0.01, ***$p$ < 0.001. Figure 5a–f, images are representative for $n$ = 120 ECs per group derived from three independent experiments. Figure 5g, mean ± s.e.m, unpaired two-sided students $t$-test, $n$ = 59, 69, 63 cells per indicated group, respectively. ***$p$ < 0.001. Figure 5h–k, images are representative for $n$ = 130 ECs per indicated group derived from three biologically independent experiments. Figure 5l, mean ± s.e.m, two-sided Mann–Whitney U test, $n$ = 67, 56, 65, 74 cells per indicated group, respectively. *$p$ = 0.0477; ***$p$ < 0.001. Figure 5m, mean ± s.e.m, unpaired two-sided students $t$-test, $n$ = 89, 83 cells per indicated group respectively, derived from three independent experiments. ***$p$ < 0.001. Figure 5o, mean ± s.e.m, unpaired two-sided students $t$-test, $n$ = 73, 97 cells per indicated group, examined over three independent experiments. ***$p$ < 0.001. Figure 5p, mean ± s.e.m, unpaired two-sided students $t$-test, $n$ = 101, 146 cells per indicated group examined over three independent experiments. ***$p$ < 0.001. Figure 5q–s, images are representative for $n$ = 120 ECs per group derived from three separate experiments. Figure 5t–v, images are representative for $n$ = 120 ECs per group derived from three biologically independent experiments. Figure 5w, cell size was measured of $n$ = 78, 82, 103, and 80 cells for indicated scenario, respectively, derived from three separate experiments. Mean ± s.e.m, unpaired two-sided students $t$-test. ns, not significant, ***$p$ < 0.001. Figure 6a, images are representative for $n$ = 100 cells per group of three independent experiments. Figure 6b, images are representative for $n$ = 115 cells per group of three independent experiments. Figure 6c, d, mean ± s.e.m, two-sided Mann–Whitney U test, $n$ = 69, 59, 65, 69 cells per indicated group (c). mean ± s.e.m, two-sided Mann–Whitney U test, $n$ = 69, 59, 65, and 69 cells per group (d). ***$p$ < 0.001. Figure 6e, images are representative for $n$ = 60 cells per condition derived from three independent experiments. Figure 6f, (left panel) mean ± s.e.m, unpaired two-sided students $t$-test, $n$ = 50, 110 cells per group. (right panel) mean ± s.e.m, unpaired two-sided students $t$-test, $n$ = 50, 110 cells per group. ***$p$ < 0.001. Figure 6g, images are representative for $n$ = 90 cells derived from three separate experiments. Figure 6h, images are representative for $n$ = 90 cells derived from three independent experiments. Figure 6i, images are representative for $n$ = 120 cells derived from three individual experiments. Figure 6j, images are representative for $n$ = 120 cells derived from three separate experiments. Figure 6k, images are representative for $n$ = 120 cells derived from three independent experiments. Figure 6l, images are representative for $n$ = 120 cells derived from three separate experiments. Figure 6m, n, images show representative phenotypes of $n$ = 160 cells out of four autonomous experiments. Figure 6o, image is a representative scheme derived from $n$ = 160 cells out of four autonomous experiments. Figure 6p, images are representative for all indicated numbers of ECs analyzed in 6q–s, examined over three independent experiments. Figure 6q–s, mean ± s.e.m, unpaired two-sided students $t$-test, $n$ = 11, 7, 17, 13, 11, and 25 cells per indicated group. ns, not significant ***$p$ < 0.001. Figure 7a, experiment was carried out in triplicate, three independent times. Mean ± s.e.m, unpaired two-sided students $t$-test, ***$p$ < 0.001. Figure 7b, experiment was carried out in duplicate, three independent times from each other. Mean ± s.e.m, unpaired two-sided students $t$-test, **$p$ = 0.0063, ***$p$ < 0.001. Figure 7c, experiment was carried out in duplicate, three independent times from each other. Mean ± s.e.m, unpaired two-sided students $t$-test. ns, not significant. Figure 7d–f, images are representative for $n$ = 120 cells per condition derived from four independent experiments. Figure 7g–i, images are representative for $n$ = 120 cells per condition examined over four independent experiments. Figure 7j–l, images are representative for $n$ = 120 cells per condition derived from four independent experiments. Figure 7m, mean ± s.e.m, two-sided Mann–Whitney U test, $n$ = 69, 26, and 22 cells per group derived from four independent experiments. ***$p$ < 0.001. Figure 7n, determined were the cell sizes of $n$ = 52 cells per condition in two separate experiments. Mean ± s.e.m, unpaired two-sided students $t$-test. ns, not significant, ***$p$ < 0.001. Figure 7o, p, images are representative for $n$ = 100 cells per condition derived from three independent experiments. Figure 7q, r, images are representative for $n$ = 100 cells per condition derived from three independent experiments. Figure 7s, mean ± s.e.m, unpaired two-sided students $t$-test, $n$ = 7, 9 cells per indicated group. ***$p$ < 0.001. Figure 7t, images are representative for $n$ = 130 cells examined over three autonomous experiments. Figure 7u, images are representative for $n$ = 130 cells examined over three autonomous experiments. Figure 7v, images are representative for $n$ = 130 cells examined over three independent experiments. Figure 7w, images

are representative for $n = 130$ cells examined over three independent experiments. Figure 8a, b, images are representative for $n = 80$ aISVs per genotype examined over four separate experiments. Figure 8c, d, images are representative for $n = 76,75$ aISVs per genotype examined over three independent experiments. Figure 8e, mean ± s.e.m, two-sided Mann–Whitney U test, $n = 35$ aISVs per genotype, ***$p < 0.001$. Figure 8f, mean ± s.e.m, two-sided Mann–Whitney U test, $n = 34, 48$ aISVs per indicated genotype, ***$p < 0.001$. Figure 8g, mean ± s.e.m, two-sided Mann–Whitney U test, $n = 51, 70$ aISVs/genotype. ns, not significant. Figure 8h–k, images are representative for $n = 96, 83, 72, 79$ examined aISVs per genotype derived from three independent experiments. Figure 8l–n, images are representative for $n = 116, 103, 112$ examined aISVs per genotype. Figure 8o, (left panel) mean ± s.e.m, unpaired two-sided students $t$-test, $n = 18, 19, 13, 14$ aISVs per indicated genotype. ns, not significant; ***$p < 0.001$; (right panel) mean ± s.e.m, unpaired two-sided students $t$-test, $n = 21, 24, 15, 12$ cells per genotype. ns, not significant; ***$p < 0.001$. Figure 8p–s, images are representative for $n = 84, 78, 89, 92$ aISVs per genotype. Figure 8t, mean ± s.e.m, unpaired two-sided students $t$-test, $n = 15, 21, 16$, and 13 cells per indicated genotype, ***$p < 0.001$. Figure 8u, $n = 12, 12, 10, 18$ cells per indicated genotype, mean ± s.e.m. Figure 8v, w, mean ± s.e.m, unpaired two-sided students $t$-test, $n = 8, 10, 10$ (v) and $n = 8, 16, 14$ (w) embryos per group. *$p < 0.05$, **$p = 0.0061$. Figure 9a, mean ± s.e.m, unpaired two-sided students $t$-test, $n = 8, 9, 8, 10$ embryos per group. **$p = 0.0011$; ***$p = 0.0001$. Figure 9b, mean ± s.e.m, unpaired two-sided students $t$-test, $n = 18, 20, 22, 16$ cells derived from six biologically independent embryos per group; ***$p < 0.001$; **$p = 0.0055$; *$p = 0.0100$.

**Reporting summary**. Further information on research design is available in the Nature Research Reporting Summary linked to this article.

## Data availability
The authors declare that all data supporting the findings of this study are available within the article and its Supplementary files or from the corresponding author upon reasonable request.

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

## Acknowledgements
We thank the colleagues of the KIT-European Zebrafish Resource Center (EZRC) for handling and maintenance of the zebrafish lines. We are grateful to Joachim Berger, Australian Regenerative Medicine Institute, Monash University, Clayton for sharing the p5E_503unc entry clone. J.D.v.B. is supported by a grant of the LSBR foundation (#2267) and NWO-Vici. K.A. is supported by the Helsinki Institute of Life Sciences and Hospital District of Helsinki and Uusimaa Research Grant (TYH2018201). S.S.-M. and F.l.N. are members of the EuFishBioMed zebrafish initiative. F.l.N. and S.S.-M. are supported by a grant from the Deutsche Forschungsgemeinschaft (DFG)—FOR2325 'Interactions at the Neurovascular Interface', I.S. is funded by the CiM Graduate School, Münster.

## Author contributions
A.K., J.v.R., and A.S.R. designed and performed experiments, interpreted experimental data, and participated in manuscript preparation. J.H., M.M., L.D., and L.P. performed experiments and analyzed data. R.W. designed and analyzed Flt1-HA knockin and transgenesis experiments, and interpreted experimental data. R.H., C.K., R.K., and K.A. participated in the conceptual development, manuscript preparation, and editing. R.V. and S.S. performed computational analyses, S.S.-M. and I.S-M. generated the *flt4* mutant and performed experiments. J.D.v.B. supervised the in vitro experiments, provided conceptual input, and contributed to editing of the manuscript. F.l.N. conceived and designed the project, performed experiments, analyzed data, supervised the overall project, and wrote the manuscript, with input from all co-authors.

## Funding

## Competing interests
The authors declare no competing interests.
