## [Peer Review File · Nature Communications]

Reviewers' Comments:

Reviewer #1:

Remarks to the Author:

This manuscript analyzes the role of Rac1 and its GEF Trio in artery diameter and endothelial cell spreading. They report that plgf over-expression or Flt1 knockdown in zebrafish embryos expands arterial ISVs via an increase in endothelial cell area with modest effects on cell number. These effects are attributed to increased vegfaa availability. In vitro experiments show that VEGF stimulation of HUVECs activates Rac1 and triggers cell spreading and cell-cell junction formation, which requires Trio. Trio over-expression also induces cell and artery expansion in vivo, and both Trio. Plgf opposes the effect of hyperglycemia in reducing aISV diameter, suggesting it may be a treatment for diabetic vasculopathy.

This is an extensive study combining in vivo and in vitro approaches and many state of the art techniques. Most of the data are presented clearly and in and of themselves appear convincing. What is much more problematic is how these data fit with the literature and with each other, and how they are interpreted. There are many discrepancies with the literature with no effort at reconciliation or in most cases even recognition. Many results are drastically over-interpreted. A major limitation for the entire study is the applicability of embryonic conditions in zebrafish to adult humans. For example, there is little evidence that VEGF induces vascular leak in early embryos or that hyperglycemia is inflammatory in this setting. In the end, it is difficult to know what to make of it.

Specific comments:

1. Intro: the inflammatory component of artery remodeling is seen in postnatal animals, it is highly unlikely that it plays any role at these early stages of development. In any case, no evidence is cited to support this view.
2. The statement that "One component thus far neglected in generating arteries with a structurally larger lumen is to enlarge the size of the endothelial cells lining the vessel segment" ignores the main conclusion from ref 10, Sugden 2017 Nat Cell Biol. 2017 Jun;19(6):653-665 (incorrectly cited as Siekmann et al.) That paper showed changes in aortic lumen area due to increased endothelial area.
3. Fig 1. It is a concern that over-expression of vegfaa and suppression of flt1 or expression of plgf give distinct outcomes, yet the authors conclude that suppression of flt1 and overexpression of plgf work by increasing vegfaa availability. Flit1 might transmit signals directly for example. Similarly, the authors treat plgf as if its only function is to displace vegfaa from Flt1, which is certainly not the case. Their results that plgf has a markedly stronger effect than removing Flt1 (fig 1n), and that inhibiting VEGFR2 does not fully block the effect of plgf over-expression (Fig 2g) disfavor their simple interpretation that vegfaa availability is the only determinant of arterial diameter.
4. Fig 2h lacks a positive control to validate effective knockdown of Flt4.
5. Fig 2l-p provides evidence that arterial diameter is independent of blood flow. This result contradicts published papers that examine the dorsal aorta (refs 10, 15) but it may well be that ISVs at this stage are regulated differently. The authors need to report the diameter of the dorsal aorta as a positive control to address whether their results disagree with the published literature or the aorta and ISVs are regulated differently.
6. Published papers, not cited or discussed, report that Rac activation by VEGF in endothelial cells is mediated by other GEFs. See Abraham et al. Nat Commun. 2015 Jul 1;6:7286 and Garrett et al Exp Cell Res. 2007 Sep 10;313(15):3285-97. In fact, it is unusual for a single GEF to fully mediate

responses to growth factors. The critical experiment that supports Trio as the only GEF that mediates VEGF activation of Rac1 uses a single shRNA sequence with no controls, thus, is questionable.

7. It is puzzling that the authors go to the trouble of mapping the Trio domain that activates rac when this is well known to be GEF1.

8. Regarding the in vitro experiments, it is well established that Rac activation results in increased cell spreading in many systems including endothelial cells. This part of the paper is of limited novelty. That Trio appears to specifically affect junctional Rac and tension within the junctional actin cables is of higher interest and novelty. It is consistent with previous work linking Trio to endothelial cell-cell junctions (ref 66).

9. The experiments with inhibitors of myosin in Fig 5 are relatively crude and hard to interpret. Myosin could contribute to endothelial cell structure in many different ways, it is not clear what role it has in the observed effects.

10. The experiment in Fig 5 with suppression of VE-cadherin is also hard to interpret. Is VE-cadherin replaced by N-cadherin? Are cell-cell junctions lost? Are junctions maintained without cadherins? These questions could be addressed by staining for b-catenin and N-cadherin.

11. The experiment in Fig 6 in which over-expression of TrioN and plgf are combined does not support a linear pathway in which VEGF activates Trio to induce cell enlargement. The additive effects instead suggest independent, parallel pathways.

12. It is not at all clear that the experiments in fig 6 that test effects of plgf and TrioN in hyperglycemia have any relevance to human disease. The effects of hyperglycemia in adult mammals are strongly dependent on oxidative stress and inflammation. No evidence is presented for similar effects in zebrafish embryos. Moreover, the effects seem to be essentially additive, with hyperglycemia causing smaller arteries while TrioN and plgf induce larger arteries in ways that appear largely independent. While I understand the temptation to add results from a poorly characterized to boost the paper's supposed impact, the authors may be better off saving this for a more detailed study where they can assess the effects of hyperglycemia in more depth.

13. Discussion: As noted above, the claim that "The major novel finding reported here is that endothelial cells can dynamically enlarge, thus increasing the lumen diameter of growing arteries" is not novel, having been reported in ref 10. What could conceivably be novel is the role of Trio in driving that expansion, however, the in vivo role for Trio is not adequately addressed, for example, by using morpholinos in the developing embryo.

Reviewer #2:

Remarks to the Author:

The authors described a novel mechanism whereby the developing arterial lumen is expanded in size. The authors convincingly showed that Rac1 and Trio cell autonomously function to modulate this process. While the main message of the manuscript is novel and interesting, some parts require additional improvements prior to publication.

Major comments:

1. In aISVs, the expansion of endothelial cells is not restricted by the vascular smooth muscle. However, in larger caliber vessels in post-developmental stages, endothelial cells are usually sheathed by vascular smooth muscle. Therefore, the model provided by the authors may not be applicable for post-embryonic stages. Therefore, the authors should provide more evidence indicating that Rac1-Trio regulation on lumen size could be universally applied, or tone down the

conclusion.

2. There are a number of known regulators for Rac1. Although Trio is known to regulate the function of Rac1 as the authors indicated, it is possible that multiple regulators coordinate to regulate the function of Rac1 in this context. Therefore, the authors need to consider the effects of other regulators, or even the effects of different Rac1 regulators in combination.
3. The authors manipulated the expression level of Rac1/Trio in all arterial endothelial cells. Do authors find similar expansion of lumen size in other vascular beds?
4. The proposed model by the authors implies that coordination between adjacent endothelial cells are critical for luminal expansion. To test this interesting possibility further, the authors may wish to perform mosaic experiments by mixing Rac1/Trio activated endothelial cells with non-transfected endothelial cells to examine whether the increase of cell size is indeed coordinated.
5. Epistatic relationship between FLT1 and Trio needs to be examined in detail. The authors showed that there is no additive effect of PlGF and TrioN overexpression, but did not provide direct evidence indicating that PlGF is the major activator for TrioN.
6. Testing potential application of the Rac1-Trio mediated regulation of arteriogenesis in the hyperglycemia zebrafish model is interesting, but done in a cursory manner. It appears that this part of the manuscript was simply added on to increase the applicability of the authors' finding. For instance, the authors need to determine whether Rac1/TrioN manipulation has any additional impacts on hyperglycemic zebrafish such as altering the metabolism of arterial endothelial cells.

Minor comments:

1. It is not clear whether FLT1 is indeed a decoy receptor for VEGF signaling. Although it has a high affinity to VEGF-A ligand with a significantly less robust kinase activity compared to other VEGF receptors, it is nonetheless triggers phosphorylation of downstream targets. Therefore, it may be inappropriate to label FLT1 as a decoy receptor.
2. p7, tree should be three
3. Administration of LatB would be too harsh treatment. The authors should provide additional way to manipulate F-actin polymerization.

Reviewer #3:

Remarks to the Author:

I am not an expert in arterial remodelling but am commenting on issues related to Rac1.

As far as I could judge it, the employed Rac1 mutants (active and light-sensitive) and the procedure for assaying active, GTP-bound Rac1 are fine. However, there are some issues concerning the use of controls and the lack of data on possible effects on other members of the Rho family of small GTPases, particularly Rac1b and RhoG. Specifically, my concerns are as follows:

1. The Rac1 inhibitor used significantly inhibited aISV diameter expansion in plgfmusc embryos, suggesting that Rac1 was involved. However, this conclusion was based solely on results with the Rac1 inhibitor. Pharmacological inhibition likely co-inhibits also a splice isoform of Rac1, termed Rac1b, that may have quite different sometimes even antagonistic effects compared to Rac1. The authors should check if Rac1b is expressed in arterial endothelial cells (to the best of my knowledge there are no data available on expression of this isoform in blood vessels or arteries). If this is the case, the authors should go on and verify whether it has a functional role in cell shape changes in endothelial cells. Only if this possibility can be dismissed, the observed effects can be safely attributed to Rac1.
2. In Fig. 3f both the Rac1 and Trio inhibitors reversed the plgfmusc effect (relative to WT) by approx. 40%. In light of this (only) partial effect, several questions have arisen to me:
3. The authors showed that ITX3, a selective inhibitor of the Trio N-terminal RhoGEF domain,

significantly reduced the outward remodeling response of aISVs. However, through the N-terminal DH-PH unit Trio can also mediate GDP to GTP exchange on RhoG (for Ref. see doi: 10.4161/cam.21418). This dual specificity provokes the question of whether RhoG also has a function in changes of endothelial cell shape. Did the authors consider this possibility or did they even perform any experiments to rule it out?

4. The Rac1 inhibitor used interferes with the interaction between Rac1 and Trio but also between Rac1 and Tiam1. Did the authors analyse a possible involvement of Tiam1 in their system?

5. There are several isoforms of Trio (Trio A, B, C, D, and E). Did the authors analyse which of these isoforms was responsible for the increase in endothelial cell size?

Reviewer #4:

Remarks to the Author:

The current manuscript investigates methods of expanding vascular lumens using Zebrafish intersomitic vessels as a model system. They generate multiple new lines that aid in protein localization and that increase bioavailable VEGF_{aa}. Increasing VEGF_{Gaa} in three experimental ways results in expanded ISV diameters in a VEGFR2-dependent, VEGFR3/flow-independent manner. The cellular mechanism is by EC proliferation and cell size increase. The authors also perform cell biology experiments showing that TrioN/actin dynamics also increase vessel diameter through increasing cell size. Finally, simultaneously increasing VEGF_{aa} and activating TrioN has an additive effect resulting in functional vessels with 2.5X the normal size. Methods of regulating vessel size could lead to clinically relevant ways of increasing blood flow. It is known that VEGF can increase vessel diameters, but the current study more thoroughly examines this in an *in vivo* setting. It is also not surprising that the known modulators of the actin cytoskeleton Trio and Rac1 increases cell size, but the authors test this in an intact vasculature.

Major points

1. The data demonstrating that vessels are non-leaky is hard to interpret and not quantified. Can the authors show larger views in Fig. 1M and include a quantification of the amount of dextran present in the perivascular space? The claim is also made in Fig. 6 for Plgf+TrioN without data or quantifications.
2. There is no effect with some morpholinos. The authors should confirm that the genes are indeed knocked down and that the lack of effects are not due to insufficient depletion (for *flt4* and *ve-cadherin*).
3. The authors claim that there is F-actin reorganization evident between control and experimental conditions in Fig. 3d,e and movies 2 and 3. It was not clear how they were measuring this and what they exactly observed.
4. The proper way to display deviations with the type of data in this paper is through error bars that indicate standard deviation, not SEM, which is in most graphs. Can the authors correct this?
5. Fig. 3f, inhibitor studies.
 - a. The experiment does not have WT alone with drugs. These conditions should be included to properly conclude the effect of the treatment. Is the phenotype reversal merely because this treatment decreases diameter or is it due to specific increases in cytoskeletal activity with Plgf?
 - b. The picture of the ISVs in this experiment are very small fields of view. Can the authors show the entire ISV as in Fig. 2? This will show whether the vessels normal except for decreased size or whether the decreased sizes are secondary to massive structural defects.
 - c. If the authors are suggesting that the cytoskeletal rearrangements inhibited here are leading to cell size, they should measure cell size with the inhibitor treatments.
6. Fig. 3M. There are loading inconsistencies between control and Trio siRNA that could falsely accentuate differences between the two groups. They should re-run the gel with the proper ratios.
7. Fig. 4b and c, suppl. Fig. 4a. The FRET signal does not appear to be concentrated at junctions, and the authors definition of junctional in this panel does not match with their definition in Fig. 4l.

Do the authors have a junctional protein that can serve as a positive control? If not, they should not claim the signal is junctional.

8. There is no quantification for the findings in Fig. 4 g-i and the accompanying suppl. Figure panels.

9. There is no control for comparison for data in Suppl. Fig. 4e.

10. Fig. 5h-k. What is the biological significance of this panel and how does this result influence their model? Based on the focal adhesions shown in 5f and the known function of integrins, shouldn't the integrins be localized there instead of cell-cell junctions.

Minor points

1. Fig. 1E, sFlt1 seems to be heterogeneously expressed in ECs. Can the authors comment?

2. There are citation problems throughout the text. Please check and make sure all panels in suppl. are cited.

General response to the reviewers comments:

We would like to thank the reviewers for their constructive feedback and suggestions to improve the manuscript. Their remarks including “many state of the art techniques - most of the data are presented clearly” and “the main message of the manuscript is novel and interesting” are well taken. To address the reviewers’ concerns we performed a series of additional experiments and addressed all issues raised.

To substantiate the epistatic relationship we performed an additional set of *vegfaa*, *Vegfr2*, and *Trio* loss and gain of function experiments, as well as experiments using membrane bound *Flt1* (*mFlt1*) mutants and *Neuropilin-1* (*Nrp1*) mutants. At the cellular level we substantiate the role of other GEFs and RhoGTPases. We now show that *Trio*, besides *Rac1* also activates *RhoG*, and *RhoG* loss and gain of function experiments show that *RhoG* contributes to endothelial cell shape changes. Moreover, we provide evidence showing that *Trio* directs the GEF *Tiam1* toward junctional regions, and *Tiam1* gain of function results in larger endothelial cells.

We furthermore substantiated the role of *Trio* in regulating endothelial cell size in WT and *Plgf* gain of function embryos. We now show that during the early stages of development *Trio* mediates endothelial cell enlargement, and loss of *Trio* results in smaller EC and smaller arteries. At later stages, *Trio* accounts for diameter changes upon loss of *endoglin*. We furthermore show that the impact of *Trio* is not limited to aISV but can also be observed in other vascular beds, such as the cerebral circulation as well as in a series of human arterial endothelial cells.

Response to Reviewer #1:

Reviewer #1 (Remarks to the Author):

This manuscript analyzes the role of *Rac1* and its GEF *Trio* in artery diameter and endothelial cell spreading. They report that *plgf* over-expression or *Flt1* knockdown in zebrafish embryos expands arterial ISVs via an increase in endothelial cell area with modest effects on cell number. These effects are attributed to increased *vegfaa* availability. In vitro experiments show that VEGF stimulation of HUVECs activates *Rac1* and triggers cell spreading and cell-cell junction formation, which requires *Trio*. *Trio* over-expression also induces cell and artery expansion in vivo, and both *Trio*. *Plgf* opposes the effect of hyperglycemia in reducing aISV diameter, suggesting it may be a treatment for diabetic vasculopathy.

This is an extensive study combining in vivo and in vitro approaches and many state of the art techniques. Most of the data are presented clearly and in and of themselves appear convincing. What is much more problematic is how these data fit with the literature and with each other, and how they are interpreted. There are many discrepancies with the literature with no effort at reconciliation or in most cases even recognition. Many results are drastically over-interpreted. A major limitation for the entire study is the applicability of embryonic conditions in zebrafish to adult humans. For example, there is little evidence that VEGF induces vascular leak in early embryos or that hyperglycemia is inflammatory in this setting. In the end, it is difficult to know what to make of it.

Response: We thank reviewer 1 for the kind remarks regarding “state of the art techniques” and “data are presented clearly and in themselves appear convincing”.

We are grateful for the constructive feedback in particular with regard to the suggestion to investigate a potential involvement of *Trio* in the *Endoglin* response, and the activation of *RhoG* by *Trio*. We furthermore extended the flow part, and repeated the experiments performed in *flt4* and *ve-cadherin* morphants now using *flt4* and *ve-cadherin* (*cdh5*) mutants.

Remark 1. There is little evidence that VEGF induces vascular leak in early embryos.

Answer: in line with the reviewer's suggestion we now show remodeling defects and vascular leakage in the inducible *vegfaa* gain-of-function scenario (new Supplement Figure 1b,d,e-w).

New Supplement Figure 1T-V: *In vivo* confocal imaging of plasma extravasation in inducible *vegfaa* gain of function embryo after injection of Dextran-Texas Red. Note the aberrant morphology of aISVs (T) and extensive plasma extravasation (U).

Question 1: Intro: the inflammatory component of artery remodeling is seen in postnatal animals, it is highly unlikely that it plays any role at these early stages of development. In any case, no evidence is cited to support this view.

Answer: We agree with the reviewer that relatively few studies exist addressing the role of macrophages and inflammatory components in embryonic vascular remodeling. The reason why we decided to address macrophages is because studies in mouse attribute the pro-arteriogenic effect of Plgf to activation of mFlt1 signaling in macrophages (*Pipp et al, Circ. Res. 2003*). To rule out this possibility, we therefore decided to investigate Plgf in a macrophage loss of function model. We found no evidence for a role of macrophages in our setting.

Zebrafish embryos can regenerate organs like heart, spinal cord and tailfin, a process that requires activation of the inflammatory system, and recruitment of macrophages to injured areas (*Lai et al, eLife 2017; e25605*). In zebrafish embryos, *hif-1 α* (a transcriptional regulator of Vegf) regulates interactions between macrophages and endothelial cells starting with the mobilization of macrophages from the aorta-gonad-mesonephros (AGM). Macrophage ablation is sufficient to recapitulate the vascular phenotypes observed in *hif-1 α* mutants, suggesting a macrophage-dependent angiogenic process during development (*Gerri et al, Nat Commun. 2017*). In zebrafish embryos, macrophages have been shown to contribute to flow driven arterial outward remodeling in *gridlock* mutant zebrafish embryos (*Gray et al, ATVB, 2007*). *Gridlock* mutants develop a stenosis in the dorsal aorta, which triggers the recruitment of collaterals allowing bypassing of blood flow around the stenosis. Loss of macrophages significantly impaired the aortic collateral remodeling process and flow delivery in *gridlock* mutants. More recently, macrophages were shown to contribute to sprout-sprout anastomosis formation upon muscle injury in the trunk of the zebrafish embryo (*Gurevich et al, EMBO J, 2018*), and anastomosis formation of the DLAV (*Fantin et al, Development, 2010*). Similar observations were made in the mouse embryonic hindbrain and postnatal retina where macrophages guide and stabilize anastomosis formation between developing vessels involving Notch signaling (*Tammela et al, Nat Cell Biol, 2011*). We now cite some of these reports in the revised manuscript.

Question 2. The statement that “One component thus far neglected in generating arteries with a structurally larger lumen is to enlarge the size of the endothelial cells lining the vessel segment” ignores the main conclusion from ref 10, Sugden 2017 Nat Cell Biol. 2017 Jun;19(6):653-665 (incorrectly cited as Siekmann et al.) That paper showed changes in aortic lumen area due to increased endothelial area.

Answer: We apologize for the glitch in the reference. Indeed, Arndt Siekmann's group recently reported that loss of *endoglin* in zebrafish embryos resulted in a larger aortic lumen diameter due to an increase in aortic endothelial size. They showed that the TGF β co-receptor Endoglin functions to limit endothelial shape changes, and promoting arterial contraction in response to increased shear stress. In line with the reviewer's suggestion, we have extended the text of the introduction to indicate that Endoglin acts to restrict endothelial cell size in response to increases in flow. In addition, we performed experiments and investigated potential synergistic and antagonistic

effects of Trio and Endoglin in mediating arterial diameter and EC shape changes – see also response to item 13.

Question 3. Fig 1. It is a concern that over-expression of *vegfaa* and suppression of *flt1* or expression of *plgf* give distinct outcomes, yet the authors conclude that suppression of *flt1* and overexpression of *plgf* work by increasing *vegfaa* availability. Flt1 might transmit signals directly for example. Similarly, the authors treat *plgf* as if its only function is to displace *vegfaa* from Flt1, which is certainly not the case. Their results that *plgf* has a markedly stronger effect than removing Flt1 (fig 1n), and that inhibiting VEGFR2 does not fully block the effect of *plgf* over-expression (Fig 2g) disfavor their simple interpretation that *vegfaa* availability is the only determinant of arterial diameter.

Answer: we addressed the reviewer's concerns with new experiments. We fully agree with the reviewer that mFlt1 may transmit signals, and therefore performed additional experiments in *mflt1* mutant zebrafish embryos. In line with the reviewer's suggestion, we adapted the text on mFlt1 signaling and added the results of the new experiments with the *mflt1* mutant to the revised manuscript (**new Figure 1R-Y**). In addition, we performed experiments that provide an explanation for the difference between *Plgf* gain of function, and *flt1* loss of function. To substantiate the Vegf-R2 axis, we examined aISV diameter in *flt1*^{-/-} mutants treated with a higher dosage of R2-tyrosine kinase inhibitor and in *flt1*^{-/-} mutants injected with a *vegfaa* targeting morpholino (**new Figure 2A-F**). We furthermore addressed the potential contribution of neuropilin-1 receptors (**Suppl Fig.3h**). To address the role of mFlt1 signaling, we examined aISV diameter in *mflt1*^{-/-} deficient zebrafish embryos (*ref. Wild et al, Nat Commun, 2017*). We observed a very small but consistent increase in arterial ISV diameter in *mflt1*^{-/-} mutants (**new Figure 1r-u**). To determine if mFlt1 signaling is required for Plgf induced arterial remodeling, we next overexpressed *plgf* in *mflt1*^{-/-} mutants.

New Figure 1R-U: *In vivo* confocal imaging of aISV in WT (R); *mflt1*^{-/-} mutant, *mflt1*^{ka605} (S); *mflt1*^{-/-} combined with *plgf* gain of function, *plgf*^{musc} (T). The areas indicated by the red dotted boxes are displayed at higher magnification in the lower panels. (U) Quantification of images in R-T. Mean±s.e.m, *t*-test, n=7,13 and 24 for indicated scenario. **p<0.01, ***p<0.001. Note: significantly increased aISV diameter in *mflt1*^{-/-}+*plgf*^{musc}.

Overexpression of *plgf* in *mflt1*^{-/-} mutants resulted in a significant increase in aISV diameter (**new Figure 1r-u**). These data suggests that Plgf can induce arterial diameter growth in the absence of mFlt1 signaling.

Although both loss of *flt1* and *plgf* gain of function associate with a Vegf gain of function scenario, one component that may contribute to the difference in arterial diameter between *plgf*^{musc} embryos and *flt1*^{-/-} mutants involves the spatial distribution of arterial Flt1 and Kdr1 receptors. In *plgf*^{musc}, Flt1 and Kdr1 are both expressed in the same arterial ECs (**Figure 1g**). Such expression pattern is absent in *flt1*^{-/-} mutants as Flt1 is not expressed.

If Flt1-Kdr1 co-expression in arterial EC indeed contributes to the observed difference, the model predicts that reducing *flt1* expression in *plgf*^{musc}, should annihilate the difference between *plgf*^{musc} and *flt1*^{-/-} mutants. Nevertheless, arteries in *plgf*^{musc} transgenics injected with a *flt1* targeting morpholino should still show a degree of arterial diameter growth as the knockdown of *flt1* itself induces a *vegfaa* gain of function scenario, similar as in *flt1*^{-/-} mutants. We tested these assumptions, and in line with our hypothesis we indeed found that knock-down of *flt1* in *plgf*^{musc} embryos reduced arterial diameter growth, to the levels observed in the *flt1*^{-/-} mutants (**new Figure 1V-Y**).

New Figure 1V-Y: *In vivo* confocal imaging of aISV in *plgf^{musc}* (V), *plgf^{musc}* concomitant with morpholino knock-down of *flt1* (W; *plgf^{musc} + flt1-MO*), *flt1^{-/-}* mutant, *flt1^{ka601}* (X). The areas indicated by the red dotted boxes are displayed at higher magnification in the lower panels. (Y) Quantification of images in V-X. mean±s.e.m, *t*-test, n=13-40 aISVs for each scenario. ns, not significant, **p<0.01.

To address the reviewers comment on Vegfaa and R2 signaling we performed additional experiments using a slightly higher Vegfr2 inhibitor dosage; in addition we targeted *vegfaa* using an ATG blocking morpholino. In line with Vegfaa-Kdr1 signaling in *plgf^{musc}* embryos, we found a dose dependent reduction in aISV diameter upon inhibiting Kdr1/R2 signaling (**new Figure 2a-e**). With the high R2 inhibitor dosage we obtained about 90%-95% reduction in arterial growth. Furthermore, reducing *vegfaa* using a low dose *vegfaa* ATG targeting morpholino reduced lumen diameter growth in *plgf^{musc}* (**Figure 2a-f**). Although both inhibiting R2 or reducing *vegfaa* reduced the Plgf effect, there was still some residual diameter growth left (see dotted lines in panels E,F). This may either be due to incomplete blockade of the Vegf-Kdr1 signaling pathway with the used inhibitor or morpholino dosage. Alternatively this residual diameter growth was due to another signaling receptor beyond mFlt1, or Kdr1/R2, like neuropilin-1 (Lähtenvuo *et al*, *Circulation*, 2009).

New Figure 2A-F: (A-D) *In vivo* confocal imaging of aISV in WT (upper panels) and *plgf^{musc}* embryos (lower panels) treated with DMSO vehicle - control (A), low dose Vegf-R2 inhibitor (B), high dose Vegf-R2 inhibitor (C), or *vegfaa* ATG targeting morpholino (D). Red dotted box is displayed at higher magnification in panel below. (E) Quantification of aISV diameter at 50hpf upon R2 inhibition. Mean±s.e.m, ANOVA, n=15-20 for each scenario. **p<0.01, ***p<0.001. (F) Quantification of aISV diameter at 48hpf upon morpholino mediated knockdown of *vegfaa*. Mean±s.e.m, ANOVA, n=15-20 for each scenario. **p<0.01, ***p<0.001. Dotted line indicates average diameter in WT scenario.

Plgf can bind to neuropilin-1 receptors (*Mamluk et al, J Biol Chem, 2002*). To address a potential involvement of neuropilin-1, we examined *plgf* gain-of-function in *nrp1a* mutants. Overexpression of *plgf* significantly augmented aISV diameter (**new Suppl Fig. 3h**). Besides *nrp1a*, zebrafish express an additional *nrp1* orthologue, *nrp1b*. We started to generate a *nrp1b* mutant. To 100% prove that the observed residual 5-10% of diameter growth involves neuropilin-1 independent of mFlt1, Plgf should be expressed in *nrp1a*^{-/-};*nrp1b*^{-/-} double mutants, and *nrp1a*^{-/-};*nrp1b*^{-/-};*mflt1*^{-/-} triple mutants; the breeding of which takes > 1 year, which is beyond the current revision. We include the *nrp1a* mutant results in the revised MS and in line with the reviewer's suggestion address the potential role of Nrp1 and mFlt1 in mediating effects of Plgf overexpression. We adapted our text and now conclude that *kdrl* is the main signaling receptor responsible for the diameter effects upon overexpression of Plgf.

New Suppl. Fig. 3H: Quantification of Plgf induced aISV diameter growth in *nrp1a*^{-/-} mutants. Mean±s.e.m, t-test, ***p<0.001.

Question 4. Fig 2h lacks a positive control to validate effective knockdown of Flt4.

Answer: to rule out ineffective knockdown, we repeated all experiments in *flt4*^{-/-} mutants.

We furthermore provide data showing the effective knockdown of *flt4* using the morpholino approach. The results obtained in *flt4*^{-/-} mutants are comparable to those obtained in *flt4*^{-/-} morphants (see **new Figure 2g-k**, and **Supplement Figure 3f-g**). Loss of *flt1* in *flt4*^{-/-} mutants significantly increased aISV diameter, to the level we reported with the *flt4* targeting morpholino in *flt1*^{-/-} mutants. Likewise arterial diameter was significantly larger in *plgf*^{musc} crossed with *flt4*^{-/-} embryos, when compared with WT or *flt4*^{-/-} mutant alone.

new Figure 2g-k: (G-J) *In vivo* confocal imaging of WT (G), *flt4*^{-/-} mutant, *flt4*^{mu407} (H), *flt4*^{-/-} mutant injected with *flt1* ATG targeting morpholino (I), *flt4*^{-/-} mutant injected with *plgf*^{musc} plasmid (J). (K) Quantification of images in G-J.

To validate the efficacy of *flt4* morpholino knock-down we examined the percentage of aISVs and vISVs in the trunk vasculature: **new Supplement Figure 3f,g**. In WT this ratio is 50:50; effective loss of *flt4* inhibits venous sprouting, resulting in a trunk predominantly consisting of aISVs (see images in **Supplement Figure 3a-d**; & *Wild et al, Nat Commun, 2017*). In line with *flt4* being effectively targeted we found that morpholino mediated knockdown of *flt4* resulted in shift in the AV ratio to 94% aISVs : 6% vISV. The data obtained in *flt4* mutants and *flt4* morphants suggest that Flt4 is not required for Plgf induced arterial growth.

New Supplement Figure 3F, G: (F) Perfusion characteristics of trunk ISVs at 2dpf, of three *plgf*^{musc} embryos, and three *plgf*^{musc} embryos injected with *flt4* targeting morpholino. Arrows indicate flow direction, red is aISV, blue is vISV. All ISVs were perfused. (G) Quantification of arterial-venous vessel identity in *plgf*^{musc}, and in *plgf*^{musc} injected with *flt4*

targeting morpholino. red is aISV, blue is vISV. Note a significant shift in the aISV/vISV ratio from about 50:50 to 96:4 ***p<0.001.

Question 5. Fig 2l-p provides evidence that arterial diameter is independent of blood flow. This result contradicts published papers that examine the dorsal aorta (refs 10, 15) but it may well be that ISVs at this stage are regulated differently. The authors need to report the diameter of the dorsal aorta as a positive control to address whether their results disagree with the published literature or the aorta and ISVs are regulated differently.

Answer: we fully agree with the reviewer that blood flow is an important regulator of arterial diameter. We also agree with the reviewer that optimal shear stress values may vary between different vascular beds, in particular between the central conductance arteries and the arterioles of the microcirculation.

References 10 and 15 used the L-type calcium channel blocker nifedipine to reduce heart rate and cardiac output. To be compatible with references 10 and 15 we repeated our experiments with nifedipine. Blood flow is an important regulator of arterial diameter, and an increase in shear stress can promote outward remodeling. To rule out increased flow as trigger for Plgf induced arterial remodeling, we exposed aISVs of *plgf^{muscle}* embryos to low flow conditions. To reduce heart rate and trunk perfusion, we treated WT and *plgf^{muscle}* embryos with the L-type calcium channel blocker nifedipine (**Figure 2o-r**). WT embryos exposed to nifedipine showed reduced aISV and aorta diameter (**Figure 2o-r; Supplement Fig. 4a-d**).

However, *plgf^{muscle}* embryos exposed to nifedipine still showed a significant increase in aISV diameter when compared to WT with normal flow, or WT with nifedipine (**Figure 2r**).

New Figure 2O-R: (O-Q) *In vivo* confocal imaging of WT (O), WT treated with L-type calcium channel blocker nifedipine (P), and *plgf^{muscle}* treated with nifedipine (Q). (R) Quantification of images in O-Q. Mean±s.e.m, *t*-test, n=11-20 aISVs per treatment group. ***p<0.001.

New Supplement Figure 4A-D: (A-C) *In vivo* confocal imaging of the dorsal aorta in WT (A), WT treated with nifedipine (B), and *plgf^{muscle}* treated with nifedipine (C). (D) Quantification of images in A-C. Mean±s.e.m, *t*-test. ns, not significant, *p<0.05, **p<0.01.

Similar observations were made in *tnnt2* morphants (**new Figure 2s-v**). Knocking down cardiac troponin T2 (*tnnt2*) is an established approach for creating a silent heart and block trunk perfusion. *Tnnt2* morphants showed a reduced aISV size and aortic diameter (**Figure 2v; Supplement Fig. 4h-k**). However, despite the loss of flow, aISV size was still larger in *plgf^{muscle}* embryos injected with *tnnt2* targeting morpholino, when compared to WT with flow, or WT injected with *tnnt2* morpholino (**Figure 2v**). While *tnnt2* morphants showed deficits in lumen formation (**Figure 2t**), *plgf^{muscle}* – *tnnt2* morphants showed a clear lumen on confocal sections of 3D stacks (**Figure 2u**).

New Figure 2S-V: (S-U) Confocal cross-section to show lumen dimensions of aISV in WT (S), WT injected with

tnnt2 targeting morpholino (T), and *plgf^{muscle}* embryo injected with *tnnt2* targeting morpholino (U). (V) Quantification of images in S-U. Mean±s.e.m, *t*-test, n=20 aISVs per group. ***p<0.001.

Question 6. Published papers, not cited or discussed, report that Rac activation by VEGF in endothelial cells is mediated by other GEFs. See Abraham et al. Nat Commun. 2015 Jul 1;6:7286 and Garrett et al Exp Cell Res. 2007 Sep 10;313(15):3285-97. In fact, it is unusual for a single GEF to fully mediate responses to growth factors. The critical experiment that supports Trio as the only GEF that mediates VEGF activation of Rac1 uses a single shRNA sequence with no controls, thus, is questionable.

Answer: we agree with the reviewer and now provide evidence showing that other GEFs and the RhoGTPase RhoG may be involved as well. As the reviewer indicated, Vegf has been linked to the activation of RhoG, with RhoG subsequently activating Rac1 through ELMO and Dock180 (Abraham et al, Nat Commun, 2015). In addition, the GEF1 domain of Trio has been shown to directly activate both Rac1 and RhoG (Bellanger et al, Nat Cell Biol, 2000). We therefore decided to examine the effect of RhoG loss and gain of function on endothelial cell size. Overexpression of a dominant active RhoG form (*GFP-RhoG-Q61L*) significantly increased endothelial cell area. Conversely, silencing of RhoG (*shRhoG-461*) in endothelial cells overexpressing *TrioN*, significantly reduced the *TrioN* induced cell size increase. We conclude that, in agreement with the suggestions made by the reviewer, that RhoG could play a role in Trio induced endothelial cell enlargement. We added the indicated references to our revised manuscript and we discuss them in light of our findings. We furthermore included a second shRNA against Trio (new Suppl. Fig. 8J,K).

New Figure 4M,P: (M) Surface area of ECs transfected with control plasmid or constitutive active RhoG Q61L. Note overexpression of *RhoG* increases EC size. (P) Changes in EC size upon *TrioN* gain of function after silencing of *RhoG*. Mean \pm s.e.m, *t*-test, $n=101-146$ cells/group. Note: *TrioN* induced EC enlargement requires *RhoG*.

Besides Trio, there are a number of other known regulators for Rac1, including Tiam1, and multiple regulators may coordinate to regulate the function of Rac1. Interestingly, in endothelial cells overexpressing *TrioN*, we observed that Tiam1 was expressed in junctional regions, as opposed to the more cytosolic localization observed in control transfected cells (new Figure 4q-v). Endothelial overexpression of *Tiam1* resulted in larger endothelial cells, similar to the effects observed upon overexpression of *TrioN* (new Figure 4w). Combining *TrioN* and *Tiam1* overexpression had no additive effect on increasing EC size when compared to *TrioN* alone (Figure 4w). These data suggest that one part of cell enlargement may require careful spatial positioning of GEFs, and

activation of additional RhoGTPases including RhoG. How Trio achieves this spatial positioning of other GEFs is an exciting topic, but we feel it is beyond the scope of the current manuscript.

New Figure 4Q-W: (Q-S) EC transfected with control plasmid (ctrl) and immune stained for endogenous Tiam1 (Q), F-actin with phalloidin (R) and VE-Cadherin as junction marker (S). Tiam1 localizes cytosolic and to a lesser extent to junction regions. (T-V) Endothelial cells transfected with *TrioN* and immune stained for endogenous Tiam1 (T), F-actin with phalloidin (U) and VE-Cadherin as junction marker (V). Tiam1 localizes at junction regions (arrowheads). (W) EC size of ECs transfected with *Tiam1* (red bar), *TrioN* (green bar), or transfected with both *Tiam1* and *TrioN*. Mean \pm s.e.m, *t*-test. ns, not significant, *** $p < 0.001$. Scale bar, 20 μm .

Question 7. It is puzzling that the authors go to the trouble of mapping the Trio domain that activates rac when this is well known to be GEF1.

Answer: we agree with the reviewer that Rac1 is an established target of the Trio-GEF1 domain. Here we just wanted to firmly establish that the Trio-GEF1 domain is required for endothelial cell enlargement and vessel diameter.

Question 8. Regarding the in vitro experiments, it is well established that Rac activation results in increased cell spreading in many systems including endothelial cells. This part of the paper is of limited novelty. That Trio appears to specifically affect junctional Rac and tension within the junctional actin cables is of higher interest and novelty. It is consistent with previous work linking Trio to endothelial cell-cell junctions (ref 66).

Answer: to substantiate the mechanism, we now show that upon Trio overexpression, Tiam1 becomes localized at junctional regions. We show that Tiam1 gain of function results in larger endothelial cells. We furthermore provide evidence showing that RhoG gain of function results in larger endothelial cells, and that RhoG is also required for Trio induced cell enlargement.

Question 9. The experiments with inhibitors of myosin in Fig 5 are relatively crude and hard to interpret. Myosin could contribute to endothelial cell structure in many different ways, it is not clear what role it has in the observed effects.

Answer: to address this question we performed additional experiments. We agree with the reviewer that blebbistatin will inhibit myosin II in all parts of the cell and may therefore not only inhibit junctional tension. When using GFP-tagged Myosin II we observe that Myosin II localizes on F-actin bundles throughout the cell, but it is most prominently present on junctional F-actin bundles (**new Figure 5G,H**). Also, the majority of active myosin, distinguished by staining for mono-phosphorylated myosin light chain S19, localizes strongly at the junction region (**new Figure 5I,J**). In addition, we have observed that fully di-phosphorylated myosin light chain (T18/S19) localizes even more specifically to junctional F-actin bundles (**new Figure 5K,L**). For these reasons we believe the effects of blebbistatin will predominantly affect the active myosin II at the junction region and thus junctional tension. However, we cannot exclude its inhibition of non-junctional myosin II may also contribute to reduced cell size. We adapted the text of the manuscript to make these points more clear.

New Figure 5G-L: (G) EC transfected with *mCherry* and *GFP-MyosinII*, and stained for F-actin (magenta) and VE-cadherin (white). Myosin-II localizes at actin bundles (arrowheads). (H) EC transfected with *mCherry-TrioN* and *GFP-MyosinII*, and stained for F-actin and VE-cadherin. Myosin-II localizes at junctional actin bundles (arrowheads). (I) EC transfected with *GFP* and stained for pMLC-S19, F-actin and VE-cadherin. pMLC-S19 localizes at actin bundles throughout the cell (arrowheads). (J) EC transfected with *GFP-TrioN*, and stained for pMLC-S19, F-actin and VE-cadherin. pMLC-S19 localizes at junctional and peripheral actin bundles (arrowheads). (K) EC transfected with *GFP*, and stained for pMLC-T18S19 (red) and F-actin. (L) EC transfected with *GFP-TrioN*, and stained for pMLC-T18S19 and F-actin. pMLC-T18S19 localizes at junctional regions (arrowheads). Scale bar, 20 μ m.

Question 10. The experiment in Fig 5 with suppression of VE-cadherin is also hard to interpret. Is VE-cadherin replaced by N-cadherin? Are cell-cell junctions lost? Are junctions maintained without cadherins? These questions could be addressed by staining for b-catenin and N-cadherin.

Answer: To address these questions we analyzed expression of N-Cadherin, and junction integrity in endothelial cells in which overexpression of TrioN was combined with suppression of VE-Cadherin. We find that N-Cadherin can compensate for the loss of VE-Cadherin. We furthermore show that upon *TrioN* expression, junction integrity is maintained, even in the absence of VE-Cadherin or alpha-catenin expression.

We addressed VE-Cadherin because it was reported that Trio may interact with VE-Cadherin at junctions (*Timmerman et al, J Cell Sci, 2015*). Knock-down of *VE-Cadherin* impairs endothelial cell-cell junctions, but endothelial junctions may not be completely lost as other junction molecules such as PECAM, JAMs, ESAM, Claudins and N-Cadherin can partly compensate (*Carmeliet et al., Cell 1999; Frye et al, J. Exp Med 2015; Duong CN et al, ATVB 2020*). In line with this, we find expression of N-Cadherin in junction regions in endothelial cells in which overexpression of *TrioN* was combined with suppression of *VE-Cadherin* (**new Supplement Figure 10A-L**).

New Suppl Fig 10A-L. (A-C) EC treated with shCTRL and transfected with *GFP* (C) and stained for VE-cadherin (A) and N-cadherin (B). (D-F) EC treated with shCTRL and transfected with *GFP-TrioN* (F) and stained for VE-cadherin (D) and N-cadherin (E). (G-I) EC treated with shVE-cadherin and transfected with *GFP* (I) and stained for VE-cadherin (G) and N-cadherin (H). (J-L) EC treated with shVE-cadherin and transfected with *GFP-TrioN* (L) and stained for VE-cadherin (J) and N-cadherin (K).

Endothelial cells gain stability by making contacts with neighboring cells through VE-cadherin-based cell-cell junctions. Junctional strength can be measured in an electrical cell-substrate impedance (ECIS) assay. Endothelial monolayers transfected with *TrioN* *in vitro* showed augmented junctional integrity as indicated by increased electrical cell-substrate impedance (**new Figure 6a**). This increase in junctional strength was independent of VE-Cadherin as *TrioN* overexpression augmented electrical cell-substrate impedance even in the absence of *VE-Cadherin* (**new Figure 6b,c**).

New Figure 6A-C: (A) EC were transfected with control plasmid or *TrioN* and cultured on Electrical Cell-Substrate impedance (ECIS) arrays. Electrical resistance was measured after 48 hours of culturing. *TrioN* expressing EC showed a significant increase in barrier integrity. Mean±s.e.m, *t*-test, **p*<0.001. (B) EC were transfected with control plasmid or *TrioN* and co-

transfected with control shRNA (shControl) or shRNA VEC to silence VE-Cadherin expression. ECs were cultured on Electrical Cell-Substrate impedance (ECIS) arrays. Electrical resistance was measured after 48 hours of culturing. Note the increase in resistance upon silencing of VE-Cadherin in *TrioN*. (C) Comparison of *TrioN* induced increase in impedance in control (delta shControl-shControl+*TrioN*) and shVEC (delta shVEC-shVEC+*TrioN*) transfected cells based on B. Note equal increase in resistance in both groups. Mean±s.e.m, *t*-test. ns, not significant, ***p*<0.01, ****p*<0.001.

In addition we would like to provide additional evidence at the discretion of Reviewer-1; results that we would only like to include in this “response to the reviewer” letter but not in the revised MS. To

substantiate our findings we next measured electrical conductance in monolayers of *TrioN* overexpressing endothelial cells, in which *alpha-catenin* was silenced and compared this with *TrioN* GOF cells with intact *alpha-catenin* expression. Loss of *alpha-catenin* results in inhibition of both VE-Cadherin and N-Cadherin function. We found that *TrioN* augmented electrical resistance, even upon loss of *alpha-catenin*; the change in resistance is comparable between the scenarios. We thus conclude that the positive effect of *TrioN* on junctional stability does not require VE-Cadherin or *alpha-catenin* associated Cadherins. How this is achieved mechanistically we believe is better suited for another manuscript.

Figure for Reviewer-1: Trio maintains junctions in the absence of α -catenin.

Electrical resistance in endothelial cells transfected with control shRNA (shCTRL, blue), cells transfected with *TrioN* and control shRNA (pink), cells transfected with shRNA targeting α -catenin (green), cells transfected with *TrioN* and shRNA targeting α -catenin. 3 separate experiments. Mean \pm SEM, t-test, **p<0.01, ns, not significant.

Question 11. The experiment in Fig 6 in which over-expression of *TrioN* and *plgf* are combined does not support a linear pathway in which VEGF activates *Trio* to induce cell enlargement. The additive effects instead suggest independent, parallel pathways.

Answer: we agree that this experiment may have caused some confusion. One factor to consider is that transgenic overexpression of *Trio* in endothelial cells generates supra-physiological levels of *Trio* (compare for example the *Vegf* overdose obtained in the constitutive and inducible *vegfaa* gain of function transgenics in Figure 1 & Suppl. Fig.1). This is in fact the reason why we decided to combine *TrioN* with *Plgf^{musc}* - to see what would happen if the system is maximally stimulated (maximal cell enlargement plus local EC proliferation) – and indeed we got supra-large arteries. We have rephrased the text to make this issue more clear. To substantiate the role of *Trio* in *plgf^{musc}* transgenics, we performed additional experiments (**new Figure 3G-T**), in which we show that loss of *Trio* reduces EC size and diameter growth in *plgf^{musc}* – see also response to question 12. The reduced arterial diameters upon loss of *Trio* or inhibition of *Trio* with ITX3 (**Figure 3F**) suggest that *Trio* mediated EC enlargement is one component that significantly contributes to the arterial growth observed in *plgf^{musc}* embryos – the other component is the increase in cell number.

Question 12. It is not at all clear that the experiments in fig 6 that test effects of *plgf* and *TrioN* in hyperglycemia have any relevance to human disease. The effects of hyperglycemia in adult mammals are strongly dependent on oxidative stress and inflammation. No evidence is presented for similar effects in zebrafish embryos. The authors may be better off saving this for a more detailed study where they can assess the effects of hyperglycemia in more depth.

Answer: we agree with the reviewer that the zebrafish diabetes model may require some additional characterization of important factors like oxidative stress and inflammation. To do so we have generated a series of mutants for genes implied in oxygen radical production including *Nox2*, *Nox4*, *Nox5*, and transgenics harboring redox sensors to monitor oxidative stress in macrophages and endothelial cells in zebrafish *in vivo*. We feel that these data are more suitable for a separate manuscript and have removed the diabetes results section from the revised manuscript.

Question 13. As noted above, the claim that “The major novel finding reported here is that endothelial cells can dynamically enlarge, thus increasing the lumen diameter of growing arteries” is not novel, having been reported in ref 10. What could conceivably be novel is the role of Trio in driving that expansion, however, the in vivo role for Trio is not adequately addressed, for example, by using morpholinos in the developing embryo.

Answer: In line with the reviewer’s suggestion, we performed additional experiments. We found that morpholino mediated knockdown of Trio resulted in a dose-dependent decrease of aISV diameter and EC surface area in *plgf^{musc}* embryos and WT (**new Fig. 3G-T**).

New Figure 3G-T: (G-I) Confocal imaging of aISV in WT (G), WT injected with 1.7 ng *Trio* targeting morpholino (H), and WT injected with 5 ng *Trio* targeting morpholino (I) in *Tg(kdrl:has.HRAS-mCherry)^{S916}*. (J-L) Confocal imaging of aISV in *plgf^{musc}* (J), *plgf^{musc}* injected with 1.7 ng *Trio* targeting morpholino (K), and *plgf^{musc}* injected with 5 ng *Trio* targeting morpholino (L) in *Tg(kdrl:has.HRAS-mCherry)^{S916}*. (M) Quantification of images in G-L. Mean±s.e.m, ***p<0.001. (N-P) Confocal imaging of aISV in WT (N), WT injected with 1.7 ng *Trio* targeting morpholino (O), and WT injected with 5 ng *Trio* targeting morpholino (P) in *Tg(fli1a:lifectEGFP)^{S916}*. (Q-S) Confocal imaging of aISV in *plgf^{musc}* (Q), *plgf^{musc}* injected with 1.7 ng *Trio* targeting morpholino (R), and *plgf^{musc}* injected with 5 ng *Trio* targeting morpholino (S) in *Tg(fli1a:lifectEGFP)^{S916}*. (T) Quantification of images in N-S. Mean±s.e.m. **p<0.01, ***p<0.001.

In zebrafish embryos, loss of the TGFβ co-receptor *endoglin* augments aortic diameter in response to flow increases (*Sugden et al, Nat Cell Biol. 2017*). Morpholino-mediated loss of *endoglin* increased aorta diameter at 3dpf, not 2dpf, in line with the *endoglin* mutant phenotype (**New Supplement Figure 12r,u**). We found no evidence for loss of *endoglin* and gain of *Trio* acting synergistically to influence aISV or aortic diameters in 2dpf embryos (**New Supplement Fig. 12a-i; Supplement Fig. 12j-r**). Loss of *endoglin* did not further augment aISV diameters in *flt1^{-/-}* mutants, nor in *plgf^{musc}* and *flt1^{enh}:TrioN* transgenics at 2dpf (**Supplement Fig. 12i**). *In vitro*, cell size of *TrioN* overexpressing cells was not significantly different from the size of *TrioN* expressing cells in which *endoglin* was silenced (**New Supplement Figure 12s,t**).

New Supplement Fig. 12S,T: (S) Western blot *Endoglin* (ENG) for control of knockdown efficiency using two different siRNA smart pools siENG-1 and siENG-2, and actin for protein loading control, as indicated. (T) Changes in endothelial cell size upon *TrioN* gain of function after silencing of *Endoglin* using two different siRNA smart pools siENG-1 and siENG-2 (as indicated); mean ± s.e.m, Kruskal-Wallis-test. ns, not significant.

----- response continued on the next page -----

At 3dpf *endoglin* morphants showed an increased aorta diameter, similar to the phenotype reported in *endoglin* mutants (**New Supplement Fig. 12u**). Since in the *endoglin* mutant, the change in aorta diameter was attributed to an increase in EC size and not EC number, we considered a potential contribution of Trio in mediating the diameter increase. Accordingly, loss of *Trio* in *endoglin* morphants, or inhibiting Trio function using ITX3 in *endoglin* morphants inhibited the aorta diameter increase and reduced EC surface area (**New Figure 7x,y**). Thus Trio may be required for loss of endoglin induced vessel caliber changes. This furthermore suggests that at 3dpf the TGF β pathway may act as an inhibitor of Trio function and diameter control.

New Figure 7X,Y: (X) Dorsal aorta diameter at 3dpf in *endoglin* morphants (red bar), *endoglin* morphants injected with *Trio* targeting morpholino (blue bar), *endoglin* morphants treated with Trio inhibitor ITX3 (dark green bar) and WT treated with ITX3 (light green bar). Note that loss of Trio or inhibiting Trio reduced diameter growth in *endoglin* morphants. Mean \pm s.e.m, *t*-test, ns, not significant; *** p <0.001; * p <0.05.

(Y) EC surface area at 3dpf in *endoglin* morphants (red bar), *endoglin* morphants injected with *Trio* targeting morpholino (blue bar), *endoglin* morphants treated with Trio inhibitor ITX3 (dark green bar) and WT treated with ITX3 (light green bar). Note that loss of Trio or inhibiting Trio reduced EC surface area in *endoglin* morphants. Mean \pm s.e.m, *t*-test; ns, not significant; *** p <0.001; ** p <0.01; * p <0.05.

Response to Reviewer #2:

Reviewer #2 (Remarks to the Author):

The authors described a novel mechanism whereby the developing arterial lumen is expanded in size. The authors convincingly showed that Rac1 and Trio cell autonomously function to modulate this process. While the main message of the manuscript is novel and interesting, some parts require additional improvements prior to publication.

Response: We would like to thank reviewer 2 for the kind comments. The statement that our “main message is novel and interesting” is well taken. We addressed the concerns of reviewer 2 and examined the more broader application of our findings, and the epistatic relation between Flt1, Trio and EC size. In addition we addressed the contribution of additional GEFs and RhoGTPases.

Question 1. In aISVs, the expansion of endothelial cells is not restricted by the vascular smooth muscle. However, in larger caliber vessels in post-developmental stages, endothelial cells are usually sheathed by vascular smooth muscle. Therefore, the model provided by the authors may not be applicable for post-embryonic stages. Therefore, the authors should provide more evidence indicating that Rac1-Trio regulation on lumen size could be universally applied, or tone down the conclusion.

Answer: We agree with the reviewer that the impact of Trio may be localized to a particular part of the circulation, and several layers of smooth muscle cells most likely restrict outward remodeling. We therefore down-tuned our statement and rephrased our text and fine-tuned in particular to which type of vessels our findings could apply, in embryos and in the adult setting. We furthermore performed additional experiments to show that Trio induced endothelial enlargement is conserved in other endothelial cells and vessel types.

In post-natal stages, in an adult carotid artery, or in a peripheral resistance artery, the multiple layers of smooth muscle cells, and collagen/elastin components, may restrict structural diameter remodeling. In these vessels, the contractile status of the smooth muscle cells will most likely prevail in determining lumen diameter and outward remodeling potential. In line with the reviewer’s suggestion we added a paragraph to the discussion addressing this issue. To substantiate the more general application of Trio induced EC enlargement, we explored if human endothelial cells

from other origins also enlarge upon overexpression of Trio. We examined Human Aortic Endothelial Cells (HAEC), and Human Umbilical Artery Endothelial Cells (HUAEC). Overexpression of *TrioN* significantly augmented the size of HAEC and HUAEC (new Supplement Figure 11f). Taken together we now demonstrate TrioN induced enlargement in HAEC, HUAEC and HUVEC (endothelium derived from vessels with smooth muscle).

New Suppl Fig. 11f: Trio augments size of human arterial endothelial cells. Cells were transfected with GFP-control or GFP-TrioN construct, and cell area measured 24 hours upon transfection. Measured were >20 cells from 3 separate experiments, Mean±SEM, t-test, *** p<0.001, HAEC, human aortic endothelial cells; HUAC, Human umbilical artery endothelial cells.

We next examined if the effect of Trio on arterial diameter *in vivo* is conserved when using alternative vascular promoters. The *kdrl* and *fli1a* promoter are ubiquitously expressed in endothelial cells of the zebrafish embryo trunk vasculature. For both *kdrl* and *fli1a* we observed significantly increased aorta and aISV diameters, and EC size, in both WT and in *plgf^{musc}* embryos (new

Figure 7v,w, & Supplement Fig. 11g,h).

New Figure 7V,W: (V,W) Diameter and EC surface in aorta in *kdrl:TrioN* or *fli1a:TrioN* overexpression scenario. Mean ± s.e.m, t-test. *p<0.05, **p<0.01.

Note larger EC and aorta size in the TrioN gain of function scenarios.

As Vegf is the major driver of Trio activation in our setting, we hypothesized that our findings regarding enlargement may be conserved in other *vegf* gain of function scenarios. Von Hippel Lindau (Vhl) is a protein relevant for probing Hypoxia Inducible Factor-1 α (HiF-1 α), the main driver of *vegf* expression, for proteasomal degradation. *Vhl*^{-/-} mutants show increased *vegfaa* expression, in particular in developing neuronal tissue (Wild et al, Nat Commun, 2017). We therefore decided to investigate arterial diameters in the cerebral vasculature of *vhl*^{-/-} mutants. *Vhl*^{-/-} mutants showed increased arterial diameters, without significant change in endothelial cell numbers (New Supplement Figure 11i-p). *Vhl* morphants also showed increased vessel diameter, concomitantly with increased EC surface area (New Supplement Figure 11t,u). Blocking Trio function using ITX3 inhibited the diameter increase in *vhl* morphants (New Supplement Figure 11u). This suggest that Trio mediated cell and vessel dimension changes is conserved in this Vegf gain of function scenario.

New Supplement Fig. 11I-U: (I-N) *In vivo* confocal imaging of cerebral vasculature (I,L) and endothelial cell nuclei (J,M), with merged image in K,N showing enlarged vessels in *vhl*^{-/-} mutants (*vhl*^{hu2117}) when compared with siblings. Scale bar indicates 20 μ m. (O,P) Quantification of the images in K,N. Note the increase in diameter without significant change in EC numbers in *vhl*^{-/-}. Sibling, 6 embryos/8-13 CtAs per embryo; *vhl*^{-/-} 8 embryos/10-13 CtAs per embryo. Mean \pm s.e.m, *t*-test, *** *p*<0.001; ns, not significant. (Q-S) *In vivo* confocal imaging of cerebral vasculature in WT (Q), *vhl* morphants (R), and *vhl* morphants treated with the Trio inhibitor ITX₃ (S). (T,U) Quantification of CtA EC area (T), and CtA diameter (U) and for indicated scenario. (T) WT, 5 embryos/4-5 CtAs per embryo; *vhl* morphant, 6 embryos/3-5 CtAs per embryo; *vhl* morphant + ITX₃ 6 embryos/3-5 CtAs per embryo. (U) WT, 11 embryos/3 CtA-ECs per embryo; *vhl* morphant, 10 embryos/2 CtA-ECs per animal. Mean \pm s.e.m, *t*-test, *** *p*<0.001, **p*<0.05; ns, not significant.

Question 2. There are a number of known regulators for Rac1. Although Trio is known to regulate the function of Rac1 as the authors indicated, it is possible that multiple regulators coordinate to regulate the function of Rac1 in this context. Therefore, the authors need to consider the effects of other regulators, or even the effects of different Rac1 regulators in combination.

Answer: we agree with the reviewer and analyzed additional regulators, and regulators in combination, as well as RhoG, and provide evidence for their role in mediating shape changes.

Besides Trio, there are a number of other known regulators for Rac1, including Tiam1, and multiple regulators may coordinate to regulate the function of Rac1 in this context. Interestingly, in endothelial cells overexpressing *TrioN*, we observed that Tiam1 was expressed in junctional regions, as opposed to the more cytosolic localization observed in control transfected cells (new Figure 4q-v). Endothelial overexpression of *Tiam1* resulted in larger endothelial cells, similar to the effects observed upon overexpression of *TrioN* (new Figure 4w). Combining *TrioN* and *Tiam1* overexpression had no additive effect on increasing EC size when compared to *TrioN* alone (new

Figure 4w). These data suggest that one part of cell enlargement may require careful spatial positioning of GEFs. How Trio achieves this spatial positioning of other GEFs is an exciting topic, but we feel it is beyond the scope of the current manuscript.

New Figure 4Q-W: (Q-S) Endothelial cells transfected with control plasmid (ctrl) and immune stained for endogenous Tiam1 (Q), F-actin with phalloidin (R) and VE-Cadherin as junction marker (S). Tiam1 localizes cytosolic and to a lesser extent to junction regions. Scale bar, 20 μm. (T-V) Endothelial cells transfected with *TrioN* and immune stained for endogenous Tiam1 (T), F-actin with phalloidin (U) and VE-Cadherin as junction marker (V). Tiam1 localizes at junction regions (arrowheads). Scale bar, 20 μm. (W) Endothelial cell size of ECs transfected with *Tiam1* (red bar), *TrioN* (green bar), or transfected with both *Tiam1* and *TrioN*. Measured were the cell-size of at least 20 cells derived from 3 separate experiments. Data are mean ± SEM. Mean ± s.e.m, *t*-test. ns, not significant, ****p*<0.001. Scale bar, 20 μm.

In addition, the GEF1 domain of Trio has been shown to directly activate both Rac1 and RhoG (Bellanger et al, Nat Cell Biol, 2000). We therefore decided to examine the effect of RhoG loss and gain of function on endothelial cell size. Overexpression of a dominant active RhoG form (*GFP-RhoG-Q61L*) significantly increased endothelial cell area (New Fig. 4m). Conversely, silencing of RhoG (*shRhoG-461*) in endothelial cells overexpressing *TrioN*, significantly reduced the *TrioN* induced cell size increase (new Fig. 4p). We conclude that, in agreement with the suggestions made by the reviewer, that RhoG could play a role in Trio induced endothelial cell enlargement. We added these data to the revised manuscript and added a paragraph to the discussion addressing these new results.

New Figure 4M,P: (M) Surface area of ECs transfected with control plasmid or constitutive active RhoG Q61L. Note overexpression of *RhoG* increases EC size. (P) Changes in EC size upon *TrioN* gain of function after silencing of *RhoG*. Mean ± s.e.m, *t*-test, n=101-146 cells/group. Note: *TrioN* induced EC enlargement requires *RhoG*.

Question 3. The authors manipulated the expression level of Rac1/Trio in all arterial endothelial cells. Do authors find similar expansion of lumen size in other vascular beds?

Answer: Unfortunately, the *flt1^{enh}* promoter is not active in all arterial endothelial cells. The activity of the *flt1^{enh}* promoter is restricted mainly to aISVs, and the aorta shows a mosaic *flt1^{enh}* expression pattern. We therefore included additional endothelial promoters and analyzed aortic and aISV diameter in *Tg(kdrl:TrioN)* and *Tg(fli1a:TrioN)* embryos. We found increased aortic and aISV- EC areas and diameters in both these scenarios (new Figure 7v,w; new Suppl. Fig. 11g,h). With regard to other vascular beds, please also see the response to question 1.

Question 4. The proposed model by the authors implies that coordination between adjacent endothelial cells are critical for luminal expansion. To test this interesting possibility further, the authors may wish to perform mosaic experiments by mixing Rac1/Trio activated endothelial cells with non-transfected endothelial cells to examine whether the increase of cell size is indeed coordinated.

Answer: In line with the reviewer's suggestion we performed the requested *in vitro* mosaic expression experiment (see new Supplement Fig. 10m & New Figure 6n). For this purpose we mixed GFP-TrioN transfected cells with mCherry-control transfected cells and measured endothelial cell size. In both the "mosaic mixed" and in the "homogenous expression" scenario we find that *TrioN* overexpression augmented EC size. The magnitude of cell size increase was not significantly different between the two scenarios (new Figure 6n).

New Figure 6n: Endothelial surface area upon mosaic overexpression of *TrioN*.

(left panel) Schematic illustration of the experimental setup. Control and TrioN expression cells were either not mixed (homogenous expression, left two lanes), or mixed (mosaic, right two lanes).

(right panel) Endothelial cells were transfected with mCherry-control, or GFP-TrioN, and either grown in separate dishes or mixed to obtain mosaic TrioN expression; in this case EC size was measure in EC expressing TrioN with neighbouring cells expressing the control plasmid. Trio overexpression augmented endothelial cell area to a similar extent, in both the "non-mixed" and the mosaic scenario. We measured the cell-area of at least 52 cells in two separate experiments. Data are mean ± SEM. ***p<0.001. NS = not significant.

Question 5. Epistatic relationship between FLT1 and Trio needs to be examined in detail. The authors showed that there is no additive effect of PlGF and TrioN overexpression, but did not provide direct evidence indicating that PlGF is the major activator for TrioN.

Answer: to address this question, we performed additional experiments. Plgf can bind to mFlt1 and to Neuropilin-1 receptors and initiate signaling. Alternatively, Plgf can displace Vegf from Flt1, and the displaced Vegf can subsequently activate Kdr1 signaling. To discern between these 3 different scenarios we performed Plgf gain of function experiments in *mFlt1*^{-/-} and *Nrp1*^{-/-} mutants. In addition we performed *Trio* loss of function experiments in *Plgf*^{musc} transgenics.

In zebrafish, Flt1 exists in two isoforms: membrane bound Flt1 (mFlt1) and soluble Flt1 (sFlt1). Soluble Flt1 acts as Vegf scavenging receptor. mFlt1 has weak tyrosine kinase signaling properties, but can also act as Vegf scavenging receptor. Flt1 has a 10 fold higher affinity for Vegf than Vegf-receptor-2/Kdr, hence, Vegf has a higher likelihood to bind to Flt1 than to Kdr. Through this binding principle, Flt1 can regulate Vegf bio-availability and thus Kdr signaling strength (in zebrafish *kdr1* signaling). Studies in mouse suggest that Plgf can affect angiogenesis via mFlt1 signaling. To rule out that Plgf-mFlt1 signaling accounted for arterial diameter growth we examined *mflt1* mutant zebrafish (Wild et al, Nat Commun, 2017). Overexpression of *plgf* in *mflt1*^{-/-} mutants resulted in arterial diameter growth, indicating that mFlt1 signaling is not required for the Plgf effect on vascular remodeling (New Figure 1R-U).

New Figure 1R-U:

In vivo confocal imaging of aISV in WT (R); *mflt1*^{-/-} mutant, *mflt1*^{ka605} (S); *mflt1*^{-/-} combined with *plgf* gain of function, *plgf*^{musc} (T). (U) Quantification of images in R-T. **p<0.01, *** p<0.001.

Instead our data suggest that Flt1 (both mFlt1 and sFlt1) acts as a “sink” or decoy receptor for Vegf, capable of “storing” the Vegf produced by the developing somites surrounding the developing aISVs. In the Plgf gain of function scenario (*Plgf^{musc}* transgenic), excessive Plgf displaces Vegf from arterial Flt1, thereby resulting in a local Vegf gain of function scenario. Vegf activates Kdr1 signaling, inducing endothelial cell proliferation and enlargement of endothelial cells. The proposed signaling cascade derived from these observation is: Plgf gain of function, displacement of Vegf from Flt1, increased Vegf bio-availability – stimulation of Kdr1 signaling – activating Trio/Rac1, driving actin remodeling – larger vessel. To substantiate the Vegf-R2 signaling cascade we now show that reducing *vegfaa* using a *vegfaa* ATG targeting MO, inhibiting R2-tyrosine kinase activity using 2 different dosages, or reducing *Trio* expression significantly reduced the diameter increase in *Plgf^{musc}* (**New Figure 2A-F & new Fig. 3G-T**). We found a dose dependent reduction in aISV diameter upon inhibiting Kdr1/R2 signaling. With the high R2 inhibitor dosage we obtained about 90%-95% reduction in arterial growth. In addition, reducing *vegfaa* using a low dose *vegfaa* ATG targeting morpholino reduced lumen diameter growth in *plgf^{musc}* (**new Figure 2A-F**). Although both inhibition strategies reduced the Plgf effect, there was still some residual 5-10% diameter growth left (see dotted lines in panels 2E,F). This may either be due to incomplete blockade of the Vegf-Kdr1 signaling pathway with the used inhibitor or morpholino dosage. Alternatively this residual diameter growth was due to another signaling receptor beyond mFlt1, or Kdr1/R2, like neuropilin-1 (Nrp1).

New Figure 2A-F: (A-D) In vivo confocal imaging of aISV in WT (upper panels) and *plgf^{musc}* embryos (lower panels) treated with DMSO vehicle - control (A), low dose Vegf-R2 inhibitor (B), high dose Vegf-R2 inhibitor (C), or *vegfaa* ATG targeting morpholino (D). Red dotted box is displayed at higher magnification in panel below. **(E)** Quantification of aISV diameter at 50hpf upon R2 inhibition. Mean±s.e.m, ANOVA, n=15-20 for each scenario. **p<0.01, ***p<0.001. **(F)** Quantification of aISV diameter at 48hpf upon morpholino mediated knockdown of *vegfaa*. Mean±s.e.m, ANOVA, n=15-20 for each scenario. **p<0.01, ***p<0.001. Dotted line indicates average diameter in WT scenario.

Plgf can bind to Nrp1 receptors (*Migdal et al, J Biol Chem, 1998*). To address a potential involvement of Nrp1, we examined *plgf* gain-of-function in *nrp1a* mutants and found significantly enlarged aISVs (**new Suppl Fig. 3h**). However, besides *nrp1a*, zebrafish express an additional *nrp1* orthologue, *nrp1b* which may be involved as well. To prove that the observed residual 5-10% of diameter growth involves neuropilin-1, requires investigating Plgf in *nrp1a^{-/-};nrp1b^{-/-}* double mutants, and *nrp1a^{-/-};nrp1b^{-/-};mflt1^{-/-}* triple mutants; the breeding of which takes > 1 year, which is beyond the current revision. Our data suggest that Kdr1 is the main signaling receptor responsible for the Plgf effect. We added a paragraph addressing the potential role of Nrp1 and mFlt1 in mediating effects of Plgf overexpression.

-----response continued on the next page -----

We furthermore demonstrate that morpholino mediated knockdown of endogenous Trio results in a dose-dependent decrease of aISV diameter and EC surface area in *plgf^{musc}* embryos (**new Fig. 3G-T**).

New Figure 3G-T: (G-I) Confocal imaging of aISV in WT (G), WT injected with 1.7 ng *Trio* targeting morpholino (H), and WT injected with 5 ng *Trio* targeting morpholino (I) in *Tg(kdr:has.HRAS-mCherry)^{S916}*. **(J-L)** Confocal imaging of aISV in *plgf^{musc}* (J), *plgf^{musc}* injected with 1.7 ng *Trio* targeting morpholino (K), and *plgf^{musc}* injected with 5 ng *Trio* targeting morpholino (L) in *Tg(kdr:has.HRAS-mCherry)^{S916}*. **(M)** Quantification of images in G-L. Mean±s.e.m, ***p<0.001. **(N-P)** Confocal imaging of aISV in WT (N), WT injected with 1.7 ng *Trio* targeting morpholino (O), and WT injected with 5 ng *Trio* targeting morpholino (P) in *Tg(fli1a:lifeactEGFP)^{S916}*. **(Q-S)** Confocal imaging of aISV in *plgf^{musc}* (Q), *plgf^{musc}* injected with 1.7 ng *Trio* targeting morpholino (R), and *plgf^{musc}* injected with 5 ng *Trio* targeting morpholino (S) in *Tg(fli1a:lifeactEGFP)^{S916}*. **(T)** Quantification of N-S. Mean±s.e.m. **p<0.01, ***p<0.001.

We furthermore substantiated the *in vitro* data and show that Trio influences the intracellular distribution of Tiam1, and Tiam1 gain-of-function results in larger endothelial cells (**new Figure 4w**). There is no additive effect of Trio and Tiam1 overexpression. We furthermore show that the Trio-GEF1 target RhoG contributes to endothelial cell size changes, acting downstream of Trio (**new Figure 4m**). Finally we substantiated Trio dependent Vegf induced Rac1 activation with a different Trio targeting shRNA (shTrio-2) (**new Supplement Fig. 8j,k**).

Taken together: our data suggest that Plgf gain of function results in displacement of Vegf from Flt1, thereby augmenting Vegf bio-availability, resulting in stimulation of Kdr1 signaling. Our data suggest that Vegf-Kdr1 is the main driver, accounting for at least 90-95% of the observed diameter growth. We found no evidence for Plgf-mFlt1 signaling in diameter remodeling. *Nrp1a^{-/-}* mutants showed diameter growth in response to Plgf. Analysis in *nrp1a,nrp1b* double mutants, and *nrp1a,nrp1b,mflt1* triple mutants will show to what extent *nrp1b* may account for the small residual diameter growth observed upon blockade of Vegf-Kdr1. Plgf driven diameter growth *in vivo* requires Trio, as morpholino mediated targeting of Trio in *Plgf^{musc}* (**Figure 3m,t**) or inhibition of the Trio-GEF1 domain with ITX3 (**Figure 3f**) reduced diameter remodeling and EC area in the Plgf gain of function scenario.

Question 6. Testing potential application of the Rac1-Trio mediated regulation of arteriogenesis in the hyperglycemia zebrafish model is interesting, but done in a cursory manner. It appears that this part of the manuscript was simply added on to increase the applicability of the authors' finding. For instance, the authors need to determine whether Rac1/TrioN manipulation has any additional impacts on hyperglycemic zebrafish such as altering the metabolism of arterial endothelial cells.

Answer: we agree with the reviewer that a more in depth analysis of the metabolism in hyperglycemic embryos would strengthen our findings. Measuring in particular the metabolism of hyperglycemic arterial endothelial cells of zebrafish embryos would greatly improve the applicability of the model. Technically this requires detailed bio-chemical analysis of glycolysis & citric acid cycle products, as well as fatty acid metabolism. In order to have it arterial cell specific, arterial cells need to be FAC sorted from zebrafish embryos prior to performing such biochemical analyses. Pending the sensitivity of the bio-chemistry assays, you would probably require >10.000 48hpf embryos to sort sufficient numbers of arterial endothelial cells. The point of the reviewer is well taken, but at present, we feel it is beyond our technical possibilities to address this in a timely manner. We removed the hyperglycemia data from the current manuscript.

Minor comments:

Remark 1. It is not clear whether FLT1 is indeed a decoy receptor for VEGF signaling. Although it has a high affinity to VEGF-A ligand with a significantly less robust kinase activity compared to other VEGF receptors, it is nonetheless triggers phosphorylation of downstream targets. Therefore, it may be inappropriate to label FLT1 as a decoy receptor.

Answer: we agree with the reviewer; some reports suggest that mFlt1 signaling, via release of nitric oxide promotes angiogenesis, and therefore regard mFlt1 as a pro-angiogenic receptor (*Ahmad et al, Circ. Res. 2006*). In the revised manuscript we have now included analyses of *mflt1* mutants. We adapted the text of the manuscript and address the contribution of mFlt1 signaling versus Flt1 acting as a Vegf trap, in our setting. We furthermore included the reports suggesting mFlt1 as pro-angiogenic signaling receptor in our revised manuscript.

Remark 2. p7, tree should be three.

Answer: we corrected this typing error.

Remark 3. Administration of LatB would be too harsh treatment. The authors should provide additional way to manipulate F-actin polymerization.

Answer: we agree with the reviewer that most actin inhibitors are harsh. As alternative to LatB we assessed cytochalasin-B in our *in vitro* setting. We find that treating with cyto-B significantly reduced the *TrioN* induced increase in cell size.

Fig. Quantification of EC surface area upon Cytochalasin B (1 ng/mL) treatment; note significantly smaller cells. mean±s.e.m, Students *t*-test, n=96-107 cells/group.

Response to Reviewer #3:

Reviewer #3 (Remarks to the Author):

I am not an expert in arterial remodelling but am commenting on issues related to Rac1.

As far as I could judge it, the employed Rac1 mutants (active and light-sensitive) and the procedure for assaying active, GTP-bound Rac1 are fine. However, there are some issues concerning the use of controls and the lack of data on possible effects on other members of the Rho family of small GTPases, particularly Rac1b and RhoG. Specifically, my concerns are as follows:

Response: We like to thank reviewer 3 for the constructive remarks that helped us to substantiate the involvement of Rac1 and Trio isoforms. In addition we substantiated the role of Tiam1 and of RhoG.

Question 1: The Rac1 inhibitor used significantly inhibited aISV diameter expansion in plgfmusc embryos, suggesting that Rac1 was involved. However, this conclusion was based solely on results with the Rac1 inhibitor. Pharmacological inhibition likely co-inhibits also a splice isoform of Rac1, termed Rac1b, that may have quite different sometimes even antagonistic effects compared to Rac1. The authors should check if Rac1b is expressed in arterial endothelial cells (to the best of my knowledge there are no data available on expression of this isoform in blood vessels or arteries). If this is the case, the authors should go on and verify whether it has a functional role in cell shape changes in endothelial cells. Only if this possibility can be dismissed, the observed effects can be safely attributed to Rac1.

Answer to Question 1: We agree with the reviewer that in human, the *RAC1* gene encodes for two highly homologous splice variants termed *RAC1* and *RAC1B*. The splice isoform *RAC1B* includes an additional exon 3b comprising a 19 amino acid in-frame insertion (*Orlichenko et al, J Biol Chem, 2010*).

In line with reviewer's suggestion we compared *RAC1* and *RAC1B* expression levels in human endothelial cells derived from 5 different regions using publically available RNA deep sequencing databases (from: Angiogenes, Mueller et al, Scientific Reports, 6, (2016): <https://doi.org/10.1038/srep32475>).

Suppl. Table 1: Expression of RAC1 and RAC1B in human endothelial cells of different origins.

Transcript ID	BECs	HAoEC	HHSEC	HMVEC-D	HUVEC
ENST00000348035	131.345±9.35	82.970±36.61	33.578±0.00	155.024±0.00	89.816±38.24
ENST00000356142	Not detected	Not detected	Not detected	Not detected	35.099±12.25

ENST00000348035 encodes for *RAC1*

ENST00000356142 encodes for *RAC1B*

(BEC, blood vascular endothelial cell; HAoEC, human aortic endothelial cell; HHSEC, human hepatic sinusoidal endothelial cell; HMVEC, human lung microvascular endothelial cell; HUVEC, human umbilical vein endothelial cell).

RAC1B was only detected in HUVEC endothelial cells, not in endothelial cells of 4 other origins including the aorta (HAoEC). In HUVEC, *RAC1B* expression was lower than *RAC1* expression (Table 1). Overexpression of *RAC1B* had no impact on the size of HUVEC. Conversely, knock-down of *RAC1B* in HUVEC overexpressing *TrioN* had no significant impact on *TrioN* induced increase in cell size (**New Supplement Figure 8C,E,F**). From these *RAC1B* loss and gain of function experiments we conclude that human *RAC1B* is not involved in endothelial cell shape changes in our setting.

New Supplement Figure 8C,E,F: (C) Quantification of EC surface area after transfection with control plasmid, or with *RAC1B*. Note no significant upon *RAC1B*. At least 25 cells were quantified per experiment. The experiment was carried out 3 independent times. Mean±s.e.m, *t*-test. ns, not significant. (E) Western blot for *RAC1B* in endothelial cells upon control siRNA and si*RAC1B* transfection shows reduced *RAC1B* protein levels. Actin for protein loading control. (F) ECs were silenced for *RAC1b* or control siRNA as indicated in (E) and subsequently both populations of cells were transfected with *TrioN*. Loss of *RAC1B* had no impact on *TrioN* induced EC cell size increase. N=45 cells / 3 separate experiments. Mean±s.e.m, *t*-test. ns, not significant.

Regarding zebrafish: we found no evidence for alternative splicing of the zebrafish *rac1* gene into a *rac1b* like splice isoform with homology to human *RAC1B*. Using multiple sequence analyses and comparing the human *RAC1* exon3 region encoding the additional 19AA, with comparable regions of the known *rac1* orthologues in zebrafish, revealed no homology (New Suppl. Fig. 8D). The *RAC1* gene derived splice-isoform *RAC1B* was only found in human and mouse species, however not in chicken, drosophila or zebrafish (Boueux et al, Mol Biol Evol, 2007). These data suggest that *RAC1b* is a mammalian specific splice-isoform that is not conserved in zebrafish. We therefore conclude that is unlikely that a *Rac1b* isoform played a role *in vivo* in zebrafish vessels.

New Supplement Figure 8D: comparison of Rac1b in different species. Multiple sequence analysis of human, mouse, chicken, drosophila and zebrafish Rac1. Aligned were the protein structures of human *RAC1* with *Rac1* in Mouse, Chicken, Zebrafish and Drosophila. Due to gene duplication zebrafish express two *rac1* co-orthologues: these genes are termed *rac1a* and *rac1b* – these are two different genes and not splice products. The zebrafish gene termed *rac1b*, should therefore not be confused with human *RAC1b*, which results from alternative splicing of the *RAC1* gene. The exon3 region encoding *RAC1B* in human and mouse is color indicated. Note that the *RAC1B* protein isoform contains an additional 19 amino acids (VGETYGRDITSRGKDKPIA, indicated by red box). This isoform was only observed in human and mouse, not in zebrafish, chicken or drosophila. Except for the exon3b region, a high level of protein homology was observed between *Rac1* of the different species consistent with an evolutionary conserved functional role for *Rac1*.

Question 2 & 3. (2) In Fig. 3f both the *Rac1* and *Trio* inhibitors reversed the plgfmusc effect (relative to WT) by approx. 40%. In light of this (only) partial effect, several questions have arisen to me: (3). The authors showed that ITX3, a selective inhibitor of the *Trio* N-terminal RhoGEF domain, significantly reduced the outward remodeling response of aISVs. However, through the N-terminal DH-PH unit *Trio* can also mediate GDP to GTP exchange on RhoG (for Ref. see doi: 10.4161/cam.21418). This dual specificity provokes the question of whether RhoG also has a function in changes of endothelial cell shape. Did the authors consider this possibility or did they even perform any experiments to rule it out?

Answer: we agree with the reviewer that the GEF1 domain of *Trio* can activate RhoG. We therefore examined the impact of RhoG loss and gain of function of endothelial size (New Figure 4M,P). Overexpression of a dominant active RhoG form (*GFP-RhoG-Q61L*) significantly increased

endothelial cell area. Conversely, silencing of RhoG (*shRhoG-461*) in endothelial cells overexpressing *TrioN*, significantly reduced the *TrioN* induced cell size increase. We conclude that, in agreement with the suggestions made by the reviewer, that RhoG could play a role in Trio induced endothelial cell enlargement.

New Figure 4M,P: (M) Surface area of ECs transfected with control plasmid or constitutive active RhoG Q61L. Note overexpression of RhoG increases EC size. (P) Changes in EC size upon *TrioN* gain of function after silencing of RhoG. Mean \pm s.e.m, *t*-test, $n=101-146$ cells/group. Note: *TrioN* induced EC enlargement requires RhoG.

Question 4. The Rac1 inhibitor used interferes with the interaction between Rac1 and Trio but also between Rac1 and Tiam1. Did the authors analyse a possible involvement of Tiam1 in their system?

Answer: in line with the reviewer's suggestion we examined the contribution of Tiam1. In endothelial cells overexpressing *TrioN*, we observed that Tiam1 was expressed in junctional regions, as opposed to the more cytosolic localization observed in control transfected cells (Figure 4q-v). Endothelial overexpression of *Tiam1* resulted in larger endothelial cells, similar to the effects observed upon overexpression of *TrioN* (Figure 4w). Combining *TrioN* and *Tiam1* overexpression had no additive effect on increasing EC size when compared to *TrioN* alone (Figure 4w). These data suggest that one part of cell enlargement may require careful spatial positioning of other GEFs. How Trio achieves this spatial positioning of other GEFs is an exciting topic, but we feel it is beyond the scope of the current manuscript.

New Figure 4Q-W: (Q-S) Endothelial cells transfected with control plasmid (ctrl) and immune stained for endogenous Tiam1 (Q), F-actin with phalloidin (R) and VE-Cadherin as junction marker (S). Tiam1 localizes cytosolic and to a lesser extent to junction regions. Scale bar, 20 μm . (T-V) Endothelial cells transfected with *TrioN* and immune stained for endogenous Tiam1 (T), F-actin with phalloidin (U) and VE-Cadherin as junction marker (V). Tiam1 localizes at junction regions (arrowheads). Scale bar, 20 μm . (W) Endothelial cell size of ECs transfected with *Tiam1* (red bar), *TrioN* (green bar), or transfected with both *Tiam1* and *TrioN*. Measured were the cell-size of at least 20 cells derived from 3 separate experiments. Data are mean \pm SEM. Mean \pm s.e.m, *t*-test. ns, not significant, $***p<0.001$. Scale bar, 20 μm .

Question 5. There are several isoforms of Trio (Trio A, B, C, D, and E). Did the authors analyze which of these isoforms was responsible for the increase in endothelial cell size?

Answer:

In human, besides full length Trio, 5 distinct Trio protein isoforms (A-E) have been described. We showed that the GEF1 domain is critical for EC enlargement. Since Trio-E is lacking this GEF1 domain, we ruled out a contribution of Trio-E. We next performed a Western blot with an antibody directed against the spectrin domain (Antibody rbt CT232), allowing detection of full length Trio and the A-D isoforms (**new Suppl. Fig. 8A,B**). We found that full length Trio was the most abundant Trio form. The isoforms A/D and B-C were detectable, albeit at lower levels. The Trio-B isoform is truncated at the C-terminus and similar to TrioN, only contains the GEF1 domain. In support of the GEF1 domain, endothelial overexpression of Trio-B increased endothelial

New Suppl. Fig. 8A,B: (A) Western Blot with an antibody directed against the spectrin domain for Trio showing full-length TRIO (most upper band) and indicated TRIO splice-isoforms TRIO-A/D, TRIO-C and TRIO-B. Size marker (kDa) on the left. (B) Quantification of EC surface area after transfection with control plasmid, or with TRIO-B. Note the significant increase in EC size upon TRIO-B. Mean±s.e.m, *t*-test. ****p*<0.001.

size (**new Suppl. Fig. 8A,B**). We found that full length Trio was the most abundant Trio form. The isoforms A/D and B-C were detectable, albeit at lower levels. The Trio-B isoform is truncated at the C-terminus and similar to TrioN, only contains the GEF1 domain. In support of the GEF1 domain, endothelial overexpression of Trio-B increased endothelial

size (**new Suppl. Fig. 8A,B**). Since Full-length Trio, Trio-B and Trio-A,C,D are homologous for their spectrin and GEF1 domain, we propose that Trio-D/A/C may also have the capacity to mediate endothelial cell enlargement. However, based on abundance, it is most likely that full length Trio is the major form responsible for modulating endothelial cell shape.

Response to Reviewer #4:

Reviewer #4 (Remarks to the Author):

The current manuscript investigates methods of expanding vascular lumens using Zebrafish intersomitic vessels as a model system. They generate multiple new lines that aid in protein localization and that increase bioavailable VEGF_{aa}. Increasing VEGF_{aa} in three experimental ways results in expanded ISV diameters in a VEGFR2-dependent, VEGFR3/flow-independent manner. The cellular mechanism is by EC proliferation and cell size increase. The authors also perform cell biology experiments showing that TrioN/actin dynamics also increase vessel diameter through increasing cell size. Finally, simultaneously increasing VEGF_{aa} and activating TrioN has an additive effect resulting in functional vessels with 2.5X the normal size. Methods of regulating vessel size could lead to clinically relevant ways of increasing blood flow. It is known that VEGF can increase vessel diameters, but the current study more thoroughly examines this in an *in vivo* setting. It is also not surprising that the known modulators of the actin cytoskeleton Trio and Rac1 increases cell size, but the authors test this in an intact vasculature.

Response: We would like to thank reviewer 4 for the helpful suggestions to improve our data sets. We performed additional experiments in *flt4* and *ve-cadherin* mutants, substantiated our *in vitro* data and now show how RhoG may contribute to Trio induced EC enlargement. In addition we performed the requested control experiments and substantiated quantifications.

Question 1. The data demonstrating that vessels are non-leaky is hard to interpret and not quantified. Can the authors show larger views in Fig. 1M and include a quantification of the amount of dextran present in the perivascular space? The claim is also made in Fig. 6 for Plgf+TrioN without data or quantifications.

Answer: In line with the reviewer's suggestion, we quantified the dextran extravasation in the *TrioN*, and *Plgf*^{musc} gain of function scenarios. We adapted Fig1M and present it with a larger view. See new Figure 1O-Q, and Supplement Figure 11A-D for TrioN.

New Figure 10,P: (O,P) Imaging of plasma extravasation in WT and *plgf*^{musc} injected with 70kD Dextran Texas-Red. **New Suppl. Fig. 11D: (D)** Quantification of plasma extravasation. mean±s.e.m; *t*-test, n=6-8 embryos/group. ns, not significantly different.

New Suppl. Fig. 11A-C: (A-C) Imaging of plasma extravasation in *flt1*^{enh}; *TrioN* injected with 70kD Dextran Texas-Red.

Question 2. There is no effect with some morpholinos. The authors should confirm that the genes are indeed knocked down and that the lack of effects are not due to insufficient depletion (for *flt4* and *ve-cadherin*).

Answer: to rule out insufficient depletion, we repeated all experiments in *flt4*^{-/-} mutants, and in *ve-cadherin*^{-/-} mutants respectively. We obtained similar results in the mutants as with the morphants. For *flt4*^{-/-} mutants: see **new Figure 2g-k**. For *ve-cadherin* (*cdh5*) mutants see **new Suppl. Fig. 11e**.

Question 3. The authors claim that there is F-actin reorganization evident between control and experimental conditions in Fig. 3d,e and movies 2 and 3. It was not clear how they were measuring this and what they exactly observed.

Answer: we display the *in vivo* life-act images and the schematic representation of the actin remodeling events *in vivo* at the cell level in **Supplement Figure 6g**. We removed the statement on F-actin reorganization. Instead we used the *in vitro* system to address and quantify the F-actin remodeling events in endothelial cells (**new Figure 5E,F**). – see also response to question 8.

Time lapse imaging of F-actin dynamics in *TrioN*-transfected cells showed formation of lamellipodia, and enlargement of the endothelial cells along the front of the lamellipodia extensions (**Supplement Fig. 9e-g; Supplement movie 6**). Control-transfected cells also showed lamellipodia formation, but in control cells these lamellipodia more often retracted and no subsequent enlargement was observed (**Supplement Fig. 9h-j; Supplement movie 7**). Quantification of lamellipodia protrusion lifetime showed that *TrioN* significantly increased the lifetime of lateral lamellipodia when compared to transfected cells; accordingly the number of protrusions per hour was reduced in *TrioN* transfected cells (**new Figure 5e,f**).

New Figure 5E,F: (E) Kymograph illustrating the actin cytoskeleton dynamics along the indicated line in images on left, in time: in control transfected cells and *TrioN* transfected cells. Scale bar, 20 μ m. (F) Quantification of protrusion lifetime (left panel). *TrioN* (green bar) significantly increased the lifetime of lateral lamellipodia compared to control conditions. Mean \pm s.e.m, *t*-test, n=50-110 cells/group. Quantification of protrusion dynamics expressed as protrusion number per hour (right panel). *TrioN* (green bar) significantly decreased protrusion dynamics. Mean \pm s.e.m, *t*-test, n=50-110 cells/group. ****p*<0.001.

Question 4. The proper way to display deviations with the type of data in this paper is through error bars that indicate standard deviation, not SEM, which is in most graphs. Can the authors correct this?

Answer: Since the standard error takes sample size into account in addition to sample variability, and we will properly state in legends to figures that we use mean \pm sem where appropriate, we may be allowed to keep our 72 panels that way. For experiments with non-normal distribution we applied non-parametric tests (Mann Whitney U, Kruskal Wallis test) and show the distribution of individual values where appropriate.

Question 5. Fig. 3f, inhibitor studies.

5a. The experiment does not have WT alone with drugs. These conditions should be included to properly conclude the effect of the treatment. Is the phenotype reversal merely because this treatment decreases diameter or is it due to specific increases in cytoskeletal activity with Plgf?

5b. The picture of the ISVs in this experiment are very small fields of view. Can the authors show the entire ISV as in Fig. 2? This will show whether the vessels normal except for decreased size or whether the decreased sizes are secondary to massive structural defects.

5c. If the authors are suggesting that the cytoskeletal rearrangements inhibited here are leading to cell size, they should measure cell size with the inhibitor treatments.

Answer: in line with the reviewer's questions, we increased the field of view of the treated aISVs, and added the WT alone with drugs.

New Supplement Figure 6H-P:(H-K) In vivo confocal imaging of aISVs in WT treated with vehicle control (H), with Latrunculin-B (LatB) (I), with Rac1 inhibitor CAS 1177865-17-6 (J), with GEF Trio inhibitor ITX3 (K). (L-O) In vivo confocal imaging of aISVs in *plgf^{musc}* treated with vehicle control (L), with Latrunculin-B (LatB) (M), with Rac1 inhibitor CAS 1177865-17-6 (N), with GEF Trio inhibitor ITX3 (O). (P) Diameter quantification of images in H-K. Mean±s.e.m, *t*-test. #*p*<0.05 versus WT-DMSO.

ITX3 is a highly specific pharmacological inhibitor of Trio function. To further substantiate the contribution of Trio in our setting we examined loss of *Trio* in WT and *plgf^{musc}* embryos using a morpholino approach (new Figure 3g-t). Morpholino mediated knockdown of Trio resulted in a dose-dependent decrease of endothelial cell size and aISV diameter in both WT and *plgf^{musc}* embryos (new Figure 3g-t).

New Figure 3G-T: (G-I) Confocal imaging of aISV in WT (G), WT injected with 1.7 ng *Trio* targeting morpholino (H), and WT injected with 5 ng *Trio* targeting morpholino (I) in *Tg(kdr1:has.HRAS-mCherry)^{S916}*. (J-L) Confocal imaging of aISV in *plgf^{musc}* (J), *plgf^{musc}* injected with 1.7 ng *Trio* targeting morpholino (K), and *plgf^{musc}* injected with 5 ng *Trio* targeting morpholino (L) in *Tg(kdr1:has.HRAS-mCherry)^{S916}*. (M) Quantification of images in G-L. Mean±s.e.m, ****p*<0.001. (N-P) Confocal imaging of aISV in WT (N), WT injected with 1.7 ng *Trio* targeting morpholino (O), and WT injected with 5 ng *Trio* targeting morpholino (P) in *Tg(fli1a:lfeactEGFP)^{S916}*. (Q-S) Confocal imaging of aISV in *plgf^{musc}* (Q), *plgf^{musc}* injected with 1.7 ng *Trio* targeting morpholino (R), and *plgf^{musc}* injected with 5 ng *Trio* targeting morpholino (S) in *Tg(fli1a:lfeactEGFP)^{S916}*. (T) Quantification of images in N-S. Mean±s.e.m. ***p*<0.01, ****p*<0.001.

Question 6. Fig. 3M. There are loading inconsistencies between control and Trio siRNA that could falsely accentuate differences between the two groups. They should re-run the gel with the proper ratios.

Answer: in our experience, we tend to see somewhat reduced Rac1 levels in total cell lysates in endothelial cells when Trio is silenced in conditions where the total protein loading is equal, based on staining for “household” proteins beta-actin and tubulin. However, quantifications have always been corrected for the input. When the Western blotting shows variability in the Rac1 staining, we have incorporated this in the quantification. Additionally, we have repeated the experiment using a second shRNA against Trio (**new Supplement Figure 8J,K**). We furthermore down-tuned our statement that Trio is the only activator of Rac1, as Vegf has been reported to be able to activate Rac1 through RhoG. In *RhoG* gain and loss of function experiments we now show that RhoG can augment endothelial cell size, similar to Trio; and silencing of RhoG reduces the effect of TrioN on cell size (**new Figure 4M,P**).

Question 7. Fig. 4b and c, suppl. Fig. 4a. The FRET signal does not appear to be concentrated at junctions, and the authors definition of junctional in this panel does not match with their definition in Fig. 4I. Do the authors have a junctional protein that can serve as a positive control? If not, they should not claim the signal is junctional.

Answer: In line with the reviewer’s suggestion, we replace the old image with a new image showing the endothelial junction marker VE-cadherin as positive control (**new Figure 5A,B**). When combining the FRET signal with the VE-Cadherin signal, it is clear to see that the Rac1 biosensor is activated at junction regions (compare panel 5b’ with 5b’’’, yellow arrowheads). We have quantified the FRET ratios at these areas and took an area from the cells’ edge, positive for VE-cadherin and 10 pixels inwards. We defined these areas as junction regions. We replace the old figure with a Rac1 sensor image including the junction marker VE-Cadherin. In addition we rephrased the text in the methods section to make this item more clear.

New Figure 5A,B: TrioN activates Rac1 at junction regions. (a-a''') HUVEC transfected with mCherry-control and co-transfected with the Rac1 biosensor (Cer3, cerulean3 channel, a'). FRET signals are depicted as warm colour according to heat map (a). VE-cadherin, to label endothelial junctions, was stained with a directly labelled ALEXA647 antibody (a'''). (b-b''') HUVEC transfected with mCherry-TrioN and the Rac1 biosensor (b'), FRET signal increased (b'), in particular at junction regions (yellow arrowheads) and co-localized with VE-cadherin staining (b'''). Scale bar indicates 20µm.

Question 8. There is no quantification for the findings in Fig. 4 g-i and the accompanying suppl. Figure panels.

Answer: Time lapse imaging of F-actin dynamics in *TrioN*-transfected cells showed formation of lamellipodia, and enlargement of the endothelial cells along the front of the lamellipodia extensions (**Supplement Fig. 9e-g; Supplement movie 6**). Control-transfected cell also showed lamellipodia formation, but in control cells these lamellipodia more often retracted and no subsequent enlargement was observed (**Supplement Fig. 9h-j; Supplement movie 7**). **Quantification of lamellipodia protrusion lifetime** showed that *TrioN* significantly increased the lifetime of lateral lamellipodia when compared to transfected cells; accordingly the number of protrusions per hour was reduced in *TrioN* transfected cells (**new Figure 5e,f**). The cell-size increases were quantified in Figure 4G,W.

New Figure 5E,F: (E) Kymograph illustrating the actin cytoskeleton dynamics along the indicated line in images on left, in time: in control cells and *TrioN* transfected cells. (F) Quantification of protrusion lifetime & # of protrusions/hour. *TrioN* significantly increased the lifetime of lateral lamellipodia and decreased protrusion dynamics, compared to control conditions. Mean±s.e.m, *t*-test, n=50-110 cells/group.

Question 9. There is no control for comparison for data in Suppl. Fig. 4e.

Answer: In line with the reviewer's suggestion, we substantiated this part and show the appropriate controls. When using GFP-tagged Myosin II we observe that Myosin II localizes on F-actin bundles throughout the cell, but it is most prominently present on junctional F-actin bundles (**new Figure 5G,H**). Also, the majority of active myosin, distinguished by staining for mono-

phosphorylated myosin light chain S19, localizes strongly at the junction region (**new Figure 5I,J**). In addition, we have observed that fully di-phosphorylated myosin light chain (T18/S19) localizes even more specifically to junctional F-actin bundles (**new Figure 5K,L**), in line with active myosin II at the junction regions relevant for building up junctional tension.

New Figure 5G-L: (G) EC transfected with *mCherry* and *GFP-MyosinII*, and stained for F-actin (magenta) and VE-cadherin (white). Myosin-II localizes at actin bundles (arrowheads). (H) EC transfected with *mCherry-TrioN* and *GFP-MyosinII*, and stained for F-actin and VE-cadherin. Myosin-II localizes at junctional actin bundles (arrowheads). (I) EC transfected with *GFP* and stained for pMLC-S19, F-actin and VE-cadherin. pMLC-19 localizes at actin bundles throughout the cell (arrowheads). (J) EC transfected with *GFP-TrioN*, and stained for pMLC-S19, F-actin and VE-cadherin. pMLC-19 localizes at junctional and peripheral actin bundles (arrowheads). (K) EC transfected with *GFP*, and stained for pMLC-T18S19 (red) and F-actin. (L) EC transfected with *GFP-TrioN*, and stained for pMLC-T18S19 and F-actin. pMLC-18/19 at junctional regions (arrowheads).

Question 10. Fig. 5h-k. What is the biological significance of this panel and how does this result influence their model? Based on the focal adhesions shown in 5f and the known function of integrins, shouldn't the integrins be localized there instead of cell-cell junctions.

Answer: In an attempt to understand how Trio "translates" its downstream effects into increased cell-size, and since additional experiments (**Figure 6d-m**) showed that VE-cadherin was not required for increased cell-size (VE-cadherin-deficient ECs still managed to increase their cell-size upon Trio activation), we speculated that cell enlargement might involve integrin distribution.

Therefore, we studied the localization of Integrin beta1 and Integrin alpha5, known to mediate focal adhesions, and endothelial contact with the ECM. We noticed that upon Trio activation, the distribution of the integrins shifted from a well-known focal adhesion pattern towards a more junctional pattern (**Figure 6T-W**). This was also in line with the shift in localization of paxillin, a well-recognized marker for focal adhesions. Also, paxillin shifted from the focal adhesions to the junction regions (**Figure 6O-S**). These data indicate that integrins maybe involved in the regulation of Trio-mediated increase in cell-size by controlling ECM contacts near junctions as a means to keep the enlarged cell in its shape and preventing it from "collapsing". We have added a part in the discussion where we speculate on integrins may contribute to the Trio response.

Minor points

MP1. Fig. 1E, sFlt1 seems to be heterogeneously expressed in ECs. Can the authors comment?

Answer: we agree, the images suggest heterogeneity. There is a lot of debate in the angiogenesis community regarding the distribution of soluble Flt1 and membrane Flt1 in vascular and non-vascular tissues. Lack of high-quality whole mount Flt1 immune-stainings thus far obscured detailed mapping of the expression domains of the different Flt1 isoforms at the tissue/cellular level. To address this question we therefore generated a set of tools including the sflt1-d7-HA transgenic, and the Flt1 HA-tag knockin. We agree with the reviewer that the immune-staining shows a degree of heterogeneity: we hypothesize that polarization in the secretion of soluble Flt1 (luminal/ab-luminal) may contribute to this heterogeneous expression pattern. Alternatively, slight differences in Notch signaling status (Notch is proposed to be upstream of Flt1 expression) – causing small differences in local sFlt1 expression - may be involved. Understanding the regulation of sFlt1 secretion, and the physiological consequences of sFlt1 distribution on vascular morphogenesis (diameter, branching patterning) are exciting topics that we are currently trying to address with colleagues investigating sprout target selection, but believe are beyond the context of the current manuscript.

MP2. There are citation problems throughout the text. Please check and make sure all panels in suppl. are cited.

Answer: we apologize for the citations problems. We have corrected them.

Reviewers' Comments:

Reviewer #1:

Remarks to the Author:

This is a very substantially revised manuscript, more focused and with much tighter mechanistic analysis. The authors have addressed nearly all of the previous criticisms in a highly constructive manner. That includes removing the data and discussion that pertains to adult remodeling in diabetes. There remains one important issue: the authors continue to relate these findings to adult remodeling without any recognition that early zebrafish angiogenesis differs in fundamental ways. In early zebrafish embryos, mural cells are absent and vessel size is determined solely by the endothelium. These *in vivo* results compare well to endothelial-only cultures. But the results cannot be extrapolated to more complex systems. It is essential that these limitations be fully acknowledged and discussed.

Minor comment:

Fig 10. Contrary to what it claims in the text, the amount of N-cad at cell-cell junctions is not apparently affected by VE-cadherin depletion and only moderately by expression of Trio. I don't see this as a major issue but the conclusion needs to match the data.

Reviewer #2:

Remarks to the Author:

The authors addressed all of my previous concerns. This manuscript will certainly help us to understand how vessel morphogenesis regulated during development.

Reviewer #3:

Remarks to the Author:

The authors have satisfactorily responded to my critique. I appreciate the author's efforts to clarify the roles of Rac1b, RhoG, Tiam and the additional Trio isoforms. I am particularly delighted that my comments helped to reveal RhoG and Tiam to contribute to the regulation of endothelial cell size in their system.

My only criticism in the revisions is that the immunoblot shown in the new figure 8E is of poor quality. The authors claimed that Rac1b abundance was reduced upon its siRNA-mediated knockdown compared to irrelevant control siRNA, however, signal intensities were not quantified densitometrically and differences confirmed by statistical evaluation.

Hendrik Ungefroren

Reviewer #4:

Remarks to the Author:

The authors answered all my concerns to a satisfactory level. Most responses were met with additional experimental data, which was particularly appreciated given that they had 4 sets of comments in which to respond.

Manuscript Number: **NCOMMS-19-564295A**

Response to the Reviewers

We thank the reviewers and the editorial board of Nature Communication for their constructive feedback and suggestions.

Reviewer #1 (Remarks to the Author):

Remark: This is a very substantially revised manuscript, more focused and with much tighter mechanistic analysis. The authors have addressed nearly all of the previous criticisms in a highly constructive manner. That includes removing the data and discussion that pertains to adult remodeling in diabetes. There remains one important issue: the authors continue to relate these findings to adult remodeling without any recognition that early zebrafish angiogenesis differs in fundamental ways. In early zebrafish embryos, mural cells are absent and vessel size is determined solely by the endothelium. These in vivo results compare well to endothelial-only cultures. But the results cannot be extrapolated to more complex systems. It is essential that these limitations be fully acknowledged and discussed.

Answer: We agree with the reviewer that the impact of Trio on vessel diameter in mature adult arteries may differ from the effects we observed in developing arterial networks of zebrafish embryos. In line with the reviewer's suggestion we discuss and acknowledge the differences between adult and embryonic vascular networks.

We added the following text: "In early zebrafish embryos, mural cells are absent and vessel size is determined solely by the endothelium. In adult resistance sized arteries the presence of several layers of smooth muscle cells, and sympathetic nerve induced vascular tone may restrict the impact of Trio on diameter remodeling. It therefore seems unlikely that activating Trio and promoting EC size, can directly affect arterial diameter independent of vascular smooth muscle function in mature resistance sized arteries."

Minor comment:

Fig 10. Contrary to what it claims in the text, the amount of N-cad at cell-cell junctions is not apparently affected by VE-cadherin depletion and only moderately by expression of Trio. I don't see this as a major issue but the conclusion needs to match the data.

Answer: In line with the reviewer's suggestion we removed this statement from the discussion, and results section.

Reviewer #2 (Remarks to the Author):

Remark: The authors addressed all of my previous concerns. This manuscript will certainly help us to understand how vessel morphogenesis regulated during development.

Answer: We thank reviewer 2 for supporting the publication of our manuscript.

Reviewer #3 (Remarks to the Author):

Remark: The authors have satisfactorily responded to my critique. I appreciate the author's efforts to clarify the roles of Rac1b, RhoG, Tiam and the additional Trio isoforms. I am particularly delighted that my comments helped to reveal RhoG and Tiam to contribute to the regulation of endothelial cell size in their system.

My only criticism in the revisions is that the immunoblot shown in the new figure 8E is of poor quality. The authors claimed that Rac1b abundance was reduced upon its siRNA-mediated

knockdown compared to irrelevant control siRNA, however, signal intensities were not quantified densitometrically and differences confirmed by statistical evaluation.

Hendrik Ungefroren

Answer: We thank reviewer 3 for the kind words and the support for the publication of our manuscript. In line with the reviewer's question, we now quantified the signal intensities and report the corresponding statistical evaluation – see supplementary figure 8f.

Reviewer #4 (Remarks to the Author):

The authors answered all my concerns to a satisfactory level. Most responses were met with additional experimental data, which was particularly appreciated given that they had 4 sets of comments in which to respond.

Answer: We thank reviewer 4 for supporting the publication of our manuscript.